# Fusion pore dynamics of large secretory vesicles define a distinct mechanism of exocytosis

Tom Biton[1,2], Nadav Scher[1], Shari Carmon[2], Yael Elbaz-Alon[1], Eyal D. Schejter[2], Ben-Zion Shilo[2], and Ori Avinoam[1]

**Exocrine cells utilize large secretory vesicles (LSVs) up to 10 μm in diameter. LSVs fuse with the apical surface, often recruiting actomyosin to extrude their content through dynamic fusion pores. The molecular mechanism regulating pore dynamics remains largely uncharacterized. We observe that the fusion pores of LSVs in the *Drosophila* larval salivary glands expand, stabilize, and constrict. Arp2/3 is essential for pore expansion and stabilization, while myosin II is essential for pore constriction. We identify several Bin-Amphiphysin-Rvs (BAR) homology domain proteins that regulate fusion pore expansion and stabilization. We show that the I-BAR protein Missing-in-Metastasis (MIM) localizes to the fusion site and is essential for pore expansion and stabilization. The MIM I-BAR domain is essential but not sufficient for localization and function. We conclude that MIM acts in concert with actin, myosin II, and additional BAR-domain proteins to control fusion pore dynamics, mediating a distinct mode of exocytosis, which facilitates actomyosin-dependent content release that maintains apical membrane homeostasis during secretion.**

## Introduction

The fusion of secretory vesicles with the plasma membrane is essential for the exocytotic release of bioactive materials such as neurotransmitters, hormones, and digestive enzymes from neuronal, endocrine, and exocrine cells. The latter customarily utilize large secretory vesicles (LSVs), with diameters that are one to two orders of magnitude larger than those of neuronal or endocrine vesicles, challenging the conventional mechanisms of vesicle exocytosis and recycling.

Secretory vesicles dock and fuse with the cell surface, connecting the vesicle lumen and the extracellular space via a fusion pore (Breckenridge and Almers, 1987; Curran et al., 1993; Hastoy et al., 2017; Sharma and Lindau, 2018). The formation of a fusion pore at the contact site between the vesicle and cell membrane is mediated by soluble N-ethylmaleimide-sensitive fusion attachment protein receptors (SNAREs; Söllner et al., 1993a, 1993b; Südhof and Rothman, 2009; Fang et al., 2008; Ngatchou et al., 2010; Wiederhold et al., 2010). Fusion pores are initially nanometric and can flicker open and reseal spontaneously several times before stabilizing, expanding, or constricting (Chanturiya et al., 1997; Curran et al., 1993; Vardjan et al., 2013; Álvarez de Toledo et al., 2018). If the fusion pore reseals, the vesicle is recycled en bloc, retaining some of its cargo (i.e., kiss-and-run [Alvarez de Toledo et al., 1993; Rizzoli and Jahn, 2007]). If the pore expands beyond a certain diameter, the vesicle collapses and integrates into the surface, spilling out its cargo, and the added membrane is retrieved by endocytosis at, or near, the site of exocytosis (i.e., full collapse

[Rizzoli and Jahn, 2007]). Hence, fusion pore dynamics define the release kinetics and the mode of exocytosis and vesicle recycling.

Several studies from flies to mammals have demonstrated that LSV exocytosis differs from exocytosis of smaller vesicles. When LSVs fuse with the apical surface, they recruit an actomyosin meshwork that contracts on the LSV membrane and extrudes the cargo (Valentijn et al., 2000; Sokac et al., 2003; Turvey and Thorn, 2004; Nemoto et al., 2004; Yu and Bement, 2007a, 2007b; Segawa and Yamashina, 1989; Nightingale et al., 2011, 2012; Tran et al., 2015; Rousso et al., 2016). We have shown that actomyosin contractility on the LSV membrane squeezes out the content, without integrating the vesicle into the apical surface (Kamalesh et al., 2021; Fig. 1 A). Consequently, the vesicle neither collapses into nor detaches from the surface, suggesting that LSVs utilize a distinct mode of exocytosis (which we termed "membrane crumpling"). We have also shown that following fusion, the diffusion between the vesicle and apical membranes becomes quickly restricted, permitting only limited initial membrane mixing, while predominantly maintaining the composition of the apical membrane. These observations suggested that membrane homeostasis in exocrine cells is maintained by mechanochemical sequestration of the LSV membrane (Kamalesh et al., 2021; Rousso et al., 2016; Fig. 1 A).

To elucidate the mechanism of exocytosis by membrane crumpling, we used the *Drosophila melanogaster* larval salivary gland (SG) as a model for LSV exocytosis (Biyasheva et al., 2001;

[1]Department of Biomolecular Sciences, Weizmann Institute of Science, Rehovot, Israel; [2]Department of Molecular Genetics, Weizmann Institute of Science, Rehovot, Israel.

Correspondence to Ben-Zion Shilo: benny.shilo@weizmann.ac.il; Ori Avinoam: ori.avinoam@weizmann.ac.il.

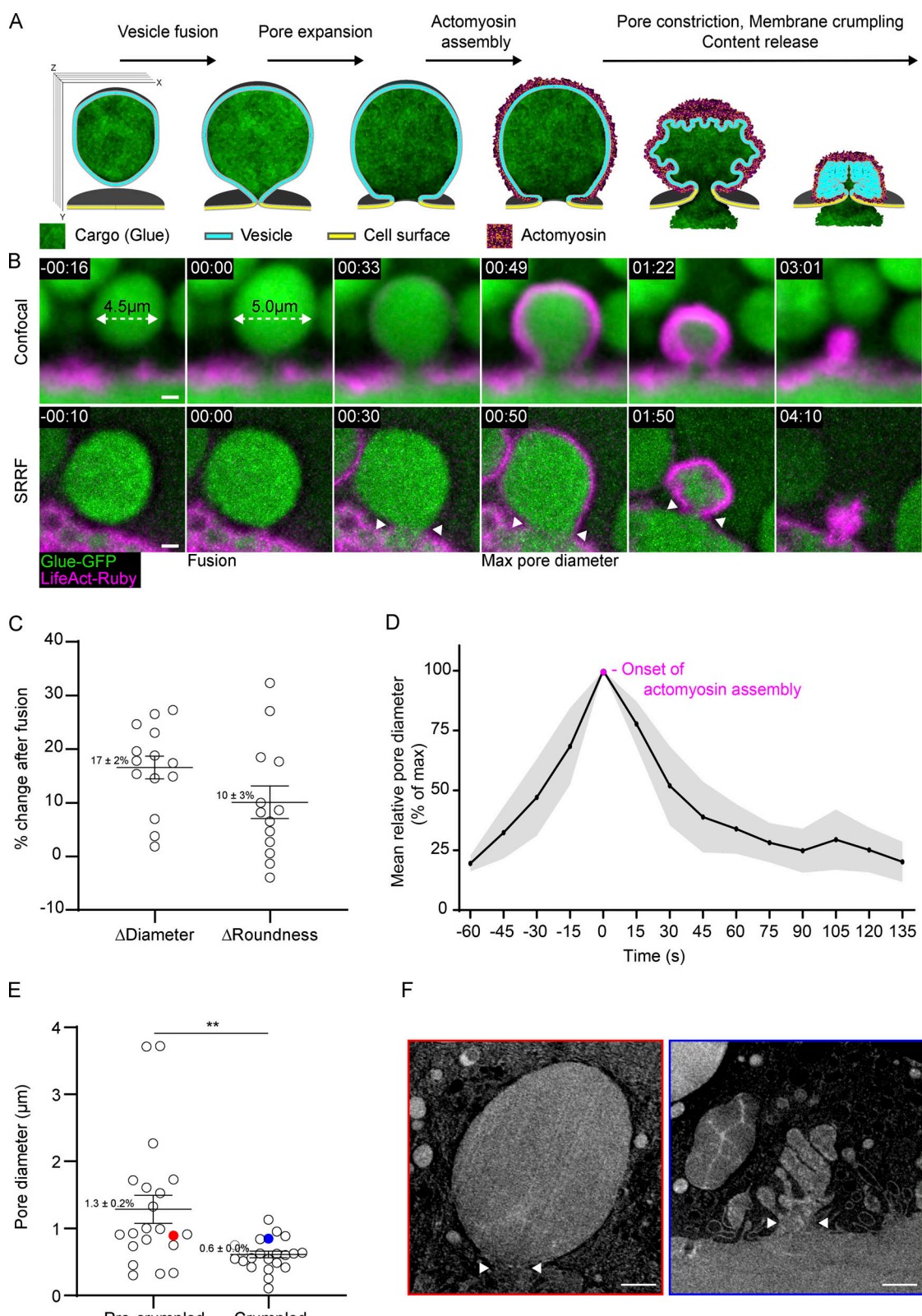

Figure 1. **The fusion pores of LSVs expand and constrict. (A)** Schematic representation illustrating the process of LSV exocytosis by vesicle membrane crumpling. Actomyosin recruitment to LSVs after fusion mediates membrane deformation and subsequent content release. Images were obtained via confocal microscopy in multiple planes, as illustrated with the unfused vesicle (left). For fusion pore visualization, we selected LSVs where the fusion pore is aligned to the XY imaging plane. The presented view was generated by capturing a stack of XY planes and subsequently projecting them along the Z plane. **(B)** Time-lapse sequence of representative exocytic events (Confocal, top; SRRF, bottom). Fusion onset is detected by LSV swelling (white dashed arrow; Fig. S1 A). LifeAct-Ruby (actomyosin) recruitment is first visible ~50 s after fusion and followed by content release. Fusion pore diameter was defined and measured at the

narrowest aperture observed at the mid-plane through the vesicle, which connects the vesicle and the lumen (white arrowheads). The fusion pore expands before actomyosin recruitment and constricts during content release. Time mm:ss, relative to fusion. Glue-GFP (green) and LifeAct-Ruby (magenta) images were generated by averaging several XY planes along the Z axis (intensity-projection). **(C)** Quantification of LSV swelling as % change in LSV diameter and roundness after fusion (n [LSVs] = 14), mean and SEM values are noted on the panel. **(D)** Mean relative pore diameter (% maximal diameter) over time. Time 0 is defined at the time of maximal pore diameter, which coincides with the onset of actomyosin assembly (n [pores] = 12; gray—SEM; see also Fig S1 B). **(E)** Pore diameter of fused vesicles visualized by FIB-SEM before and after crumpling (n [pores]) = 21 and 22 respectively; P: ** = 0.0026. Two-tailed, unpaired t test. **(F)** Red and blue circles denote the individual LSVs. Mean and SEM values are noted on the panel. **(F)** Representative slices from FIB-SEM volumes of the mid-plane of LSVs before (red) and after (blue) crumpling. White arrowheads show the narrowest aperture of the fusion pore where it was measured. Whiskers in C and E show mean and SEM. Scale bars in B and F, 1 µm.

Rousso et al., 2016; Kamalesh et al., 2021). The cargo of these vesicles are adhesive mucinous glycoproteins nicknamed "Glue," which are used by the fly larva to attach to a solid surface before metamorphosis. Recruitment of active (GTP-bound) Rho1 to the vesicle membrane after fusion triggers two parallel events that are essential for actomyosin meshwork formation (Rousso et al., 2016): activation of Diaphanous (Dia), a member of the formin family of actin nucleators, which generates linear actin strands that are visible within ~30 s after fusion; and activation of RhoA-kinase (ROCK), leading to the phosphorylation of myosin II light chain and activation of myosin motors on the actin network (Shemesh and Kozlov, 2007; Jaffe and Hall, 2005; Rousso et al., 2016; Segal et al., 2018; Kamalesh et al., 2021).

We now find that after LSVs fuse, they remain connected to the apical surface through dynamic fusion pores that expand to diameters larger than 1 µm and then constrict back to 100 nm or less (Kamalesh et al., 2021; Rousso et al., 2016). Utilizing pharmacological and genetic perturbations, we demonstrate that branched actin is essential for pore expansion and stabilization. Conversely, myosin II is required for pore constriction. When the fusion pore fails to stabilize, vesicle content is released by full collapse in the absence of actomyosin. This highlights the importance of stabilizing the fusion pore to prevent full collapse, which then allows for the recruitment and contraction of an actomyosin meshwork to mediate exocytosis by membrane crumpling. Genetic screening identified several conserved Bin-Amphiphysin-Rvs homology (BAR) domain proteins that control fusion pore dynamics and specifically revealed that the I-BAR-containing protein Missing-in-Metastasis (MIM) is essential for pore expansion and stabilization in a dose-dependent manner. MIM localizes to the fusion site on the vesicle before fusion and remains associated with the fusion pore throughout secretion. The I-BAR domain is essential but not sufficient for MIM localization and function. Collectively, our results suggest that LSV fusion pore behavior is tightly regulated by BAR domain proteins that act in concert with actomyosin and branched actin polymerization to facilitate exocytosis while maintaining apical membrane homeostasis. Moreover, they demonstrate that exocytosis by membrane crumpling is a distinct mode of vesicular secretion that depends on the structure and dynamics of the vesicle fusion pore.

## Results
### The fusion pores of LSVs expand before actomyosin assembly and constrict during content release
To follow the dynamic changes in LSV fusion pore diameter, we visualized individual vesicles during exocytosis in secreting SGs

expressing the content marker Sgs3-GFP (Biyasheva et al., 2001; Glue-GFP) under the endogenous sgs3 promoter and the F-actin probe LifeAct-Ruby (expressed via GAL4/UAS using the c135-GAL4 driver). We used the F-actin probe as an indicator for actomyosin assembly, given that F-actin polymerization precedes and is essential for myosin II localization to the LSV (Rousso et al., 2016; Fig. 1, A and B; and Video 1). The time of vesicle fusion was defined by the swelling and rounding of the LSVs, which occurs at the onset of fusion (Breckenridge and Almers, 1987; Fig. 1, B and C; and Fig. S1 A). To visualize and quantify changes in pore diameter during exocytosis, we used super-resolution radial fluctuation (SRRF) imaging, which improved the resolution and signal-to-noise ratio (Gustafsson et al., 2016; Fig. 1 B and Video 1).

After fusion, LSVs with a mean diameter of 5.3 ± 0.2 µm after swelling (n [pores] = 12) displayed pore expansion for 68 ± 6 s, reaching a maximum mean diameter of 1.6 ± 0.2 µm. This was followed by constriction to diameters below the detection limit, typically under 100 nm. The onset of actomyosin assembly begins when pores reach their maximal diameter and pore constriction is accompanied by actomyosin contraction and content release, lasting altogether 121 ± 9 s (Fig. 1, B and D; Fig. S1 B; and Video 1).

To directly visualize the fusion pore at different stages of exocytosis, we used Focused Ion Beam Scanning Electron Microscopy (FIB-SEM) and obtained three-dimensional, high-resolution information from the apical areas of secreting SGs (Fig. 1, E and F). Using this approach, we identified fused LSVs and measured their fusion pore diameter. The pore diameter of early fused LSVs, before membrane crumpling, ranged between 0.2 and 3.7 µm, with a mean of 1.3 ± 0.2 µm (n [pores] = 21). In contrast, the pore diameter of crumpled LSVs ranged between 0.1 and 1.1 µm with a mean of 0.6 ± 0.04 µm (n [pores] = 22), consistent with the live imaging data showing that pores expand until actomyosin assembly initiates, and subsequently constrict during actomyosin-mediated membrane crumpling (Fig. 1, E and F). These observations show that the LSV fusion pore follows a typical sequence of expansion and constriction, suggesting that an active mechanism regulates fusion pore dynamics.

### Branched actin nucleation is essential for fusion pore expansion and stabilization
Pharmacological and genetic inhibition of actin polymerization in chromaffin cells, Xenopus eggs, pancreatic acinar cells, and Drosophila SGs influence content release after fusion (Sokac et al., 2003; Ñeco et al., 2004; Nemoto et al., 2004; Larina

et al., 2007; Yu and Bement, 2007b; Doreian et al., 2008; Rousso et al., 2016; Shin et al., 2018). To explore the regulatory role of branched actin polymerization in LSV fusion pore dynamics, we treated SGs with a mild dose of the Arp2/3 complex inhibitor CK666 (Ck666[100 μM]) or knocked down (KD) Arp3 expression (Arp3[KD]) by an RNA interference (RNAi) construct expressed under UAS control (Fig. 2, A and C; and Fig. S2 B). When the Arp2/3 complex was perturbed by either method, we observed frequent occurrences of fusion pores that failed to expand or expanded but failed to stabilize as compared with pores in control untreated SGs. These alterations in pore dynamics resulted in LSV behaviors that are reminiscent of both the full collapse and kiss-and-run canonical modes of exocytosis (Fig. 2, A–C; Fig. S2, A and B; and Videos 2 and 3).

Full collapse events are characterized by pores that expand irreversibly without stabilizing, leading to rapid and complete opening of the pore, apparent integration of the LSV into the apical membrane, and content release before actomyosin recruitment (Fig. 2, A and B; Fig. S2, A and B; and Videos 2 and 3, top panels). Kiss-and-run events are characterized by LSVs that fuse with the apical membrane but subsequently detach. During such events, LSVs will often undergo multiple cycles of fusion and detachment (detected by vesicle swelling), with or without actomyosin recruitment and contraction (Fig. 2, A and B; Fig S2, A and B; and Videos 2 and 3, middle and bottom panels). Actomyosin recruitment to vesicles with a narrow or sealed pore can deform and displace the vesicle from the apical membrane with no apparent content release (Fig. 2, A and B; Fig. S2, A and B; and Videos 2 and 3).

Interestingly, a detailed examination of multiple fusion events in WT glands revealed that while membrane crumpling is the prevalent mode of exocytosis, a significant fraction of fusion events exhibit a kiss-and-run-like behavior and a smaller cohort leads to full collapse or stalling of the LSV (67 ± 3%, 23 ± 4%, 6 ± 3% and 4 ± 1% respectively; Fig. 2 C and Fig. S2 A). Importantly, unlike full collapse and stalling, kiss-and-run events are not terminal, allowing LSVs to undergo subsequent fusion and exocytosis. As a result, the majority of kiss-and-run events in WT SGs eventually proceed to membrane crumpling.

We observed significantly more full collapse events in Arp3[KD] SGs, and an apparent, but non-significant, increase in full collapse events in CK666[100 μM]-treated SGs, compared with WT untreated controls (27 ± 3%, 19 ± 9%, and 6 ± 3%, respectively; Fig. 2 C). Moreover, we observed significantly more kiss-and-run events in Arp3[KD] and CK666[100 μM]-treated SGs compared with WT untreated controls (50 ± 6%, 42 ± 4%, and 23 ± 4%, respectively; Fig. 2 C). Treating SGs with a higher dose of CK666 (CK666[500 μM]) resulted in the accumulation of stalled, actin-coated LSVs at the apical surface, consistent with previous studies linking Arp2/3 activity with vesicle contraction (Tran et al., 2015; Rousso et al., 2016; Fig. 2 D and Video 4, left). These LSVs were stalled with narrow pores that arrest before completing pore expansion (mean arrested pore diameter 0.5 ± 0.02 μm, n [pores] = 10; Fig. 2, E and F; and Video 4, right).

Collectively, our results demonstrate that branched actin nucleation via the Arp2/3 complex is crucial for pore expansion and stabilization. Moreover, they show that only exocytosis by

membrane crumpling or full collapse results in content release, suggesting that pore expansion is essential for Glue secretion, while pore stabilization is essential for membrane crumpling.

## Myosin II is essential for fusion pore constriction downstream of pore expansion and stabilization

We have previously shown that perturbation of myosin II, either through RNAi-mediated KD of the *Drosophila* myosin II heavy chain homolog *Zipper* (Zipper[KD]), or treatment with the Rho-associated protein kinase (ROCK) inhibitor Y-27632 (100 μM), which blocks myosin II recruitment, results in fused but stalled vesicles (Kamalesh et al., 2021; Segal et al., 2018). To investigate the role of myosin II in regulating the fusion pore, we analyzed fusion pore dynamics under these conditions. Confirming our previous findings, both Zipper[KD] and ROCK inhibition induced significant LSV stalling (36 ± 4% and 70 ± 9% of fusion events, respectively; Fig. 3, A and B), consistent with the role of myosin II in actomyosin contractility and membrane crumpling.

We further examined fusion pore behavior in these stalled LSVs and observed slower expansion and constriction kinetics compared to WT, untreated glands (Fig. 3, B and C; and Video 5). The typical fusion pores in WT glands expand for 68 ± 6 s and rapidly constrict (n [pores] = 12). In contrast, in glands treated with the ROCK inhibitor (Y-27632[100 μM]), the pores expand significantly slower for 266 ± 38 s and remain expanded without constricting (n [pores] = 10). Similarly, the fusion pores of stalled LSVs in Zipper[KD] glands expand apparently slower for 111 ± 32 s and do not fully constrict (n [pores] = 6; Fig. 3 B and Video 5). Notably, the maximal pore diameter in untreated SGs was comparable with the mean maximal pore diameter of LSVs in Zipper[KD] glands and to the arrested diameter of the expanded pores in Y-27632[100 μM] treated glands (1.6 ± 0.1 μm, 1.4 ± 0.2 μm and 1.4 ± 0.1 μm, respectively; Fig. 3 B). Taken together, these results support and expand on previously proposed mechanisms of myosin II in fusion pore expansion and constriction (see Discussion).

To elucidate the temporal order of effects on pore dynamics, we employed simultaneous treatment with both the CK666[500 μM] and Y-27632[100 μM] inhibitors. Under these conditions, SGs presented stalled LSVs with pores that arrest before completing expansion (mean arrested diameter of 0.6 ± 0.05 μm, n [pores] = 9; Fig. 3 B), similar to the effect of the Arp2/3 inhibitor alone. These results suggest that myosin II activity is essential for efficient pore expansion and following branched actin polymerization, is also critical for pore constriction.

To visualize myosin II localization during membrane crumpling, kiss-and-run, and full collapse modes of exocytosis in WT SGs, we used Sqh-mCherry, a fluorescently tagged variant of the *Drosophila* myosin II light chain homolog *spaghetti squash (sqh)*, expressed under the endogenous *sqh* promoter (Fig. 3 D and Video 6). During membrane crumpling, myosin is recruited to the LSVs shortly after F-actin when the fusion pore is at its maximal diameter (Segal et al., 2018; Kamalesh et al., 2021; Fig. 1, B and D). Membrane crumpling and content release by actomyosin contractility will then initiate ~60 s after fusion (Segal et al., 2018; Kamalesh et al., 2021).

The full collapse and kiss-and-run modes of exocytosis were found to be associated with abnormal, albeit informative

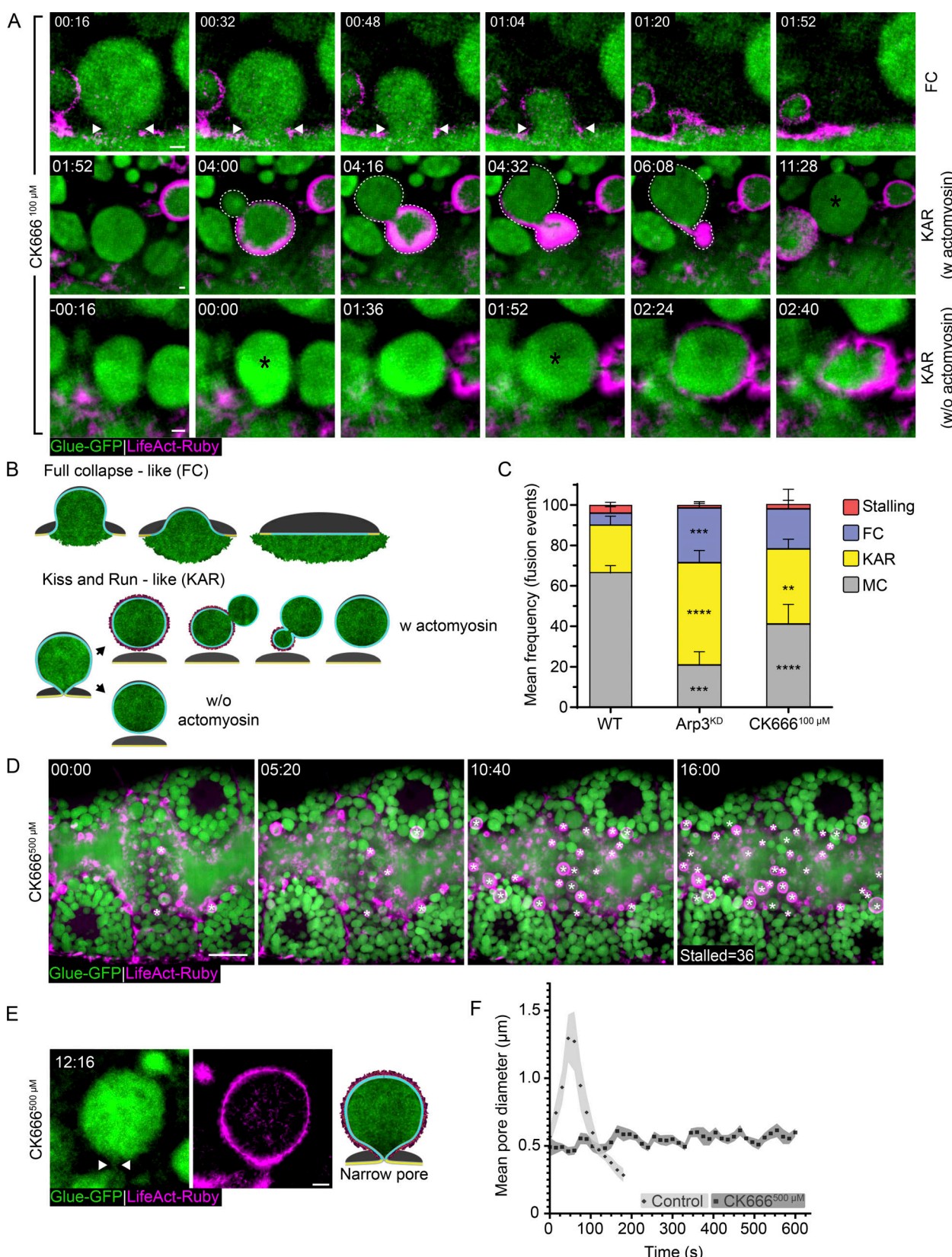

Figure 2. **Branched actin polymerization is essential for pore expansion and stabilization. (A)** Time-lapse sequence of representative LSVs from CK666[100 μM] SGs undergoing full collapse (FC) and kiss-and-run (KAR; SRRF intensity-projection; also in Video 2; see Fig. S2 and Video 3 for WT and Arp3[KD]). Top: FC appears as a content release that precedes actomyosin assembly. The fusion pore expands and the vesicle appears to integrate into the apical surface. Middle: KAR w actomyosin, appearing as deformation and displacement of the LSV from the apical membrane upon actomyosin contraction (dashed line

outlining the LSV during deformation). The same LSV fused again at 11:28, suggesting it detached between fusion events. Bottom: KAR w/o actomyosin, appearing as consecutive fusion events (asterisk) without content release. Actomyosin was recruited after the second fusion event (01:52), leading to membrane crumpling. **(B and C)** Schematic model of full collapse (FC) and kiss-and-run (KAR w. and w/o actomyosin) exocytosis (C) mean frequency (%) of stalling, FC, KAR, and membrane crumpling (MC) in WT, Arp3$^{KD}$, and CK666$^{100\ \mu M}$ SGs. Exocytosis by membrane crumpling is the dominant mode of exocytosis in WT SGs (67 ± 3%) with lower frequencies of FC and KAR and stalling (6 ± 3%, 23 ± 4%, and 4 ± 1% respectively). Arp3$^{KD}$ and CK666$^{100\ \mu M}$ present a higher frequency of full collapse (27 ± 3% (***) and 19 ± 9% (ns), respectively) and kiss-and-run (50 ± 6% (****) and 42 ± 4% (**), respectively). Error bars show SEM. Statistical significance with respect to WT frequencies. N (SGs) ≥ 3, n (events) ≥ 200. P values for CK666$^{100\ \mu M}$: **(KAR) = 0.004044, ****(MC) = 0.00002, for Arp3$^{KD}$ ***(FC) = 0.000199, ****(KAR) = 0.000007, ****(MC) < 0.000001. Two-tailed unpaired multiple *t* tests corrected using the Holm-Sidak method (also in Video 2). **(D)** Time-lapse sequence of a representative SG treated with CK666$^{500\ \mu M}$ (confocal intensity-projection). Stalled, actin-coated LSVs (white asterisks) accumulate at the cell apical membrane (also in Video 4, left). **(E)** Representative image of stalled LSVs with a narrow pore after CK666$^{500\ \mu M}$ treatment (also in Video 4, right). **(F)** Change in mean pore diameter over time in control and CK666$^{500\ \mu M}$ treated SGs showing that the pore expanded and constricted in control SGs (from Fig. S1 B; control, diamonds; mean maximal diameter = 1.6 ± 0.2 µm; n [pores] = 12; light gray = SEM) but arrest with a narrow diameter upon Arp2/3 perturbation (CK666$^{500\ \mu M}$; squares; mean arrested pore diameter = 0.5 ± 0.02 µm; n [pores] = 10; gray = SEM). Fusion = time 0. SGs express Glue-GFP (green) and LifeAct-Ruby (magenta) in A and C–E. RNAi expression in A and C under GAL4/UAS control. UAS expression is driven in A and C by *fkh*-GAL4, and in D and E by c135-GAL4. Time mm:ss in A and E, relative to fusion, in D relative to treatment. Scale bars in A and E, 1 µm; D, 20 µm.

patterns of myosin II recruitment (Fig. 3 D). Myosin II was initially absent from LSVs undergoing full collapse exocytosis, but was eventually recruited to the empty flattened LSV membrane, and displayed some contractile activity (Fig. 3 D and Video 6). During kiss-and-run exocytosis, myosin II was present on LSVs prior to their deformation and displacement from the apical surface. This suggests that actomyosin contractility is driving the displacement of LSVs with a narrow or resealed pore (Fig. 3 D and Fig. 2 A). These results demonstrate that myosin recruitment to the LSV membrane can take place regardless of fusion pore behavior, but that myosin is essential for pore constriction downstream of pore stabilization, which is necessary for actomyosin-mediated membrane crumpling.

## BAR domain proteins control pore expansion and stabilization

Several proteins have been implicated in the regulation of fusion pore dynamics in small exocytic vesicles, including the Bin-Amphiphysin-Rvs homology (BAR) domain–containing proteins Amphiphysin1, Syndapin2, and Endophilins, which localize to sites of exocytosis (Somasundaram and Taraska, 2018). Since BAR domain proteins associate with membrane curvature, they may represent putative regulators of fusion pore dynamics. Hence, we conducted a candidate-based genetic screen, using fly lines bearing transgenic RNAi constructs directed against genes encoding BAR-domain containing proteins, which are significantly expressed in secreting SGs (Fig. S3 A).

We observed that KD of five out of the eleven BAR-domain containing candidate genes we studied—Missing-in-metastasis (MIM$^{KD}$), Amphiphysin (Amph$^{KD}$), Sorting nexin 1 (SNX1$^{KD}$), Sorting nexin 6 (SNX6$^{KD}$), or Cdc42 interacting protein 4 (CIP4$^{KD}$)—resulted in an increased frequency of full collapse, kiss-and-run, or both. Interestingly, we also observed a substantial frequency of compound exocytosis events (in which vesicles fuse to a previously fused LSV), which are rarely observed in WT SGs, in SNX1$^{KD}$, SNX6$^{KD}$, Centaurin beta 1A$^{KD}$, and Syndapin$^{KD}$. SH3PX1$^{KD}$, Rho GTPase activating protein at 92B$^{KD}$, Nostrin$^{KD}$, and CG8176$^{KD}$ did not show a pore-related phenotype (Fig. 4 A, Fig. S3 B, and Video 7).

To further analyze the contribution of different BAR domains to the regulation of fusion pore dynamics, we focused on CIP4, SNX1, and MIM, which contain F-BAR, PX-BAR, and I-BAR domains, respectively (Fricke et al., 2009; Zhang et al., 2011;

Quinones et al., 2010). CIP4$^{KD}$ SGs displayed an increased frequency of kiss-and-run exocytotic events (53 ± 4%), suggesting that CIP4 plays a role in pore expansion (Fig. 4 B and Fig. S3 B). In contrast, SNX1$^{KD}$ and MIM$^{KD}$ SGs displayed an increased frequency of full collapse events (51 ± 4% and 46 ± 16%, respectively), suggesting that SNX1 and MIM are important for pore stabilization (Fig. 4 B and Fig. S3 B). Consistent with this observation, treating MIM$^{KD}$ SGs with CK666$^{500\ \mu M}$ rescued the full collapse phenotype by stalling the LSVs with a narrow pore similar to the effect of CK666 alone, indicating that pore expansion precedes pore stabilization (Fig. 2 E and Fig. 4 C). In contrast, treating MIM$^{KD}$ or SNX1$^{KD}$ SGs with Y-27632 did not alter their full collapse phenotype, demonstrating that myosin II is required for pore constriction downstream of BAR domain proteins, which are essential for pore stabilization at wide diameters (Fig. 3 B and Fig. 4 D). Furthermore, when the pore failed to stabilize, as in MIM$^{KD}$ and SNX1$^{KD}$, vesicle content release was completed even in the absence of actomyosin (Fig. S3 B). These results show that BAR domain proteins are essential for regulating pore dynamics, controlling pore expansion and stabilization.

## MIM localizes to the fusion site and the pore throughout the secretion

To determine the localization of CIP4 and SNX1, we examined larval SGs from fly lines expressing CIP4-EGFP (Fricke et al., 2009) under an ectopic UAS promoter and SNX1-GFP under its endogenous promoter. CIP4-EGFP mostly localized to the apical membrane and SNX1-GFP faintly labeled the apical and lateral membranes of the cells, as well as the LSV membranes. Neither CIP4-EGFP nor SNX1-GFP showed noticeable enrichment at the fusion pore (Fig. S4, A and B). However, this observation does not fully rule out the possibility that the endogenous proteins do localize to the fusion pore.

To determine the localization of MIM, we tagged it with fluorescent proteins and generated fly lines bearing a functional MIM-Emerald (Fig. S5 B) or MIM-mScarlet under an ectopic UAS promoter. Upon expression in SGs, we observed that both proteins localized to dynamic and mobile cytoplasmic clusters that often merged or divided into smaller clusters (Fig. 5 A, Fig. S4 C, and Video 8). Strikingly, we observed small fluorescent MIM clusters that localized to the space between the vesicle and

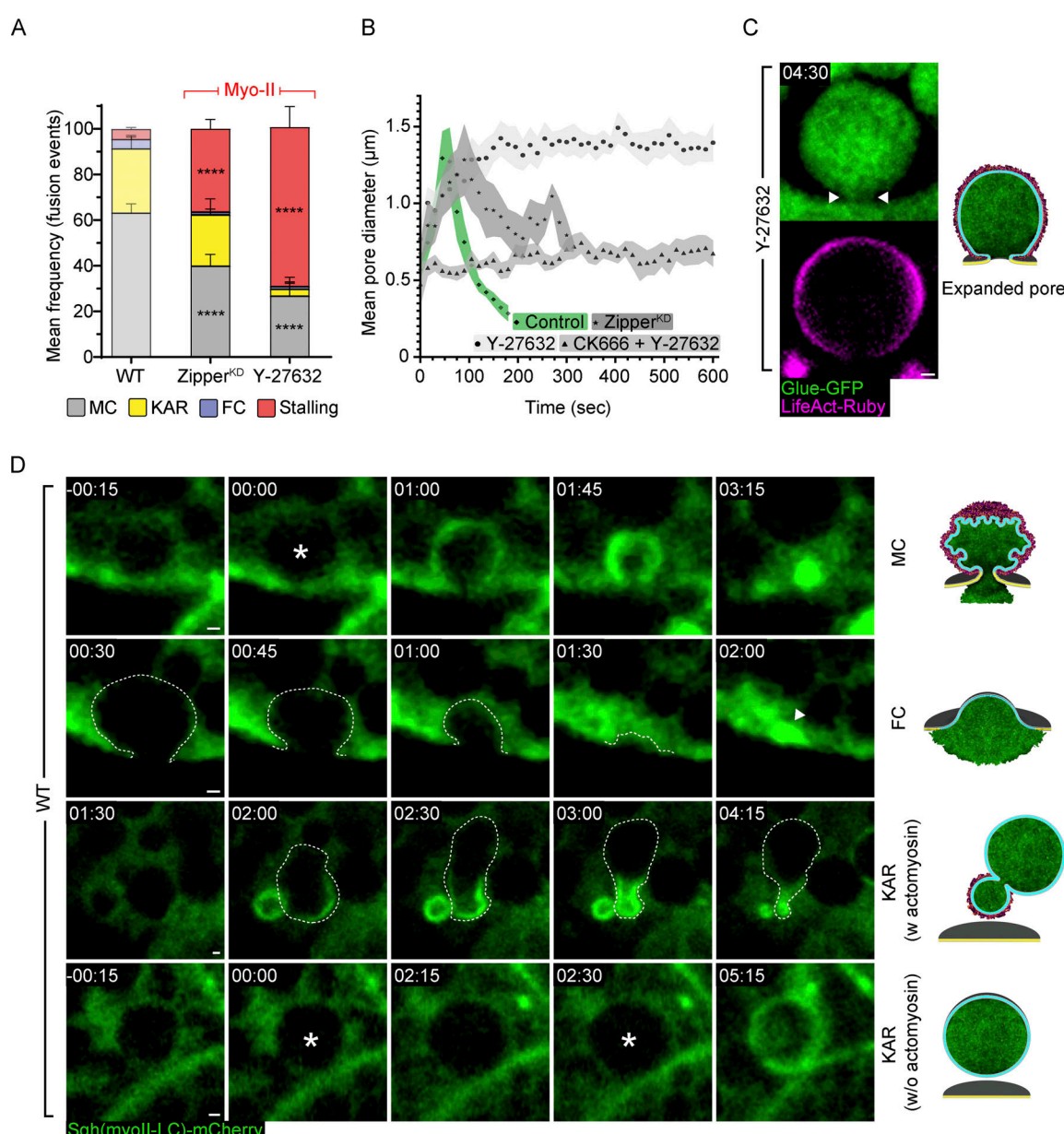

**Figure 3.** **Myosin II is essential for efficient pore expansion and constriction. (A)** Mean frequency (%) of stalling, full collapse (FC), kiss-and-run (KAR), and membrane crumpling (MC) in WT untreated, Zipper[KD] (myosin II heavy chain), and SGs treated with the ROCK inhibitor Y-27632[100 µM]. Exocytosis by membrane crumpling is the dominant mode of exocytosis in WT SGs (from Fig. 2 C, semi-transparent). Zipper[KD] and Y-27632[100 µM] treatment resulted in significant LSV stalling (36 ± 4% (****) and 70 ± 9% (***) of fusion events, respectively). Error bars show SEM. Statistical significance with respect to WT, untreated frequencies. N (SGs) ≥ 3, n (events) ≥ 140. P values: **** (stalling and MC) < 0.0001. Two-tailed unpaired multiple t-tests corrected using the Holm-Sidak method. **(B)** Change in mean pore diameter over time in WT untreated (control, diamonds; n [pores] = 12; from Fig. 2 F; green; mean maximal pore diameter = 1.6 ± 0.1 µm) compared to Zipper[KD] (stars; n [pores] = 6; mean maximal pore diameter = 1.4 ± 0.2 µm), Y-27632[100 µM] (circles; n [pores] = 10; mean arrested pore diameter = 1.4 ± 0.1 µm) or Y-27632[100 µM] + CK666[500 µM] (triangles; n [pores] = 9; mean arrested pore diameter = 0.6 ± 0.05 µm) SGs. In the presence of myosin II perturbation, the fusion pore exhibited slower expansion (WT - 68 ± 6 s, and Y-27632[100 µM] - 266 ± 38 s(***), Zipper[KD] - 111 ± 32 s(ns), P values = 0.0002 and 0.323, respectively) and constricted partially (Zipper[KD]) or remained expanded (Y-27632[100 µM]). Perturbation of both Arp2/3 and myosin resulted in a failure of pore expansion, leading to an arrested narrow pore (Y-27632[100 µM] + CK666[500 µM]). Fusion = time 0. Significance calculated using two-tailed, non-parametric, unpaired t tests using the Kolmogorov–Smirnov method. Green or shades of gray = SEM. **(C)** Representative images of stalled LSVs with wide pores after Y-27632[100 µM] treatment. Glue-GFP (green) and LifeAct-Ruby (magenta; also in Video 5). **(D)** Time-lapse sequence of representative LSVs in WT SG expressing Sqh-mCherry (myosin II light chain; green) undergoing membrane crumpling (MC, top), full collapse (FC, second row), kiss-and-run (KAR w or w/o actomyosin, third and fourth rows; also in Video 6). LSVs are visible in the background of the cytoplasmic Sqh-mCherry signal (dashed line in FC) and vesicle swelling severs as a proxy for fusion as in Glue-GFP (asterisk). In FC, Sqh-mCherry is recruited after vesicle integration (arrowhead; 02:00). In KAR w actomyosin, vesicles undergo deformation and displacement from the apical membrane upon Sqh-mCherry recruitment (dashed line; 02:00–04:15). In KAR w/o actomyosin, vesicles fuse multiple times before recruiting Sqh-mCherry (00:00 and 02:30) before proceeding to membrane crumpling. SGs express Glue-GFP and LifeAct-Ruby in A–C. UAS expression driven by c135-GAL4. Time mm:ss in C and D relative to fusion. Scale bars in C and D, 1 µm.

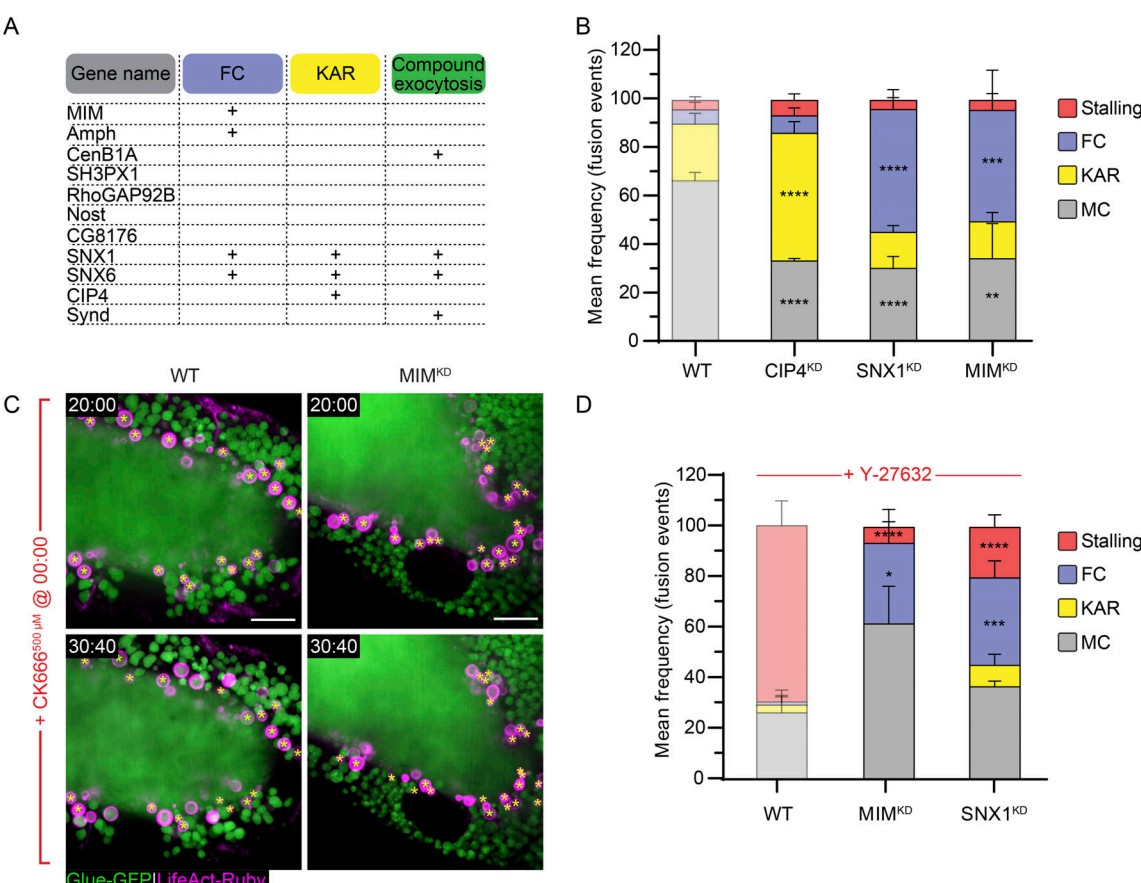

Figure 4. **BAR domain proteins regulate the fusion pore. (A)** Qualitative phenotypic RNAi screen for BAR domain-containing proteins. MIM[KD], Amph[KD], SNX1[KD], and SNX6[KD] showed frequent occurrences of full collapse (FC) events; SNX1[KD], SNX6[KD], and CIP4[KD] displayed frequent occurrences of kiss-and-run (KAR) events; and CenB1A[KD], SNX1[KD], SNX6[KD], and Synd[KD] showed events of compound exocytosis. **(B)** Mean frequency (%) of stalling, FC, KAR, and membrane crumpling (MC) in CIP4[KD], SNX1[KD], and MIM[KD]. CIP4[KD] resulted in an increase of kiss-and-run event frequency (53 ± 4% (****)), while SNX1[KD] and MIM[KD] displayed an increase of full collapse event frequency (51 ± 4% (****) and 46 ± 19% (***), respectively). Error bars = SEM. The statistical significance presented is compared with WT SGs. $N$ (SGs) ≥ 3, $n$ (events) ≥ 150. P values for CIP4[KD]: ****(KAR and MC) <0.00001, for SNX1[KD]: ****(FC and MC) <0.000001, for MIM[KD]: ***(FC) = 0.000111, **(MC) = 0.001267. Two-tailed, unpaired multiple $t$ tests corrected using the Holm-Sidak method. **(C)** Time-lapse sequence of representative CK666[500 μM] treated WT (left) and MIM[KD] (right) SGs (confocal intensity-projection). Glue-GFP (green) and LifeAct-Ruby (magenta). Inhibiting the Arp2/3 complex in MIM[KD] resulted in LSV stalling. Asterisks denote stalled LSVs. Time mm:ss; relative to CK666 treatment. Scale bars = 20 μm. **(D)** Mean frequency (%) of stalling, FC, KAR, and MC modes in Y-27632 treated WT, MIM[KD], and SNX1[KD] SGs. Compared with stalling in Y-27632-treated WT SGs (70 ± 9% of fusion events), significantly less stalling was observed, and full collapse persists at a high frequency in Y-27632 treated MIM[KD] (6 ± 6% (****) and 32 ± 8% (*), respectively) and SNX1[KD] (20 ± 4% (****) and 35 ± 6% (***), respectively) SGs. Error bars = SEM. The statistical significance presented is compared with WT-treated SGs. $N$ (SGs) ≥ 3, $n$ (events) ≥ 150. P values for MIM[KD]: ****(Stalling) = 0.000022, *(FC) = 0.011445, for SNX1[KD]: ****(Stalling) = 0.000006, ***(FC) = 0.000441. Two-tailed, unpaired multiple $t$ tests corrected using the Holm-Sidak method. WT and WT Y-27632 treated (from Fig. 2 C and Fig. 3 A; semi-transparent) are shown again for comparison convenience in B and D. SGs in A–D express Glue-GFP and LifeAct-Ruby. RNAi expression under UAS control. UAS-based expression in A driven by *fkh*-GAL4, in B and C by *c135*-GAL4.

the apical surface before fusion and remained at or in the vicinity of the fusion pore for the duration of exocytosis (Fig. 5 A, Fig. S4 C, and Video 9).

To determine whether MIM-Emerald localizes to fusion sites, we used correlative confocal and FIB-SEM imaging. We targeted cells that showed MIM-Emerald puncta close to the apical surface in secreting SGs. In these cells, MIM-Emerald was specifically detected at the vesicle membrane in the region that interacts with the apical cell surface, even before fusion and formation of the fusion pore (Fig. 5 B). Cumulatively, our data suggest that different BAR domain proteins regulate distinct steps in fusion pore dynamics and that MIM localizes specifically

to the fusion site on the vesicular membrane and subsequently to the fusion pore.

## MIM is essential for efficient pore expansion and stabilization in a dose-dependent manner

To further elucidate the role of MIM in pore regulation, we visualized pore dynamics under progressive MIM loss-of-function conditions. To this end, we crossed the chromosomal deficiency line Df(2R)BSC260, bearing a large deletion that includes the MIM gene locus (MIM[Def]) with WT flies, to generate SGs with a single functional copy of MIM (MIM[+/Def]). We did not observe a significant variation in exocytosis in MIM[+/Def] SGs compared

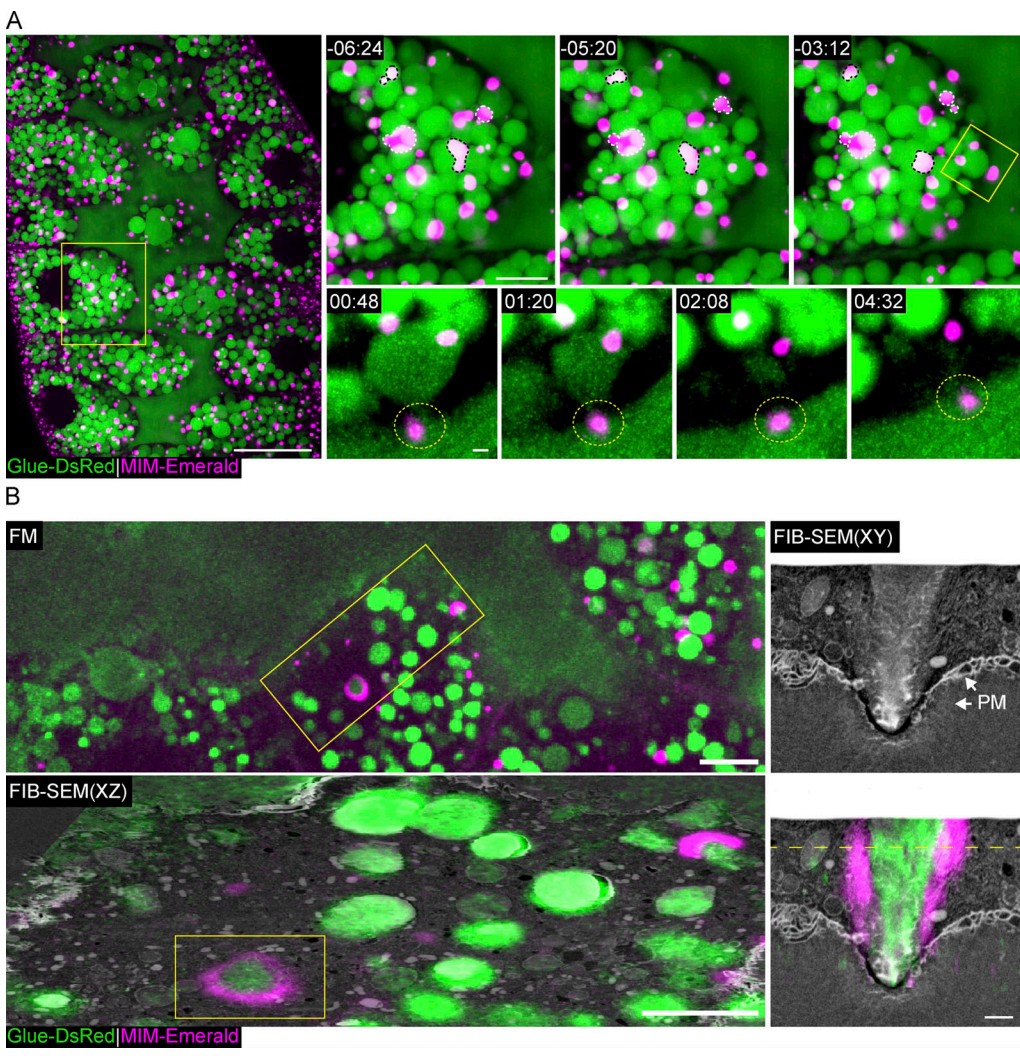

**Figure 5. MIM localizes to the site of fusion and pore formation. (A)** Confocal intensity-projection of a representative SG expressing Glue-DsRed (Costantino et al., 2008; green; expressed under the endogenous *sgs3* promoter), and a functional MIM-Emerald (magenta; Fig. S5 B), showing at increased magnification the dynamic localization of MIM to cytoplasmic and apical clusters. Yellow squares mark the magnified areas. Top right: Dynamic MIM cytoplasmic clusters merging (black dashed lines) and splitting (white dashed lines; also in Video 8). Bottom right: A representative LSV (SRRF intensity-projection) during exocytosis, showing MIM localization to the sites of fusion and pore formation (dashed circle) and remaining throughout secretion (also in Video 9). Time mm:ss; relative to fusion (of bottom right LSV). Scale bars = 20 μm (left), 10 μm (top right), 1 μm (bottom right). **(B)** Correlative confocal and FIB-SEM showing MIM-Emerald fluorescence associated with the membrane of a putative fusion site. Top left: Confocal intensity-projection showing the fluorescence from a representative SG after resin embedding. Yellow squares mark the targeted region for FIB-SEM imaging. Bottom left: Overlay of the transformed fluorescence microscopy (FM) image onto a resliced XZ plane of the FIB-SEM stack. Correlation precision can be evaluated from the correspondence between the Glue-DsRed (green) and the LSVs in FIB-SEM. Marked area (yellow rectangle) shows MIM-Emerald appearing as a ring surrounding an LSV. Right: FIB-SEM XY view of the marked area in the XZ view, showing MIM localization to the putative fusion site on the plasma membrane (PM). Plane shown in the XZ view is marked by a dashed line. Scale bars = 20 μm (left), 1 μm (middle and right). UAS expression in A and B was driven by c135-GAL4.

with WT SGs, showing that one copy of the MIM gene is sufficient for exocytosis by membrane crumpling (Fig. 6 B). Next, we crossed the $MIM^{Def}$ line with a MIM null allele wherein exons 3–10 are deleted (Quinones et al., 2010; $MIM^{Null}$; Fig. 6 A), to generate a complete MIM null background ($MIM^{Null/Def}$). We observed a significant increase in kiss-and-run events (49 ± 3%; Fig. 6, B and C). The high frequency of kiss-and-run exocytosis in $MIM^{Null/Def}$ implies that in the complete absence of MIM, pores fail to expand. These results, taken together with the observation that $MIM^{KD}$ resulted in a higher frequency of full collapse exocytosis, suggest that MIM is required both for pore

expansion and pore stabilization in a dose-dependent manner. Thus, while pores fail to expand in the complete absence of MIM ($MIM^{Null/Def}$) function, low levels of MIM ($MIM^{KD}$) are sufficient to enable pore expansion, but these pores do not stabilize.

**The MIM I-BAR domain is essential but not sufficient for MIM localization and function**

Previous studies have shown that the I-BAR domain of MIM and other BAR-domain proteins are sufficient to induce membrane remodeling in cells on its own (Saarikangas et al., 2009; Nishimura et al., 2021; Tsai et al., 2022). We, therefore, aimed to explore the

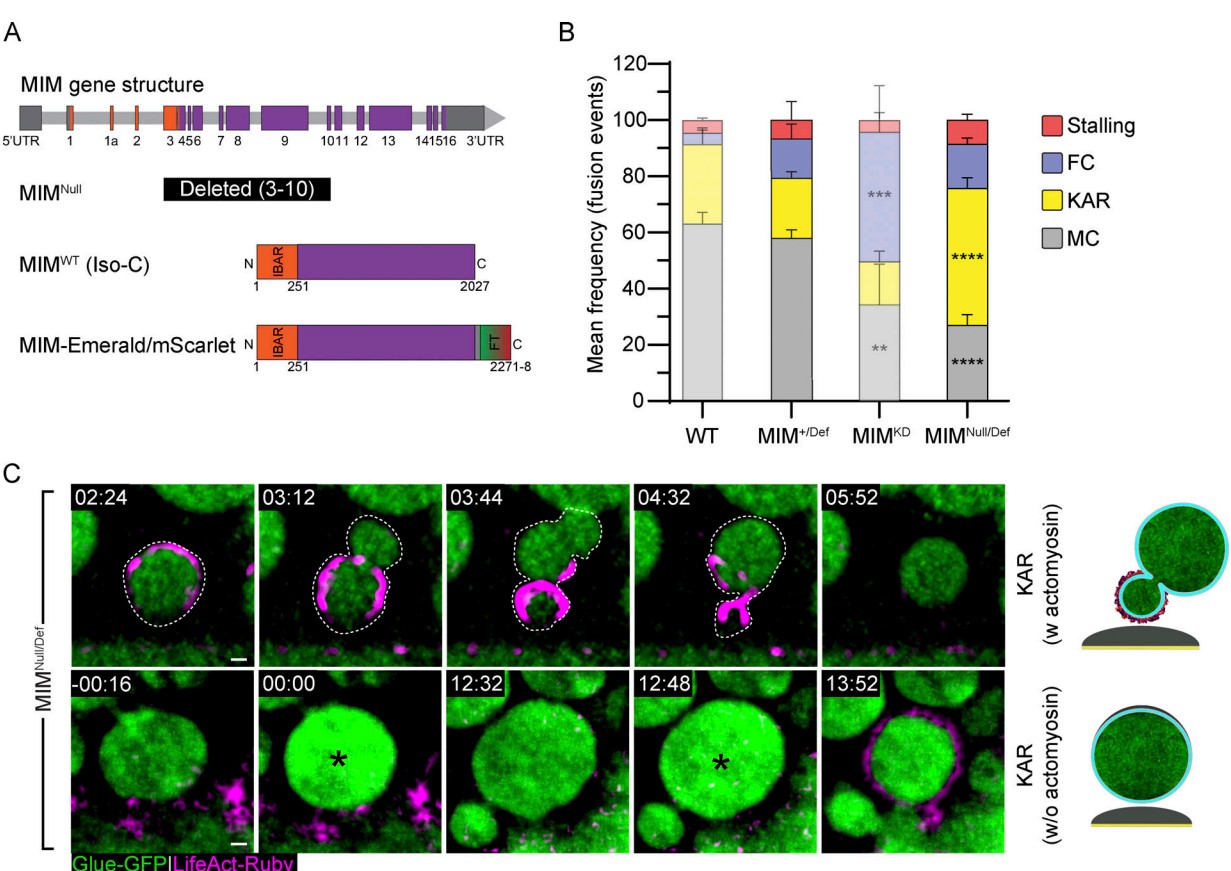

Figure 6. **MIM is essential for pore expansion and stabilization. (A)** Schematic of the MIM gene structure and constructs expressed in transgenic lines. Exons (rectangles), introns (light gray bars; not to scale), UTRs (dark gray), regions encoding for the I-BAR domain (orange). The null allele MIM$^{null}$ bears an internal deletion of exons 3–10. Isoform C (Iso-C) is the longest isoform and was used to generate MIM-Emerald/mScarlet; fluorescence tag (FT). **(B)** Mean frequency (%) of stalling, full collapse (FC), kiss-and-run (KAR), and membrane crumpling (MC) in MIM$^{+/Def}$, MIM$^{Null/Def}$, and MIM$^{KD}$ (from Fig. 4 B, semi-transparent; presented for convenience) compared with WT (from Fig. 2 C, semitransparent; presented for convenience). MIM$^{+/Def}$ is not significantly different from the WT. MIM$^{Null/Def}$ resulted in an increase of kiss-and-run event frequency (49 ± 3% (****)) as opposed to MIM$^{KD}$, which resulted in an increase of full collapse event frequency (46 ± 16% (***)). SG expressing Glue-GFP and LifeAct-Ruby. Error bars = SEM. $N$ (SGs) ≥ 3, $n$ (events) ≥ 200. Statistical significance presented is compared with MIM$^{+/Def}$ SGs. P values for MIM$^{Null/Def}$: ****(KAR) = 0.000031, ****(MC) = 0.00001. Two-tailed, unpaired multiple $t$ tests corrected using the Holm-Sidak method. **(C)** Time-lapse sequence (SRRF intensity-projection) of representative LSVs from MIM$^{Null/Def}$ undergoing kiss-and-run in SGs expressing the Glue-GFP (green) and LifeAct-Ruby (magenta) markers. Top: LSV (dashed line) undergoing kiss-and-run that recruited actin, deformed, and become displaced from the apical membrane (KAR w actomyosin). Bottom: Consecutive fusion events of the same LSV (at 00:00 and 12:48; KAR w/o actomyosin; asterisks) leading to actomyosin recruitment and membrane crumpling following the second fusion (13:52). Time mm:ss; relative to fusion. Scale bars = 1 µm. UAS expression in B and C driven by $fkh$-GAL4.

contribution of this domain to *Drosophila* MIM function and localization at the LSV fusion pore. To test whether the I-BAR domain is essential for the localization and function of MIM, we disrupted it by inserting an in-frame EGFP expression cassette into the I-BAR domain genomic sequence (MIM$^{ΔIBAR}$; Fig. 7 A). We observed that SGs exclusively expressing MIM$^{ΔIBAR}$ (MIM$^{ΔIBAR/Def}$) phenocopied MIM$^{Null/Def}$ and displayed a high frequency of kiss-and-run events (68 ± 7%), indicating that the I-BAR is essential for function (Fig. 7, B and C; and Fig. S5 A). In contrast, SGs carrying MIM$^{ΔIBAR}$ across from the wild-type MIM allele (MIM$^{ΔIBAR/+}$) phenocopied the MIM$^{KD}$ phenotype with a high frequency of full collapse events (50 ± 13%; Fig. 7, B and C). Given that a single copy of MIM is sufficient for function (Fig. 6 B), the high frequency of full collapse events observed for MIM$^{ΔIBAR/+}$ suggests that MIM$^{ΔIBAR}$ has a dominant negative (DN) effect over the wild-type MIM protein. Myosin II

is still recruited to the LSV membrane after full collapse in MIM$^{ΔIBAR/+}$ (Fig. 3 D, Fig. S5 C, and Video 10), underscoring the notion that pore stabilization by MIM is not essential for myosin recruitment and contractility. These results reinforce the conclusion that MIM activity is required in a dose-dependent manner for pore expansion and subsequently for pore stabilization and demonstrate that the I-BAR domain is essential for these functions. Importantly, overexpression of MIM-Emerald in both MIM$^{Null/Def}$ and MIM$^{ΔIBAR/Def}$ SGs rescued the loss-of-function phenotype (Fig. S5 B), strengthening the conclusion that MIM localizes to the fusion pore, as observed with MIM-Emerald, and that the observed changes in pore dynamics are a direct consequence of MIM loss-of-function.

To determine whether the I-BAR domain is sufficient for MIM function, we generated transgenic flies overexpressing the I-BAR domain in a WT background (I-BAR$^{OE}$; Fig. 7 A). We

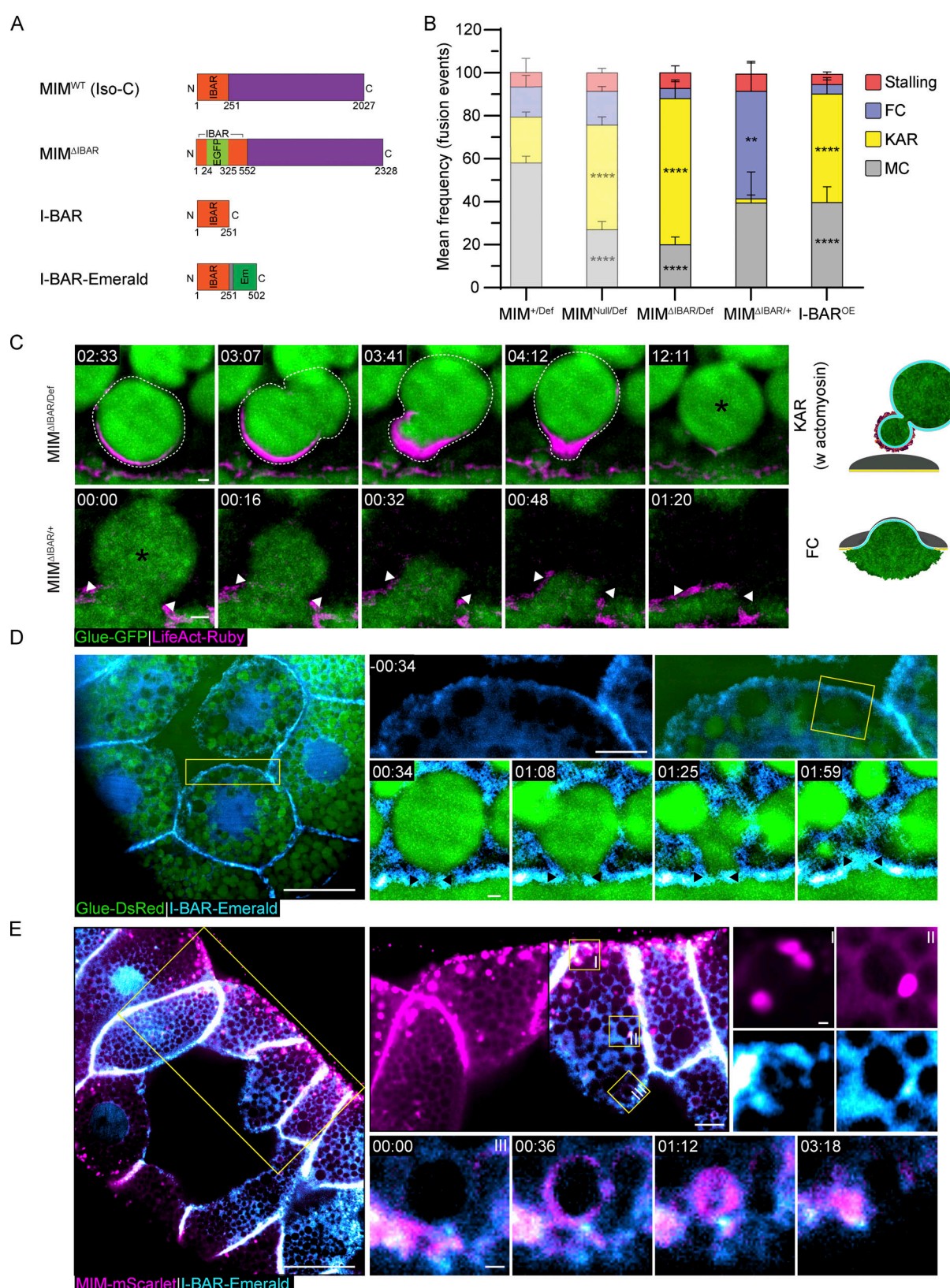

Figure 7. **The MIM I-BAR domain is essential but not sufficient for function and localization. (A)** Schematic of the constructs expressed in transgenic lines. MIM$^{\Delta IBAR}$—in-frame knock-in of an EGFP expression cassette into the I-BAR domain. MIM isoform C is illustrated, but all isoforms include the insertion. OE constructs of I-BAR and I-BAR-Emerald are under UAS control. We could not detect MIM$^{\Delta IBAR}$ EGFP signal in our imaging experiments, most likely because it is under our detection limit (over background). **(B)** Mean frequency (%) of stalling, full collapse (FC), kiss-and-run (KAR), and membrane crumpling (MC) in MIM$^{\Delta IBAR/Def}$, MIM$^{\Delta IBAR/+}$ and I-BAR$^{OE}$ compared to MIM$^{+/Def}$ and MIM$^{Null/Def}$ (from Fig. 6 B, semi-transparent; included for convenience). MIM$^{\Delta IBAR/Def}$

phenocopied MIM$^{Null/Def}$ with a high frequency of kiss-and-run events (68 ± 7% (****)). MIM$^{ΔIBAR/+}$ displayed a high frequency of full collapse events (50 ± 13% (**)), phenocopying MIM$^{KD}$. I-BAR$^{OE}$ displayed a high frequency of kiss-and-run events (50 ± 7% (****)). SGs express Glue-GFP and LifeAct-Ruby. Error bars = SEM. Statistical significance with respect to MIM$^{+/Def}$ for the MIM$^{ΔIBAR}$ expressing SGs and to WT SGs for the I-BAR$^{OE}$ SGs. N (SGs) ≥ 3, n (events) ≥ 150. P values for MIM$^{ΔIBAR/Def}$: ****(KAR) <0.000001, ****(MC) = 0.000012, for MIM$^{ΔIBAR/+}$: **(FC) = 0.003965, for I-BAR$^{OE}$: ****(KAR) = 0.000011, ****(MC) = 0.000015. Two-tailed, unpaired multiple t tests corrected using the Holm-Sidak method. **(C)** Time-lapse sequence (SRRF intensity-projection) of representative LSVs undergoing KAR w actomyosin and FC (complementary examples in Fig. S5 A) from MIM$^{ΔIBAR/Def}$ and MIM$^{ΔIBAR/+}$ SGs. Glue-GFP (green), LifeAct-Ruby (magenta). Fusion events (asterisks). Top: LSV (dashed line) that recruited actin, deformed, and moved away from the apical membrane, eventually fusing again at 11:12 (KAR w actomyosin). Bottom: Expanding pore (double arrowheads) leads to apparent vesicle integration into the apical membrane (FC). Scale bars = 1 μm. **(D)** Confocal intensity projection of a representative SG expressing Glue-DsRed (Costantino et al., 2008; green) and I-BAR-Emerald (cyan). Left: Overview of the SG showing the I-BAR-Emerald did not localize to cytoplasmic clusters. Top right: View of the apical surface showing homogenous I-BAR-Emerald localization on the apical membranes. Bottom right: Time-lapse sequence of a representative LSV (SRRF intensity-projection) during exocytosis, showing that I-BAR-Emerald did not localize specifically to the fusion pore (arrowheads). **(E)** Confocal intensity-projection of a representative SG ectopically expressing I-BAR-Emerald (cyan) and MIM-mScarlet (magenta). Left: Overview of the SG. Top right: Magnified views showing that the I-BAR-Emerald and MIM-mScarlet co-localized in cytoplasmic clusters and on the apical membrane. Bottom right: Time-lapse sequence of a representative LSV during exocytosis showing the IBAR-Emerald and MIM-mScarlet co-localizing at the fusion site and the fusion pore during secretion. UAS expression in MIM$^{+/Def}$, MIM$^{Null/Def}$, and MIM$^{ΔIBAR/Def}$ SGs driven by fkh-GAL4, in I-BAR$^{OE}$ (D and E) driven by c135-GAL4. In D and E, yellow squares mark the magnified area. Time mm:ss; relative to fusion in C–E. Scale bars = 20 μm (left), 10 μm (top right), 1 μm (bottom right).

observed that I-BAR$^{OE}$ SGs have a significantly elevated frequency of kiss-and-run events (50 ± 7%; Fig. 7 B and Fig. S5 A), indicating that overexpression of the I-BAR domain alone has a DN effect over the MIM$^{WT}$ protein and suggesting that the I-BAR domain is not sufficient for MIM function on its own.

To test whether the I-BAR domain is sufficient for MIM localization, we generated flies overexpressing a fluorescently tagged version, I-BAR-Emerald (Fig. 7 A). We observed that, unlike MIM-Emerald and MIM-mScarlet, I-BAR-Emerald did not localize to cytoplasmic clusters and did not accumulate in the vicinity of fusion sites. Instead, I-BAR-Emerald was uniformly distributed on the apical and lateral membranes and in the cytoplasm (Fig. 7 D), suggesting that the I-BAR domain by itself is not sufficient for MIM localization. To test for mutual effects on localization that may result from the formation of a complex between the I-BAR domain and full-length MIM, we coexpressed I-BAR-Emerald with MIM-mScarlet. The two proteins colocalized in cytoplasmic clusters, in the vicinity of fusion pores, and on secreting LSVs (Fig. 7 E). In addition, MIM-mScarlet localized uniformly on the apical membrane, mimicking the localization of the I-BAR domain (Fig. 7 E). These experiments demonstrate that the I-BAR domain can interact with the full-length MIM-mScarlet to produce both WT localization and the localization of the I-BAR domain alone. Collectively, these results suggest that the I-BAR domain is essential but not sufficient for MIM localization and function.

## Discussion

Secretory vesicle exocytosis is a fundamental biological process of great physiological importance. Secretory vesicle size differs between neuronal, endocrine, and exocrine cells and correlates well with their content and physiological function. Exocrine cells tend to generate large secretory vesicles (LSVs) that are at least an order of magnitude larger than conventional vesicles, sometimes reaching the size of a yeast cell. As vesicles increase in size, their surface area-to-volume ratio decreases. Hence, LSVs represent the most economical way to package large quantities of cargo. The larger size of LSVs may also facilitate the packaging of viscous macromolecular mixtures such as tears,

saliva, surfactants, and adhesives such as glue. However, increasing the size of secretory vesicles poses significant challenges to vesicle biogenesis, trafficking, fusion, content release, recycling, and maintenance of the limited apical cell surface.

We analyzed fusion pore dynamics of LSVs using the Drosophila larval salivary glands as a model for exocrine tissues. We observed that when LSVs fuse to the apical surface, the fusion pore expands rapidly, stabilizes with a wide diameter, and subsequently constricts to diameters that fall below the diffraction limit of light (Fig. 1 and Fig. S1). The ability to arrest the pore at distinct stages suggested that an intricate molecular machinery controls pore expansion, stabilization, and constriction during LSV exocytosis. Eliminating branched actin polymerization or MIM, or KD of CIP4, inhibits pore expansion, resulting in pore resealing and kiss-and-run-like exocytosis (Fig. 2, Fig. 4, Fig. 6, Fig. 7, Fig. S2, Fig. S3, and Fig. S5). On the other hand, KD of branched actin polymerization, MIM, SNX1, SNX6, and Amph inhibits pore stabilization, resulting in irreversible pore expansion and full collapse-like exocytosis (Fig. 2, Fig. 4, Fig. S2, and Fig. S4). Inhibiting myosin II prolongs pore expansion and slows or completely arrests pore constriction and membrane crumpling without affecting pore stabilization and results in stalled vesicles that fail to release their cargo (Fig. 3). Consistently, expanded fusion pores, often referred to as omega (Ω)-shaped profiles, also persist for several minutes in pancreatic acinar cells and alveolar type II cells (Nemoto et al., 2001; Haller et al., 2001). Similarly, the fusion pores that mediate secretion of Von Willebrand factor from Weibel-Palade bodies also limit vesicular membrane integration, allowing endocytosis to occur specifically from the vesicular membrane even after the pore reseals (Stevenson et al., 2017). Taken together, these observations suggest that maintaining membrane homeostasis via pore regulation is a conserved mechanism during LSV exocytosis.

We propose that membrane crumpling exocytosis is a distinct mode of vesicle secretion, which requires dedicated protein modules to expand, stabilize, and constrict the vesicle fusion pore. Fusion pore stabilization enables actomyosin recruitment to the LSV membrane, such that during actomyosin contraction, the vesicle membrane is folded without incorporating and

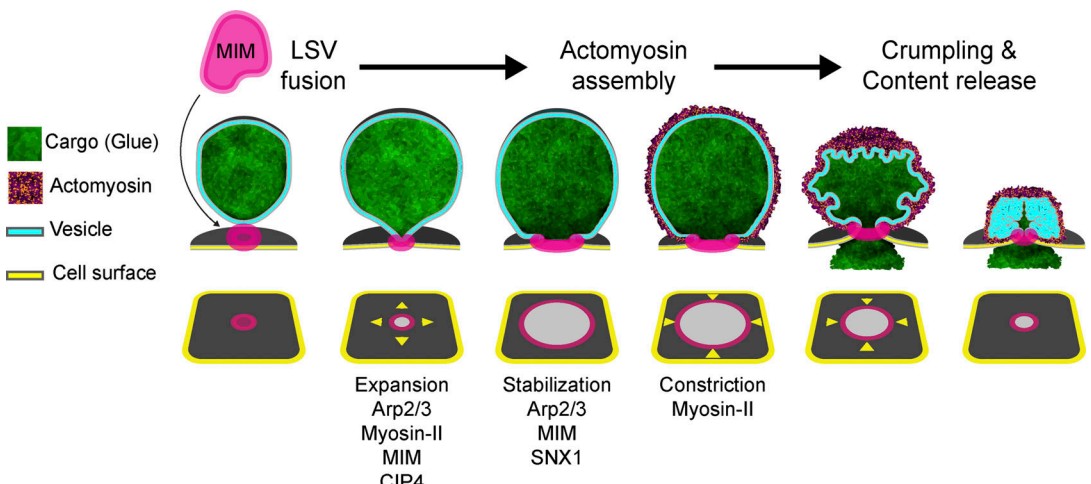

Figure 8. **BAR domain proteins, actin polymerization, and myosin II control fusion pore dynamics to facilitate crumpling exocytosis.** Schematic model of pore expansion, stabilization, and constriction as distinct steps in a sequence that facilitates exocytosis by actomyosin-mediated membrane crumpling (MC). The pore is regulated at each step by distinct components which include branched actin polymerization, myosin II and BAR domain proteins. The I-BAR protein MIM cooperates with actin and myosin II to control fusion pore dynamics of large secretory vesicles. MIM localizes to the future fusion site on the vesicle. After fusion, the pore expands in an Arp2/3, myosin II, MIM, and CIP4-dependent manner. The pore stabilizes with a wide diameter in a MIM-dependent manner, preventing full collapse and membrane integration. SNX1 is also essential for efficient pore stabilization. Pore constriction depends on myosin II and initiates during actomyosin-mediated membrane crumpling. Orchestrated dynamics of the fusion pore is essential for membrane crumpling and insulation of the apical cell membrane during exocytosis.

diluting the apical surface (Fig. 8). Thus, apical membrane homeostasis is maintained during secretion. When the fusion pore fails to stabilize, the actomyosin is still recruited, but only after the content is released by full collapse and at the expense of compromising membrane homeostasis (Fig. 8). These results show that exocytosis per se and myosin II recruitment can be uncoupled and that actomyosin is only essential for LSV exocytosis if the fusion pore is stabilized by a dedicated protein machinery (Kamalesh et al., 2021). Moreover, they show that fusion pore stabilization—and not actomyosin—physically sequesters the vesicular membrane during exocytosis.

Actomyosin recruitment and contractility still occur on the residual LSV membrane, even after its integration into the surface by full collapse (Fig. 3 and Fig. S5). This restricted actomyosin recruitment to the fused vesicle in the absence of a fusion pore indicates that the fusion pore itself does not chemically insulate the LSV membrane from the apical surface. The lack of mixing between the membranes may thus represent a distinct intrinsic set of properties between the vesicular and apical membranes. In this context, it would be interesting to test whether the vesicular and apical membranes do mix in SNX1$^{KD}$, SNX6$^{KD}$, Centaurin beta 1A$^{KD}$, and Syndapin$^{KD}$ settings that result in compound exocytosis, which might be attributed to membrane mixing and the acquisition of an apical cell membrane identity by LSVs after fusion.

The formation of dynamic fusion pores that expand, stabilize, and constrict requires transitions between membrane shapes and curvatures. Such transitions rely on molecules that can sense these changes and respond. Ca$^{2+}$, F-actin, and myosin II have been implicated in fusion pore expansion during exocytosis of LSVs (Haller et al., 2001; Larina et al., 2007; Doreian et al., 2008; Ñeco et al., 2008; Bhat and Thorn, 2009; reviewed in

Miklavc and Frick [2020]). Supporting these findings, our results demonstrate that branched actin polymerization, myosin II, and CIP4 are essential for LSV fusion pore expansion (Fig. 2, Fig. 3, Fig. 4, and Fig. S3).

In light of the CIP4$^{KD}$ phenotype and its localization (Fig. S4), we speculate that CIP4 might interact with the early fusion pore through its F-BAR domain (Shimada et al., 2007) and influence actin polymerization on the apical membrane through interactions with WASP, WAVE, Arp2/3 activators, or Dia, thereby facilitating pore expansion (Fricke et al., 2009; Yan et al., 2013). On the other hand, SNX-1$^{KD}$ resulted in a notable decrease in fusion pore stabilization efficiency, as evidenced by an increase in the frequency of full collapse events. Although we noted an apparent increase in kiss-and-run events in some of the RNAi lines, this was not fully penetrant and consistently observed and thus was not statistically significant in the quantification. These observations, coupled with the noted compound exocytosis in SNX-1$^{KD}$, may hint at a more global role of SNX-1 in LSV membrane remodeling (Fig. 4). Nevertheless, the emerging notion is that fusion pore expansion, stabilization, and constriction represent interlinked stages in a continuous process, rather than separate steps in a linear progression. This process appears to be mediated by BAR domain proteins, operating in a concentration-dependent manner at multiple junctures.

We further demonstrate a role for the actin cytoskeleton in fusion pore stabilization, as previously hypothesized (Haller et al., 2001). This function is distinct from the activity of actin in actomyosin-mediated vesicle constriction. Actomyosin recruitment and contractility are still observed under mild Arp2/3 inhibition (CK666$^{100 \; \mu M}$ and Arp3$^{KD}$), supporting the notion that actin fulfills non-overlapping roles at the fusion pore and on the vesicular membrane (Tran et al., 2015; Kamalesh et al., 2021;

Bhat and Thorn, 2009; Fig. 2 and Fig. S2). Moreover, pore expansion and stabilization occur prior to actomyosin contraction, which is only essential for extruding the cargo through a stabilized and constricting pore.

Actomyosin contractility on fused vesicles has been reported in many systems utilizing LSVs, where it might provide the force necessary to extrude the cargo through dynamic fusion pores (Sokac et al., 2003; Jerdeva et al., 2005; Yu and Bement, 2007b; Masedunskas et al., 2011; Miklavc et al., 2011, 2012, 2015; Nightingale et al., 2011, 2012; Tran et al., 2015; Kittelberger et al., 2016; Milberg et al., 2017; Kamalesh et al., 2021). Myosin II has also been implicated in pore expansion in pancreatic acinar cells and chromaffin cells utilizing large- and medium-sized vesicles respectively, which is consistent with the slower expansion kinetics we observed in Zipper$^{KD}$ and Y-27632$^{100 \ \mu M}$ treated glands (Doreian et al., 2008; Ñeco et al., 2008; Bhat and Thorn, 2009; Bretou et al., 2014). Additionally, we find that in *Drosophila* SGs, myosin II is essential for pore constriction in a dose-dependent manner (Fig. 3). We propose that myosin II regulates the fusion pore at two distinct steps by separate mechanisms: immediately after fusion, myosin II in concert with cortical actin reorganization favors conditions on the apical membrane that drive pore expansion while preventing the pore from resealing, which is in good agreement with previous studies. At later stages of LSV exocytosis, when the actomyosin contractile machinery compresses the vesicle to crumple the membrane and release the content, myosin II is essential for pore constriction, which might be attributed either to its contractile activity within the actomyosin meshwork coating the LSV, or to a dedicated machinery localized specifically around the pore. Collectively, our results support the emerging view that a combination of actin polymerization and myosin motor activity supply, at least in part, the forces needed to expand, stabilize, and constrict the LSV fusion pore, and independently fold and retain the vesicular membrane while extruding its content.

How are these and other forces directed to act at a specific time, place, and direction to control the fusion pore diameter? BAR-domain proteins have direct membrane interacting, shaping, and curvature sensing capabilities, in addition to indirect membrane remodeling activities through interactions with the cytoskeleton (reviewed in Kozlov et al. [2014] and Simunovic et al. [2019]). Indeed, we found that BAR-domain proteins are required for pore expansion and stabilization (Fig. 4 and Fig. S3). Specifically, we show that MIM is a key regulator of LSV fusion pore dynamics. MIM localizes to the fusion site prior to fusion, most likely in association with the vesicular membrane. Following fusion, MIM remains associated with the fusion pore throughout secretion (Fig. 5 and Fig. S4). One limitation of our study is that we could not directly verify the localization of endogenous MIM, due to the lack of suitable reagents. Nevertheless, the ability of MIM-Emerald to compensate for the loss-of-function phenotype, suggests that the tagged MIM protein is localized correctly. We further show that MIM is important for pore expansion and stabilization in a dose-dependent manner (Fig. 6, Fig. 7, and Fig. S5), and that the I-BAR domain of the protein is essential but not sufficient for its localization and function (Fig. 7 and Fig. S5). Hence, we conclude that MIM

fulfills opposing functions by initially promoting pore expansion and subsequently limiting pore dilation.

MIM has both membrane curvature sensing and shaping activities in vitro and in vivo (Saarikangas et al., 2015; Chaudhary et al., 2015; Kawabata Galbraith et al., 2018; Quinones et al., 2010). The MIM I-BAR domain not only favors negative membrane curvature, but can also induce membrane protrusions by directly bending the membrane (Mattila et al., 2007). MIM is also able to interact with positive membrane curvature, utilizing amphipathic α helices within the I-BAR domain that are inserted into the membrane (Drin et al., 2007; Bhatia et al., 2009). In addition, MIM harbors a proline-rich domain and a C-terminal WH2 domain that facilitates interaction with Cortactin (through its SH3 domain) and with the actin cytoskeleton, providing an extensive framework for dynamic membrane remodeling (Quinones et al., 2010; Lin et al., 2005; Mattila et al., 2003; Parker et al., 2023).

The formation of an initial nanometric pore represents an extreme change in membrane curvature. The I-BAR domain acting as a membrane curvature sensor may activate MIM to recruit branched actin polymerization through the WH2 domain (Quinones et al., 2010; Lee et al., 2007), resulting in pore expansion and recruitment of additional MIM dimers. As the pore expands, membrane curvature decreases and MIM concentration increases, promoting negative curvature on the pore that opposes the forces of actin polymerization, stabilizing the pore prior to pore constriction. Thus, our study suggests a dual function for MIM as a membrane curvature sensor and activator of local actin polymerization at low concentrations, and as a membrane-shaping protein at high concentrations, as previously proposed (Zhao et al., 2011), demonstrating the physiological relevance of such a mechanism. Consistently, MIM plays a similar role in the closure of toxin-induced transendothelial cell tunnels, by localizing to the rim of the hourglass-shaped structure and promoting Arp2/3 mediated actin polymerization (Maddugoda et al., 2011; Fedorov and Shemesh, 2017).

Strikingly, MIM also localizes in *Drosophila* SGs to cytoplasmic clusters/granules that display liquid-like behavior, which is a characteristic of protein liquid–liquid phase separation (Brangwynne et al., 2009; Fig. 5, Fig. S4, and Video 8). In silico analysis suggests that in addition to the I-BAR domain, MIM contains an extended low complexity domain with several putative phosphorylation clusters that are conserved between human and *Drosophila* MIM, indicating a high degree of regulation. In addition, clustering of IRSp53 and MIM was observed in vitro and in vivo before filopodial elongation (Disanza et al., 2013; Prévost et al., 2015; Saarikangas et al., 2015; Tsai et al., 2022). The role of the low-complexity region in generating the MIM cytoplasmic granules is underscored by the observation that the I-BAR alone mislocalizes to the apical and lateral membranes, but is targeted to MIM granules in the presence of the full-length protein (Fig. 5 and Fig. 7). An interplay between phase separation and assembly on membranes in a curvature-dependent manner, might be part of the mechanism allowing MIM and its interacting partners to preassemble the machinery that controls fusion pore dynamics.

Increasing pore size from the nanometric to the micrometric scale necessitates molecules that can assemble and control the expansion, stabilization, and constriction of larger fusion pores. Studies in smaller vesicles identified several factors affecting fusion pore dynamics, including $Ca^{2+}$ (Alés et al., 1999; Elhamdani et al., 2006), the SNARE protein complex (Archer et al., 2002; Wang et al., 2003; Wu et al., 2017; Fang et al., 2008; Vardjan et al., 2013; Neuland et al., 2014; Hastoy et al., 2017), dynamin (Graham et al., 2002; Tsuboi et al., 2004), and BAR domain proteins (Llobet et al., 2008; Somasundaram and Taraska, 2018). While some components might be shared across scales, others appear to be unique to LSVs. Our findings sketch the outlines of a dedicated regulatory machinery unique to LSV fusion pores, especially under circumstances where cargo viscosity or membrane homeostasis needs to be addressed. The unique fusion pore of LSVs defines a distinct mode of exocytosis that consists of fusion pore stabilization followed by actomyosin recruitment and contractility that extrudes the contents. This type of exocytosis assures that the vesicular and apical membranes remain distinct, such that apical membrane homeostasis is maintained, and the vesicular membrane can be retrieved by the slower process of endocytosis (Kamalesh et al., 2021). Modulating fusion pore dynamics results in aberrant kiss-and-run and full collapse types of secretion, highlighting the critical role of the underlying machinery in maintaining the integrity of the secretion process and the health of the secretory tissue.

## Materials and methods

### *Drosophila* strains and rearing conditions
*Drosophila* fly lines obtained from the Bloomington *Drosophila* Stock Center (NIH P40OD018537), the Vienna *Drosophila* Resource Center (VDRC, https://shop.vbc.ac.at/vdrc_store/) or generated (see below) by this study are summarized in Table S1. UAS-CIP4-EGFP (Fig. S4 B; Fricke et al., 2009) was received as a kind gift from Sven Bogdan (Philipps-University Marburg, Marburg, Germany), MIM[null] (Fig. 6, Fig. 7, and Fig. S5; Quinones et al., 2010) was received as a kind gift from Helen Zenner (University of Cambridge, Cambridge, UK), Sgs3-DsRed (Fig. 5 and Fig. 7; Costantino et al., 2008) was received as a kind gift from Julie Brill (University of Toronto, Toronto, Canada).

All fly stocks were reared using standard cornmeal, molasses, and yeast media at 21°C in a temperature-controlled room. Crosses and flies used from imaging experiments were grown in 25°C incubators without internal illumination. Parent flies producing progeny used for experiments were kept at low density (20–25 flies per bottle) in food bottles and transferred to a new bottle with fresh food every 3–4 d. Imaging experiments were performed on ex vivo cultures of third-instar *Drosophila* larval SGs. Larvae from crosses were used without distinguishing between sexes as no obvious sex-specific differences in secretion of SG were observed.

### Generation of transgenic flies
To generate UAS-MIM-Emerald—MIM-Emerald cDNA was synthesized (Genscript). The sequence of MIM Isoform C (longest transcript) was used, with the Emerald (Cubitt et al., 1999)

sequence flanked by an upstream 15 amino acids (aa) linker and 3 stop codons at the 3′ end. The synthesized cDNA was cloned into pUASt-attB (DGRC_1419) and injected into attP40 (for second chromosome insertion) and attP2 (for third chromosome insertion) flies. UAS-MIM-mScarlet was generated by replacing Emerald in UAS-MIM-Emerald with mScarlet (Bindels et al., 2017) to generate pUASt-attB-MIM-mScarlet which was injected into attP40 flies (second chromosome insertion).

The I-BAR domain was generated by PCR on the synthesized MIM isoform C cDNA using forward primer 5′-ACGTAGATCTAGTGATCTAAGTCTGGAACGCGATAGC-3′ and reverse primer 5′-ACGTGGTACCGCTGGCCTTAGCGTCATGG-3′ and cloned into pUASt-attB, injected to attP2 flies (third chromosome insertion). MIM was replaced with I-BAR in pUASt-MIM-Emerald to generate pUASt-attB-I-BAR-Emerald, which was injected into attP40 flies (second chromosome insertion).

To generate MIM[ΔIBAR], we used the MiMIC system (Venken et al., 2011). Plasmid 1298 (DGRC_1298: pBS-KS-attB1-2-PT-SA-SD-0-EGFP-FlAsH-StrepII-TEV-3xFlag) was injected into MI06553 (BDSC_41450), a MiMIC inserted site between exons 1 and 1a of MIM (location 2R:6947294 [+]), resulting in the EGFP expression cassette translated in all MIM isoforms inside the I-BAR domain (Fig. 6 A and Fig. 7 A). We were unable to detect the MIM[ΔIBAR] EGFP signal over the background, most probably because it was below our detection limit. All embryo microinjections were performed by BestGene Inc.

### Culturing third instar SGs for live imaging
SG culturing was performed as previously described (Rousso et al., 2016). In brief, SGs from third instar larvae were dissected out in Schneider's medium and transferred to a 35-mm dish with a 10 mm #1.5 glass bottom well (Cellvis D35-14-1.4-N), containing 100 µl of fresh medium for live imaging. SGs that were naturally secreting were identified by their expanded lumen, visible under a stereomicroscope before imaging. The SGs are visible in the live larvae such that larvae that were still not in the secreting phase could be returned to the growing bottle and used at a later time. Ecdysone treatment to induce secretion was not used in this study.

### Confocal image acquisition
Imaging of *Drosophila* SGs in Schneider's medium was performed using a Yokogawa automatic Spinning Disk confocal scanning unit (CSU-W1-T2) mounted on an inverted Olympus IX83 microscope. 60× 1.4 NA and 100×1.49 NA oil immersion objectives were used for data acquisition. Images were captured by dual back-illuminated Prime 95B sCMOS cameras (Photometrics) controlled by VisiView software (Visitron Systems GmbH). Confocal fluorescent excitation was done with solid-state laser diodes (488 nm by Toptica for GFP, EGFP, and Emerald; and 561 nm by Obis for Ruby, mCherry, DsRed, and mScarlet). The following fluorescence emission filters were used: 525/50 nm for GFP, EGFP, and Emerald; and 609/54 nm for Ruby, mCherry, DsRed, and mScarlet. When dual cameras were used, a 561-long pass D2 dichroic mirror was included in the light path before the cameras. Imaging was performed at room temperature. Image acquisition was performed using a custom

imaging script in Ironpython3 (available upon request). In brief, to produce a single image or a frame in a video, multiple images were rapidly acquired. We used Fiji (RRID: SCR_002285) to process these images in one of two ways: (i) the images were projected (intensity projection) to a single image (Schindelin et al., 2012), allowing to either improve signal-to-noise ratio using average projection or to enhance weak signals using a sum projection (otherwise not possible on this spinning disk confocal system); or (ii) the images are processed to super-resolution via the NanoJ-SRRF plugin (Gustafsson et al., 2016; see below). Live imaging data used for SRRF contained at least 10 exposure images per single image. To capture the pore (in XZ), no more than 0.5 µm was used for slice interval. To capture secretion and pore dynamics, the time interval (of final data) was no more than 18 s per frame of the time lapse.

### Measurement of vesicle swelling: Diameter and roundness
To measure changes in vesicle diameter and roundness upon fusion as shown in Fig. 1 C, we used the polygon tool of Fiji to outline the vesicle before and after the swelling is completed. In most LSVs, the swelling occurs in one frame (15–18 s), occasionally swelling can be slower and might take up to three frames (45–56 s). We then used the roundness measurement of Fiji,utilizing the formula, $4*(area/\pi*major\text{-}axis^2)$, which gives a value between 0 and 1, where 1 is a perfect round object and 0 is not round. The ratio represents the relationship between an object's area and the length of the major axis of the shape. Thus, a perfect circle, a square, or a polygon with equal faces is considered "round," but an ellipse or a non-equal faced polygon or rectangle would be "less round." A line would have 0 roundness since all its area is spread across its major axis. To measure diameter, we used the Fit circle function in Fiji, which creates a circle with the same area and centroid of the marked polygon (the outlines of the vesicle), and we extracted the diameter of that circle.

### Super-resolution image processing
Imaging data captured using one of the custom imaging scripts is made up of multiple image groups, each intended to make up a single image in the processed data. The data is processed using a semi-automated custom Fiji script (available upon request). The script accepts the appropriate imaging data folder (or multiple data folders for batch processing) and asks the user to input imaging parameters such as the magnification used, the number of slices and time points, or the number of channels that were used. The script then prepares a RAW map of the data—projecting the imaging data using the intensity projection Fiji function of average, maximum, or sum intensity projection. Users can either use this projected data as is without super-resolution or continue and select the time, slice range, and area that they wish to be processed into SRRF. The data is then processed into super-resolution using the NanoJ-SRRF Fiji plugin calling the plugin individually for each of the final images that will be created. The arguments used in the SRRF plugin are also controlled by user definitions collected by the script. The end result is a ready SRRF, 3D video, single plane video, or a Z stack in the area, time, and slices as chosen by the user.

Additional image processing was performed using Fiji for cropping and adjustment of brightness/contrast for visualization purposes only.

### Vesicle 3D segmentation
To demonstrate vesicle expansion upon fusion (Fig. S1 A), we used the surface function in Imaris software (RRID: SCR_007370). The outlines of the glue signal were marked manually in each plane to create the final surface.

### Drug treatments
To inhibit ROCK and vesicular secretion, SGs were treated with Y-27632 (100 µM final concentration; Sigma-Aldrich) for 20 min with mild shaking at RT before imaging (Segal et al., 2018). To inhibit Arp2/3, SGs were treated with CK666 (Sigma-Aldrich), 100 µM (mild inhibition) or 500 µM (stalling) final concentration. Several treatment protocols have been used depending on the experiment. For quantification of the mode of exocytosis distribution (Fig. 2, A and C; Fig. 3 A, and Fig. S2 A), SGs were treated with CK666 (100 µM) for 30 min with mild shaking at RT before imaging. To induce vesicle stalling and to measure pores (Fig. 2, E and F; and Fig. 3, B and C), SGs were treated with CK666 (500 µM) immediately prior to imaging. For double treatment with both Y-27632 and CK666 (Fig. 3 B), SGs were first treated with Y-27632(100 µM) for 15 min with mild shaking at RT, then CK666 (500 µM) was added to the imaging medium and imaging started immediately after. For the MIM KD phenotype under CK666 treatment (Fig. 4 C), SGs were treated with CK666 (500 µM) for 20 min with mild shaking at RT, before imaging. For all the conditions, imaging was performed in the presence of the inhibitor/s.

### Quantification of pore diameter and kinetics
Pore measurements were carried out exclusively on SRRF data and in the XY plane only. For pores positioned along the XZ plane, the three planes surrounding the widest plane of the pore were chosen and projected using maximal or average intensity projection using Fiji (Schindelin et al., 2012). For pores positioned along the XY plane, more planes could be used for the intensity projection as long as the pore in the membrane is clearly visible. Measurements were performed by hand in Fiji by drawing the shortest line across the fusion pore in each frame of the video. Only pores with clear outlines were chosen for quantifications. To estimate the mean expansion time, we defined the expanding phase from fusion onset up to 90% of maximal pore diameter.

### Quantification of modes of exocytosis frequency distribution
Quantification was done on averaged, bleach-corrected (Histogram matching), non-SRRF, whole gland time-lapse data sets using the Cell counter plugin in Fiji (Schindelin et al., 2012). First, fusion events (events of LSV swelling) were identified and tagged. Each tagged fusion event was verified by observing the LSV across the imaging planes. Tagging was done exclusively in the middle slice of time-lapse stacks. In Glue-GFP and LifeAct-Ruby expressing SGs, this was performed by viewing the Glue-GFP channel only to minimize bias of fusion detection to a

specific mode of exocytosis. To allow sufficient imaging time of an LSV after a fusion event, so that the mode of exocytosis can be determined, the fusion events in the last 20 frames of the time lapse are not tagged. At least 40 fusion events are tagged in a certain SG. In SGs that are secreting more rapidly (SGs with the higher frequency of full collapse events for example), two to three cells (instead of the whole field of view) were selected at random and all the fusion events in these cells were tagged. Next, each fusion event was classified as "crumpling," "full collapse," "kiss-and-run w/o actomyosin," "kiss-and-run w actomyosin," or "stalling." Scenarios where the LSV does not lose volume after fusion until actin assembles (roughly within 30–60 s) and then content release proceeds normally were classified as crumpling events. Scenarios where content release occurs before actin was observed on the vesicle were classified as full-collapse events. Scenarios where there was fusion without any actin assembled on the LSV and where content release did not occur were classified as kiss-and-run w/o actomyosin. Scenarios where there was actin on the vesicle, but squeezing results in vesicle displacement are classified as kiss-and-run w actomyosin. Finally, scenarios where there was actin assembly followed by disassembly or where the LifeAct-Ruby intensified but without content release were classified as ll events were identified in 2D and verified by viewing the same object in other slices. Every quantification includes at least 120 fusion events from three or more different organisms. Growing conditions, sample preparation, and imaging conditions were maintained between imaging days. GraphPad Prism version 8.4.2 (GraphPad software) was used for statistical analysis. We used multiple two-tailed *t* tests for comparing each of the modes in the control to the same mode in the test groups.

**Sample preparation for FIB-SEM and correlative microscopy**
Dissected *Drosophila* SGs were cryo-fixed using high-pressure freezing (Leica EM-ICE; Leica Microsystems). SGs were placed in aluminum planchettes (Wolhwend; 0.3/0 mm and 0.15/0.15 mm, #1314 and #1315, respectively) filled with Schneider's *Drosophila* Medium supplemented with 10% BSA and 10% FBS to serve as cryoprotectant. Automatic freeze substitution and resin embedding were performed in an EM-AFS2 mounted with EM-FSP (Leica Microsystems). Cryo-fixed SGs were placed in 0.1% uranyl acetate in dry acetone at –90°C for 45 h before the temperature was increased gradually, 2°C/h until it reached –45°C, and remained there for an additional 40 h, followed by three washes with acetone. Embedding with Lowicryl HM20 (cat#14340; EMS) was performed using a gradual increase in resin concentration, 10%, 25%, 50%, and 75%, 12 h each. 10% and 25% infiltration were performed at –45°C, while 50% infiltration was performed after the temperature was increased to –35°C (0.8°C/h) and kept at –35°C for 75% infiltration step. The temperature was then increased to –25°C during the first 12 h of 100% infiltration (0.8°C/h) and kept at –25°C for two additional rounds of exchange every 12–15 h. The resin was then polymerized under UV for a total of 106 h, while the temperature was increased to 20°C over the course of 10 h (4.5°C/h). Blocks were left for curing covered in foil until they turned from pink to transparent.

The blocks containing the SGs were trimmed using a razor blade from all sides, leaving a reduced block surface around the tissue. The block surface was sectioned using a diamond knife (35° ultra; Diatome) in an ultramicrotome (EM-UC7; Leica Microsystems) until the lumen of the gland was exposed and was visible on toluidine-blue stained sections. To help with targeting a region of interest (ROI), a mesh-like pattern was made manually on the polished surface with a fine razor (cat#72000; EMS).

**In-block FM for correlative imaging**
The trimmed blocks were mounted in a drop of PBS in the center of a glass-bottom imaging plate (Cellvis; D35-10-1.5N) and fixed to position with plasticine. Blocks were imaged with an Olympus spinning disc confocal IX83 microscope (details under "Confocal image acquisition"). Iterative imaging and polishing were performed in cases where the regions of interest were more than ~8 µm deep from the polished surface.

**FIB-SEM sample preparation and imaging**
Prior to FIB-SEM tomography, each block was mounted on an SEM stub (cat#75220; EMS) with a double-sided conductive carbon tape (cat#77816; EMS). The block edges were covered with additional strips of carbon tape and two to four strips of copper tape (cat#77802; EMS) and coated with a layer of colloidal silver liquid (cat#151105; EMS). Right before FIB-SEM imaging, the samples were sputter-coated with an 8–10 nm layer of Ir using a high-vacuum compact coating unit (Safematic; CCU-010 HV).

Samples were mounted on a Crossbeam 550 FIB-SEM system (Carl Zeiss), and ROIs were located under 2 kV, and either 5, 10, or 15 kV voltages, at 350 pA or 1,000 pA before tilting the stage to 54° and adjustment of working distance to 5 mm. A 1 µm-thick layer of Pt was deposited on top of the ROI using the ion beam (30 kV, 0.3–1.5 nA) and a trench was made (30 kV, 3–30 nA) with different dose factors (5–10), depending on the specific sample. The cross-section was then polished (30 kV, 1.5–15 nA) with the same dose factor before SEM imaging parameters were adjusted. Serial surface imaging was performed (30 kV, 0.7–1.5 nA) with the same dose factor as before, and SEM micrographs were acquired using either 2 kV, 0.35 nA using a mixed signal from secondary electrons type 2 (SE2) and energy selective backscattered electrons detectors with variable mixing ratios, depending on the sample, or using 1.5 kV, 1 nA with energy selective backscattered electrons detection only. Scan speed and signal-to-noise increase were different depending on the specific block and volume acquired.

**FIB SEM and correlative image processing and analysis**
The collected data was processed using Fiji (Schindelin et al., 2012). Stack alignment was performed using the registration plugin "Linear stack alignment with SIFT," allowing translation only and without interpolation. Images were filtered with an unsharp mask or local contrast enhancement (CLAHE) followed by smoothing. Fusion pore of vesicles before crumpling and after crumpling were measured in the plane where the pore is the widest, but in the narrowest point of the neck. Pore diameter

measurements from FIB-SEM data (Fig. 1 E) include 21 pores from four SGs for precrumpled LSVs and 22 pores from three SGs for crumpled LSVs. Since the data are not evenly distributed, each pore was considered as an individual measurement.

For correlative microscopy, image stacks were binned in z between two to fourfold, depending on the size of the stack before loading to ICY (de Chaumont et al., 2012) and Amira v2021.3 (Thermo Fisher Scientific) alongside the FM images of the block. Both datasets were roughly aligned in Amira as volumes to allow better identification of fiducials. Pairs of fiducials were identified in Amira volumes and placed on the data using eC-clem version 2 (in beta testing mode; Paul-Gilloteaux et al., 2017). Affine transformation was applied on the FM stack based on 15–25 fiducial pairs, and a complete error prediction of each point in the dataset was computed in the same way as described in Scher et al. (2021) and Potier et al. (2021) *Preprint*. Last, to validate that the transformation did not drastically deform the FM information, the transformed FM stack was resliced to show xz planes to observe the shape of the objects compared with the original FM stack.

### Statistics and reproducibility

Animal growing conditions, sample preparation, treatments, and imaging conditions were kept similar between experiments. Modes of exocytosis characterization (frequencies of membrane crumpling, full collapse, kiss-and-run, and stalling events; Fig. 2 C, Fig. 3 A, Fig. 4, B and D, Fig. 6 B, Fig. 7 B, and Fig. S5 B) are based on confocal live ex vivo tissue imaging of at least three SGs in which >40 fusion events per gland were identified and followed. To characterize the changes in pore diameter, we used SRRF imaging to better visualize the pores of representative events. Where phenotypes were not apparent from the time-lapse sequences, at least $n$ (pores) = 6 were quantified (as for membrane crumpling and stalling in Fig. 1 D, Fig. 2 F, and Fig. 3 B). Pore diameter measurements from FIB-SEM data (Fig. 1 E) were pooled from four individual data sets, each from a different gland ($n$ [pores] = 21 in pre-crumpled LSVs and $n$ [pores] = 22 in crumpled LSVs). Qualitative phenotypic characterization of knock-down experiments (summarized in Fig. 4 A) was based on at least four independent data sets using at least two different RNAi constructs for each gene (complete list of RNAi lines used in Table S1) and was based on visual inspection and detection of full collapse, kiss-and-run, or compound exocytosis events at apparent higher frequencies than WT SGs. GraphPad Prism version 8.4.2 (GraphPad software) was used for statistical analysis and for generating all plots. P values: ns > 0.05, * ≤ 0.05, ** ≤ 0.01 *** ≤ 0.001, **** ≤ 0.0001. Further information on sample size, exact P values, and the statistical tests used are mentioned in the figure legend and in the text for each experiment.

### Online supplemental material

Fig. S1 shows a 3D rendering of an LSV before and after vesicle swelling, and a step-by-step alignment and normalization plots used to produce Fig. 1 D. Fig. S2 shows representative events of the alternative modes of exocytosis in WT SGs, and full collapse and kiss-and-run representative events in Arp3[KD] SGs. Fig. S3 shows the expression data (modENCODE) used to choose the candidate BAR-domain containing genes for the genetic screen and representative kiss-and-run, full collapse, and compound exocytosis events from the hits of the screen. Fig. S4 shows representative exocytic events from SGs that express SNX1-GFP or CIP4-EGFP and representative images from an SG expressing the MIM-mScarlet tagged protein. Fig. S5 shows representative kiss-and-run events from MIM[ΔIBAR/def] and I-BAR[OE] SGs, results demonstrating that MIM-Emerald rescues the loss-of-function phenotype observed in MIM[null/def] and MIM[ΔIBAR/def] SGs, and a representative full collapse event in MIM[ΔIBAR/+] SG. Video 1 shows representative membrane crumpling events as visualized by confocal and SRRF imaging. Video 2 shows representative events of full collapse and kiss-and-run from CK666[500 μM] SGs. Video 3 shows representative events of full collapse and kiss-and-run in WT SGs. Video 4 shows a representative SG undergoing CK666[500 μM] treatment, which induces LSV stalling, and a representative stalling event of an LSV with an arrested narrow pore. Video 5 shows stalling in Zipper[KD] (myosin II heavy chain) and Y-27632 (ROCK inhibitor) treaded glands compared with a membrane crumpling event in an untreated WT SG. Video 6 shows representative membrane crumpling, full collapse, and kiss-and-run events in WT SGs expressing the Sqh-mCherry (myosin II light chain) marker. Video 7 shows representative compound exocytosis events from SNX1[KD] and SNX6[KD] SGs. Video 8 shows a representative MIM-Emerald cluster merging in the cytoplasm. Video 9 shows a representative exocytic event where MIM emerald is localized to the vicinity of the fusion pore. Video 10 shows a representative full collapse event in MIM[ΔIBAR/+] SG expressing the Sqh-mCherry (myosin II light chain) marker. Table S1 lists all the fly lines used in the study, including accession numbers or relevant references.

### Data availability

The data are available from the corresponding author upon reasonable request.

## Acknowledgments

We thank Prof. Sven Bogdan (Philipps-University Marburg, Marburg, Germany) for the UAS-CIP4-EGFP *Drosophila* line, Dr. Helen Zenner (University of Cambridge, Cambridge, UK) for the MIM[Null] *Drosophila* line, and Prof. Julie Brill (University of Toronto, Toronto, Canada) for the Sgs3-DsRed *Drosophila* line. We thank the EM Unit of the Weizmann Institute for assistance. We thank all members of the O. Avinoam and B.-Z. Shilo labs for fruitful comments and discussions.

This research was supported by the Israel Science Foundation (grant no. 706/20) to B.-Z. Shilo, O. Avinoam, and E.D. Schejter, and the Minerva Foundation with funding from the Federal German Ministry for Education and Research. O. Avinoam also acknowledges funding from the Henry Chanoch Krenter Institute for Biomedical Imaging and Genomics, the Schwartz Reisman Collaborative Science Program, the Yeda-Sela Center for Basic Research, and the European Research Council (ERC) under the European Union's Horizon 2020 research and innovation program (grant agreement no 851080). O. Avinoam is an

incumbent of the Miriam Berman presidential development chair. B.-Z. Shilo is an incumbent of the Hilda and Cecil Lewis Professorial Chair in Molecular Genetics.

Author contributions: Conceptualization: O. Avinoam, B.-Z. Shilo, E.D. Schejter, and T. Biton; Data curation: T. Biton, N. Scher, and S. Carmon; Formal analysis: O. Avinoam, T. Biton, and N. Scher; Funding acquisition: O. Avinoam and B-Z. Shilo; Investigation: T. Biton, N. Scher, and S. Carmon; Methodology: O. Avinoam, B.-Z. Shilo, E.D. Schejter, and T. Biton; Project administration: O. Avinoam, B.-Z. Shilo, E.D. Schejter, and Y. Elbaz-Alon; Resources: O. Avinoam and B.-Z. Shilo; Software: T. Biton and N. Scher; Supervision: O. Avinoam, B.-Z. Shilo, E.D. Schejter, and Y. Elbaz-Alon; Validation: O. Avinoam, T. Biton, Y. Elbaz-Alon, and N. Scher; Visualization: O. Avinoam, B.-Z. Shilo, E.D. Schejter, T. Biton, and N. Scher; Writing—original draft: O. Avinoam, B.-Z. Shilo, and T. Biton; Writing—review and editing: O. Avinoam, B.-Z. Shilo, E.D. Schejter, Y. Elbaz-Alon, N. Scher, S. Carmon, and T. Biton.

Disclosures: All authors have completed and submitted the ICMJE Form for Disclosure of Potential Conflicts of Interest. O. Avinoam reported non-financial support from Profuse Technology outside the submitted work. No other disclosures were reported.

Submitted: 2 March 2023

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

# Supplemental material

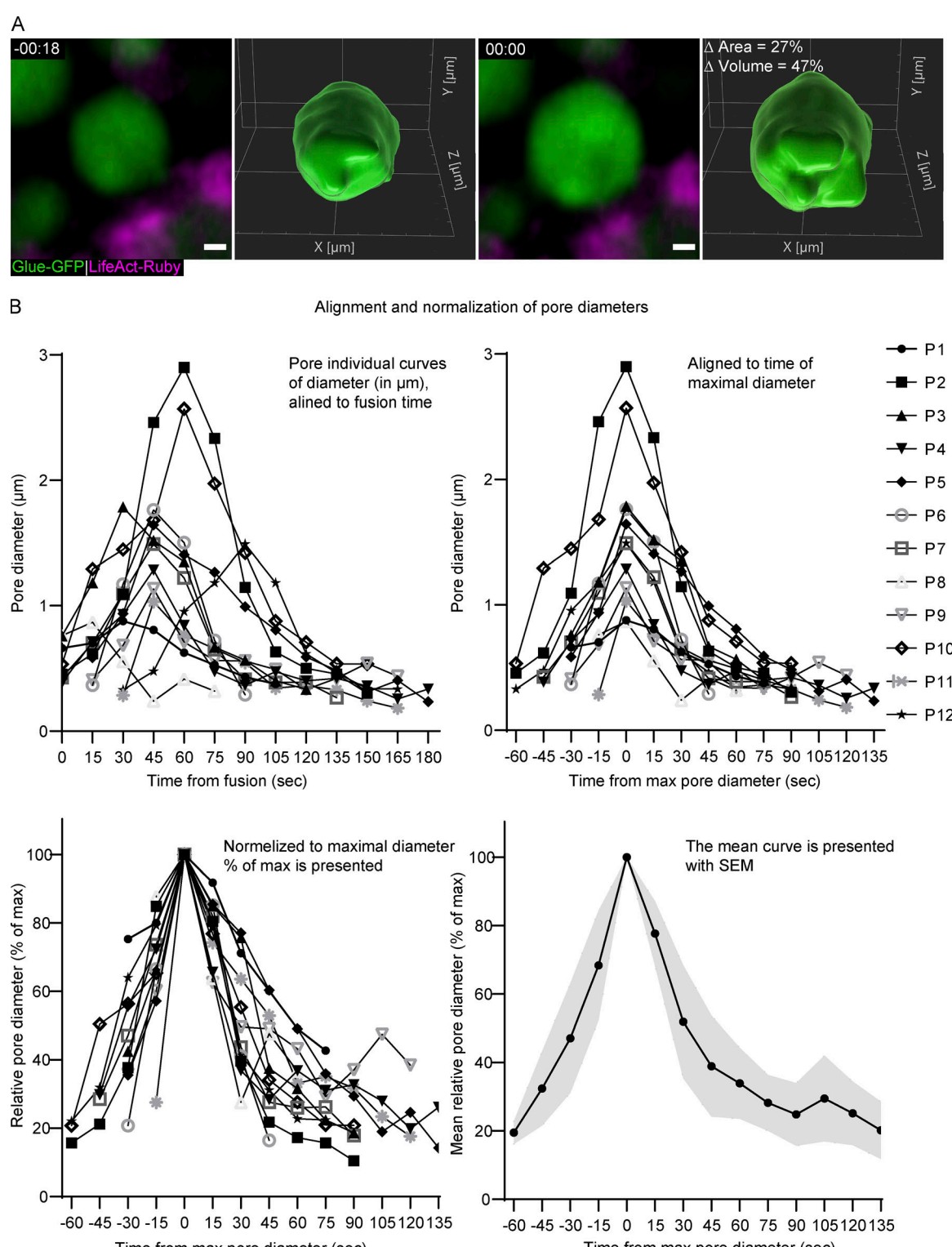

Figure S1. **LSV swelling, and pore diameter quantification and normalization. (A)** 3D segmentation of an LSV before and after fusion showing vesicle swelling. Representative images (confocal intensity-projection) and 3D segmentation of an LSV before (left) and after (right) fusion. The vesicle expands by 27% in 2D area and 47% in calculated 3D volume, implying that the observed increase in 2D is a result of LSV swelling and not LSV movement in or out of the imaging plane. Glue-GFP (green) and LifeAct-Ruby (magenta; UAS-based expression driven by c135-GAL4). Time mm:ss; relative to fusion. Scale bar is 1 μm. **(B)** The step-by-step alignment and normalization plots used to produce the pore dynamics plot in Fig. 1 D. Top left: Pore diameter measurements (μm) are plotted over time (seconds; from fusion). Top right: The curves are aligned over time (seconds; relative to maximal pore diameter). Bottom left: The curves are normalized (% of max pore diameter). Bottom right: The mean relative curve is presented with SEM (gray). *n* (pores) = 12.

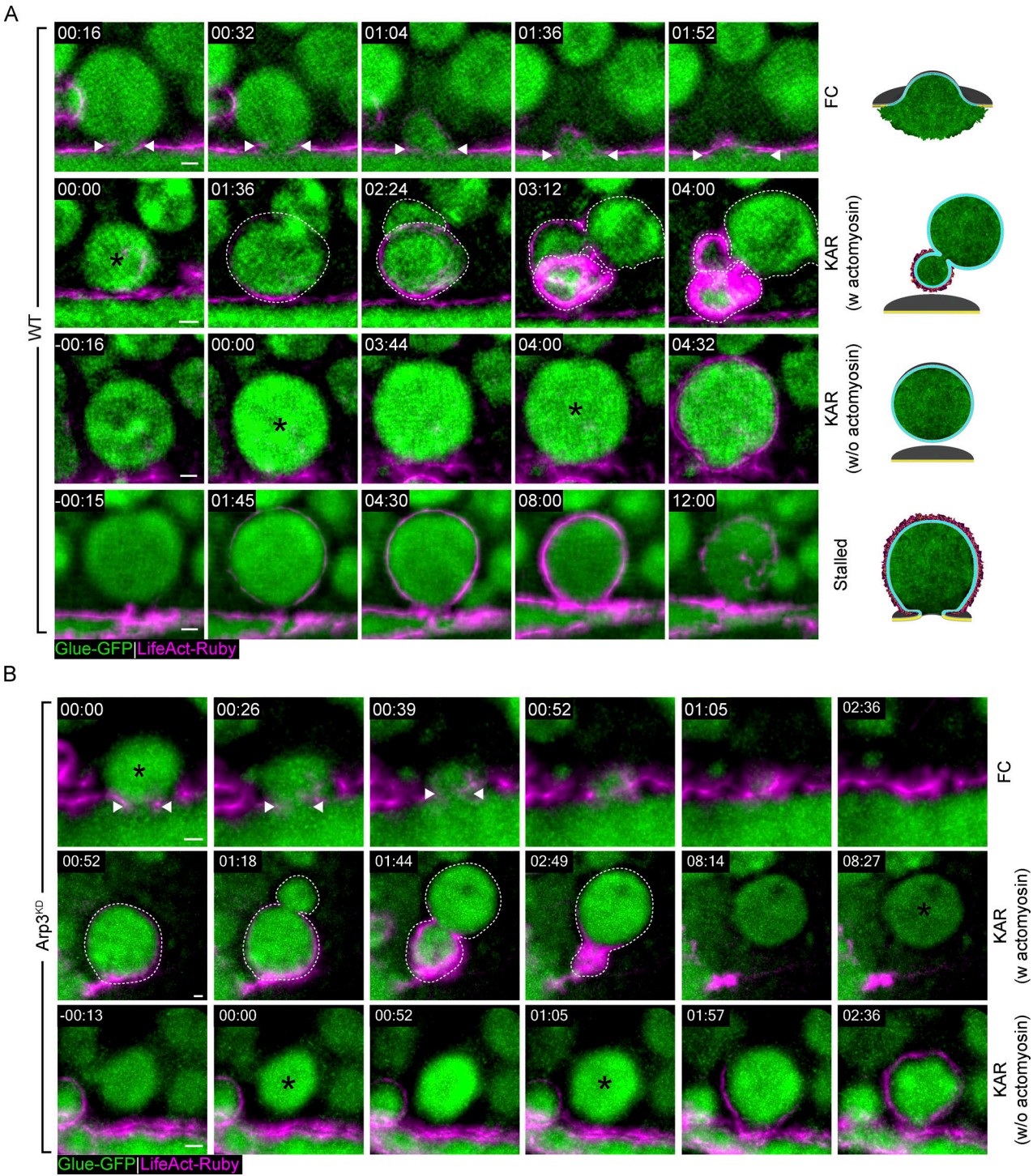

**Figure S2. Branched actin polymerization is essential for pore expansion and stabilization and modes of exocytosis in the SG. (A)** Time-lapse sequence (SRRF intensity-projection) of representative LSVs from WT SGs undergoing full collapse (FC; top; fusion pore - dual arrowheads), kiss-and-run (KAR w actomyosin—second row; KAR w/o actomyosin—third row; fusion—asterisk; "deforming LSV"—dashed line), and Stalling (bottom; vesicle recruits actin [01:45] but does not release its content; also in Video 3). **(B)** Time-lapse sequence of representative LSVs from Arp3KD SGs (SRRF intensity-projection; complementary to Fig. 2 A) undergoing FC (top; fusion pore—double arrowheads) and KAR (middle—KAR w actomyosin; "deforming LSV"—dashed line; LSV fuse again at 08:27—asterisk; bottom—KAR w/o actomyosin; consecutive fusion events of the same LSV at 00:00 and 01:05—asterisk). Glue-GFP (green) and LifeAct-Ruby (magenta). UAS-based expression in A driven by c135-GAL4 in B driven by *fkh*-GAL4. Time mm:ss; relative to fusion. Scale bars = 1 μm.

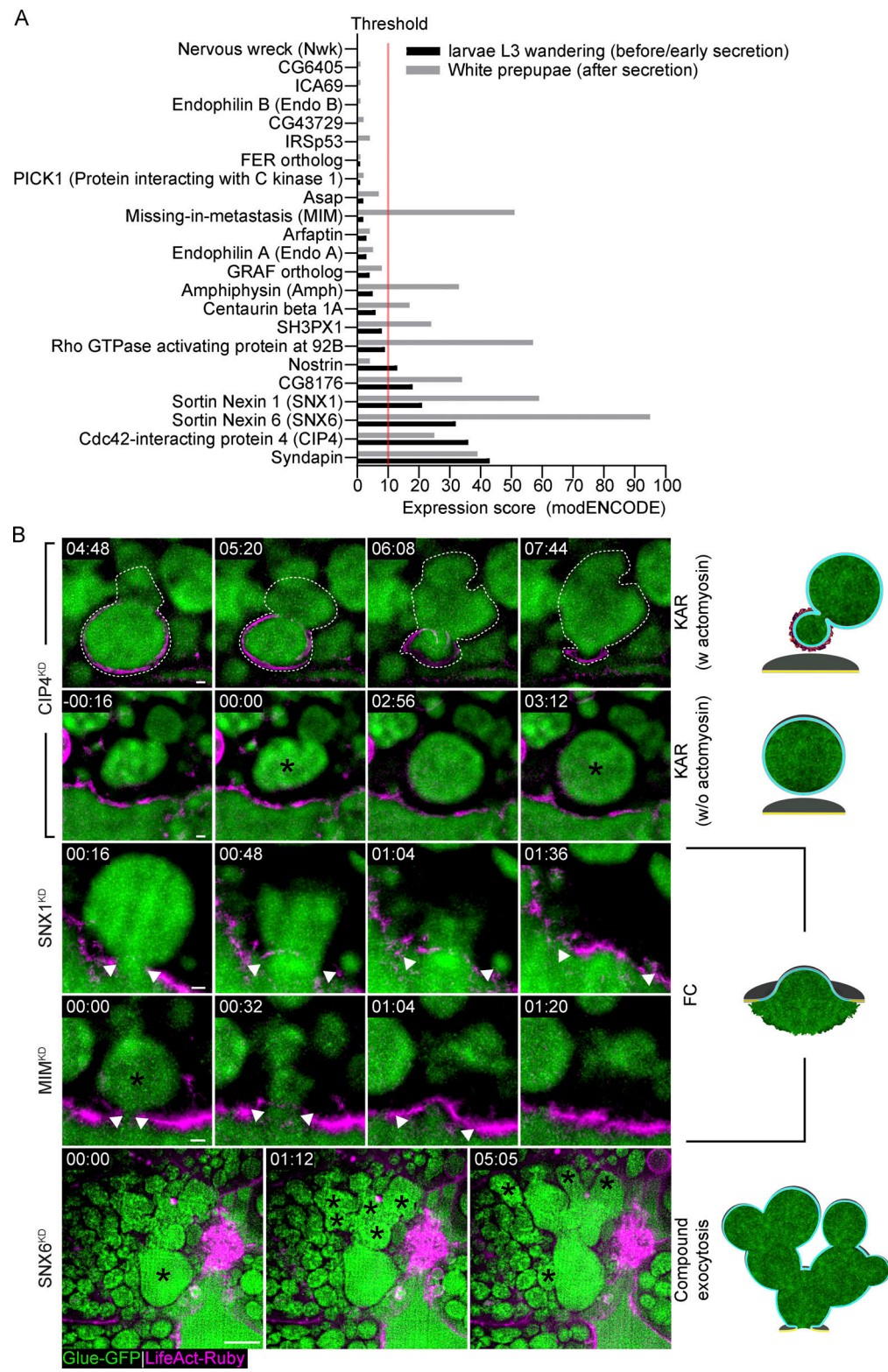

Figure S3. **Bar domain genetic screen and phenotypes. (A)** *Drosophila* BAR domain gene expression scores (modENCODE, FlyBase; Graveley et al., 2011; Gramates et al., 2022) in SGs before (black bars) and after the secretion phase (gray bars). The 11 genes expressed above threshold (red line), were included in our screen. **(B)** Time-lapse sequence of representative LSVs from BAR domain KD SGs (SRRF intensity-projection) undergoing kiss-and-run (KAR; CIP4KD; top two rows; "deforming" LSV—dashed line; consecutive fusion events at 00:00 and 03:12—asterisk), full collapse (FC; SNX1KD and MIMKD; third and fourth rows; expanding fusion pore—double arrowheads; fusion—asterisk) and compound exocytosis (SNX6KD; bottom; fusion—asterisks). In compound exocytosis, we see that after fusion, LSVs fuse consecutively to an LSV fused to the cell membrane, creating a large "blob." Glue-GFP (green) and LifeAct-Ruby (magenta). RNAi expressed under UAS control. UAS expression driven by *fkh*-Gal4. Time mm:ss; relative to fusion. Scale bars in KAR and FC 1 μm, in Compound exocytosis 10 μm.

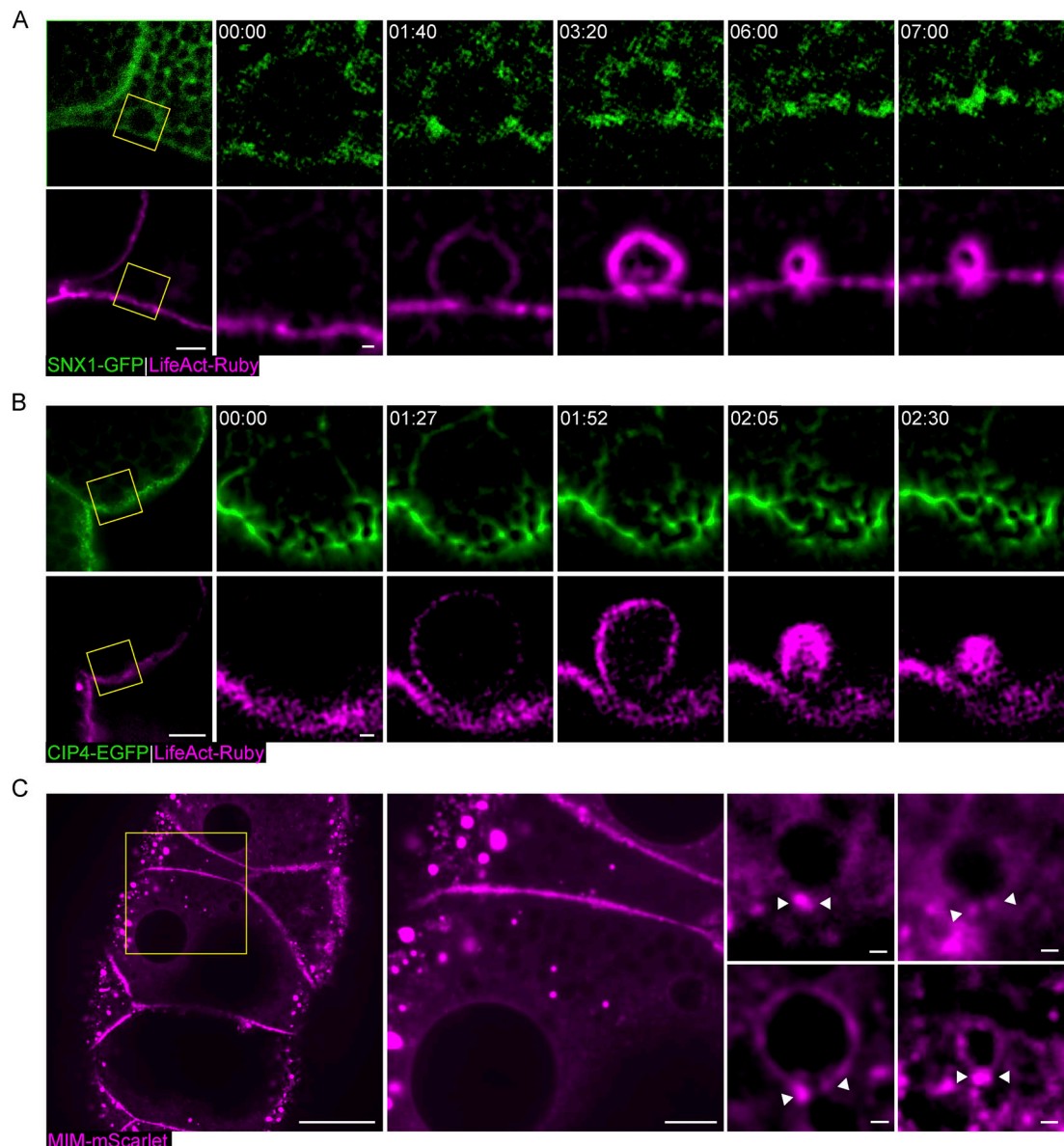

Figure S4. **SNX1-GFP, CIP4-EGFP, and MIM-mScarlet localization. (A)** Confocal intensity-projection and time-lapse sequence of a representative SG expressing SNX1-GFP (green) and LifeAcrt-Ruby (magenta) at increasing magnification. Yellow squares mark the magnified area. Left: Representative image of a cell (confocal intensity-projection); SNX1 is seen on cell membranes and in the cytoplasm. Right: Time-lapse sequence of representative secreting LSV (SRRF intensity-projection). Specific localization of SNX1-GFP to the fusion pore or to the LSV is not observed. **(B)** Confocal intensity-projection and time-lapse sequence of a representative SG expressing CIP4-EGFP (green) and LifeAct-Ruby (magenta), at increasing magnification. Yellow squares mark the magnified area. Left: Representative image of a cell (confocal intensity-projection); CIP4-EGFP is seen mostly on the apical and lateral membranes of the cell. Right: Time lapse of representative secreting LSV (SRRF intensity-projection). CIP4-EGFP is mostly apical and localizes to the LSV membrane after fusion, but specific localization to the pore was not detected. **(C)** Representative SG expressing MIM-mScarlet (Fig. 6 A; magenta). Left: Overview of a large area in the gland (confocal intensity-projection). The yellow square shows the area enlarged in the middle image. Middle: Enlarged cell overview. Like MIM-Emerald (Fig. 5 A), MIM-mScarlet localizes to cytoplasmic clusters and is absent from the apical membrane. Additionally, MIM-mScarlet localizes in the cytoplasm and to lateral membranes. Right: Representative LSVs from different vesicles, in various stages of secretion (confocal intensity-projections). MIM-mScarlet is observed at and around the pore (white arrowheads) and is localized to LSV membrane during secretion. UAS-based expression is driven by c135-GAL4. Scale bars in A (left) and B (left), 10 μm; in A (right) and B (right), 1 μm; in C (left), 40 μm; in C (middle), 10 μm; in C (right), 1 μm.

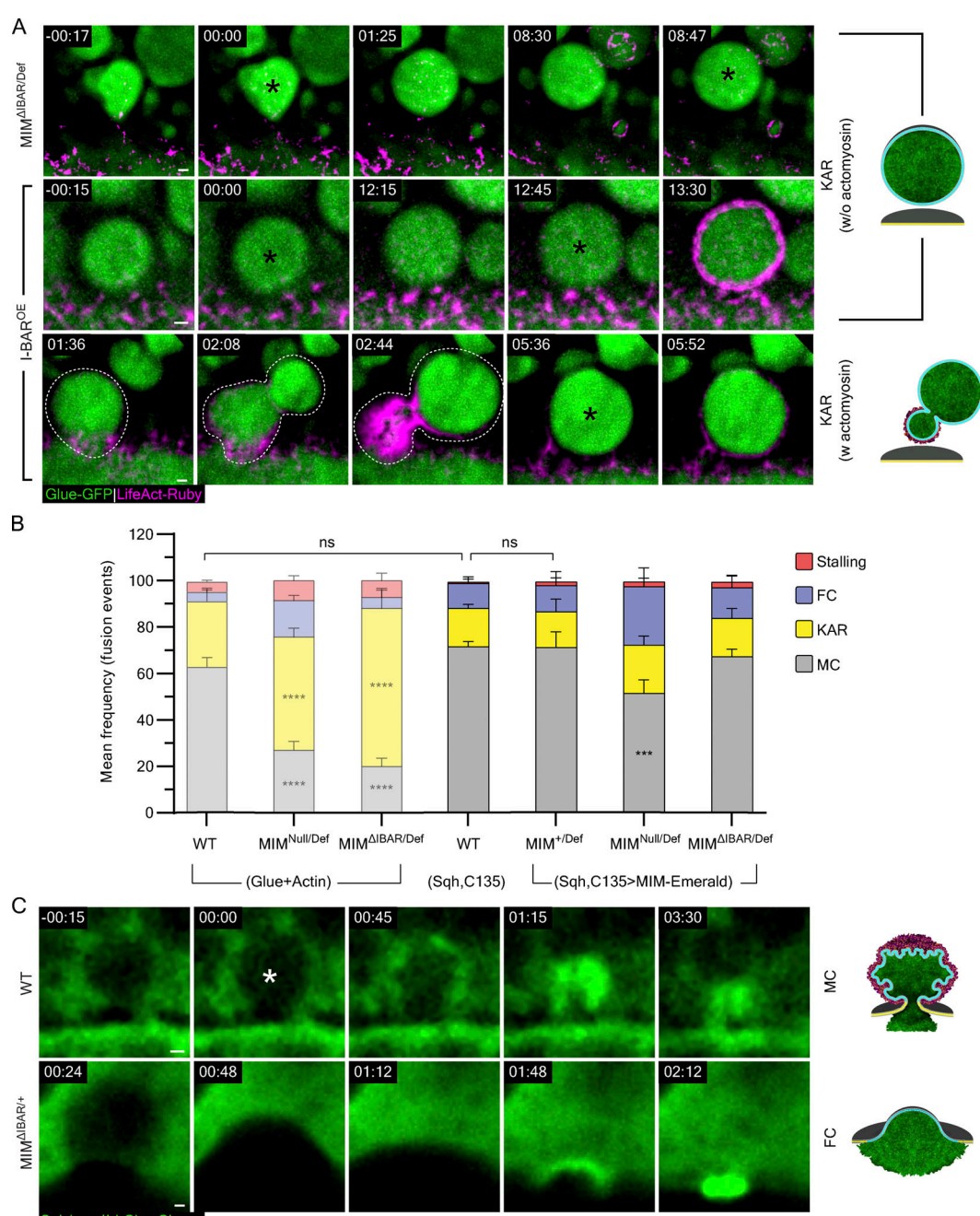

Figure S5. **Complementary examples to** Fig. 7 C; **MIM-Emerald rescue experiment and myosin localization in MIM$^{\Delta IBAR/+}$. (A)** Time-lapse sequence of representative LSVs from MIM$^{\Delta IBAR/Def}$ and I-BAR$^{OE}$ SGs undergoing kiss-and-run (KAR) w/o and w actomyosin (SRRF intensity-projection). Complementary to Fig. 6 C. Top and middle: MIM$^{\Delta IBAR/Def}$ and I-BAR$^{OE}$; KAR w/o actomyosin. Consecutive fusion events on the same LSV at 00:00 and 08:47 and at 00:00 and 12:45 (asterisks). Bottom: I-BAR$^{OE}$; KAR w actomyosin. "Deforming LSV" (dashed line) eventually fused again at 05:36 (asterisk). Glue-GFP (green) and LifeAct-Ruby (magenta). **(B)** Mean frequency (%) of stalling, full collapse (FC), kiss-and-run (KAR), and membrane crumpling (MC) in WT, MIM$^{Null/Def}$, and MIM$^{\Delta IBAR/Def}$ using the Glue-GFP and LifeAct-Ruby markers and in MIM$^{+/Def}$, MIM$^{Null/Def}$, and MIM$^{\Delta IBAR/Def}$ expressing MIM-Emerald compared with WT glands using the Sqh-mCherry (myosin II light chain) marker from Fig. 6 A and Fig. 7 A (semi-transparent bars). The majority of fusion events observed in WT SGs with Glue-GFP and LifeAct-Ruby markers result in membrane crumpling events (67 ± 3%), MIM$^{Null/Def}$ and MIM$^{\Delta IBAR/Def}$ SGs present the MIM loss-of-function phenotype and a high frequency of kiss-and-run events (49 ± 3% (****) and 68 ± 7% (****), respectively). WT SGs using the Sqh-mCherry marker do not significantly differ from the WT SGs using the Glue-GFP and LifeAct-Ruby markers. In MIM$^{+/Def}$ SGs that also expressed MIM-Emerald, the mode of exocytosis distribution did not vary significantly from the WT SGs using the Sqh-mCherry assay. When MIM-Emerald is expressed in MIM$^{Null/Def}$ or MIM$^{\Delta IBAR/Def}$ SGs, the loss-of-function phenotype is rescued, and membrane crumpling exocytosis is the major exocytosis mode observed (52 ± 5% (***) and 68 ± 3% (ns), respectively). N (SGs) ≥ 3, n (events) ≥ 150. P values for MIM$^{Null/Def}$: ***(MC) = 0.009667. Two-tailed unpaired multiple t tests corrected using the Holm-Sidak method. **(C)** Time-lapse sequence of representative LSVs from WT SGs undergoing membrane crumpling and MIM$^{\Delta IBAR/+}$ SGs undergoing full collapse (also in Video 10; confocal intensity-projection). Fusion (white asterisks). Sqh-mCherry (green). Time in A and C, mm:ss; relative to fusion. UAS-based expression in I-BAR$^{OE}$ (A) driven by c135-GAL4, in MIM$^{\Delta IABR/Def}$ (A) driven by *fkh*-GAL4, in B for the Glue-GFP and LifeAct-Ruby expressing SGs driven by *fkh*-GAL4 and for Sqh-mCherry expressing SGs driven by c135-GAL4. Scale bars in A and C, 1 μm.

**Video 1.** **Confocal and SRRF imaging of exocytic events in the *Drosophila* SG.** Time-lapse video of representative exocytic membrane crumpling events in WT SGs (confocal, top; SRRF, bottom; corresponds to Fig. 1 B). Fusion onset is detected by LSV swelling (Fig. S1 A). LifeAct-Ruby (actomyosin) recruitment is first visible ~50 s after fusion and followed by content release. Fusion pore diameter was defined and measured at the narrowest aperture observed at the mid-plane through the vesicle, which connects the vesicle and the lumen. The fusion pore expands before actomyosin recruitment and constricts during content release. Time mm:ss, relative to fusion seen in the second frames. Video frames were taken every 16 s for the confocal imaging and every 10 s for the SRRF imaging. LifeAct-Ruby (magenta; left column; expression driven by c135-GAL4) and Glue-GFP (green; middle column; endogenous promoter). The merged image of both imaging channels is presented in the right column. Scale bars = 1 μm.

**Video 2.** **Branched actin polymerization is essential for pore expansion and stabilization.** Time-lapse video of representative LSVs from CK666[100 μM] SGs undergoing full collapse (FC) and kiss-and-run (KAR; SRRF intensity-projection; corresponds to Fig 2 A). Top: FC appears as a content release that precedes actomyosin assembly. The fusion pore expands, and the vesicle appears to integrate into the apical surface. Middle: KAR w actomyosin, appearing as deformation and displacement of the LSV from the apical membrane upon actomyosin contraction. The same LSV fuse again at 11:28, suggesting it detached between fusion events. Bottom: KAR w/o actomyosin, appearing as consecutive fusion events without content release. Actomyosin was recruited after the second fusion event (01:52), leading to membrane crumpling. Time mm:ss; relative to fusion seen in the second frames. Video frames were taken every 16 s. LifeAct-Ruby (magenta; left column; expression driven by c135-GAL4) and Glue-GFP (green; middle column; endogenous promoter). The merged image of both imaging channels is presented in the right column. Scale bars = 1 μm.

**Video 3.** **Modes of exocytosis in the SG.** Time-lapse video (SRRF intensity-projection) of representative LSVs from WT SGs (corresponds to Fig. S2 A) undergoing full collapse (FC—top), kiss-and-run (KAR w actomyosin—second row; KAR w/o actomyosin—third row), and stalling (bottom; vesicle recruits actin [01:45] but does not release its content). Time mm:ss; relative to fusion seen in the second frames. Video frames were taken every sixteen (for FC and KAR) or 15 s (for stalling). LifeAct-Ruby (magenta; left column; expression driven by c135-GAL4) and Glue-GFP (green; middle column; endogenous promoter). The merged image of both imaging channels is presented on the right column. Scale bars = 1 μm.

**Video 4.** **Acute inhibition of branched actin polymerization results in an accumulation of stalled, actomyosin coated, LSVs with narrow pores.** Left: Time-lapse video of a representative SG treated with an acute dose Arp2/3 inhibitor CK666 (confocal Z-projection; corresponds to Fig. 2 D). Upon treatment with CK666[500 μM] (time 00:00), secretion slows down quickly while stalled, actin-coated LSVs accumulate at the cell apical membrane. Less fusion events are detected as time progresses. Right: Time-lapse video of a representative LSV (SRRF intensity-projection) stalled with a narrow pore that arrests before completing expansion under CK666[500 μM] treatment (corresponds to Fig. 2 E). Time mm:ss; relative to CK666 treatment (left; first frame) and to fusion (right; second frame). Video frames were taken every 16 s. On the left panel, the merged image of both imaging channels is presented, on the right, the LifeAct-Ruby (magenta; expression driven by c135-GAL4) is presented on top, the Glue-GFP (green; endogenous promoter) is presented in the middle, and the merged on the bottom. Scale bar, 20 μm (left) and 1 μm (right).

**Video 5.** **Myosin II is essential for efficient pore expansion and essential for constriction.** Time-lapse videos of representative LSVs (SRRF intensity-projection) from WT untreated SG undergoing membrane crumpling (MC; top), and from Zipper[KD] (myosin II heavy chain) and Y-27632 treated SG, undergoing stalling (middle and bottom; corresponds to data presented in Fig. 3, B and C). The video runs twice, the first run is unmarked and in the second run, the pores are outlined by two white curved lines on the Glue-GFP image (middle column), and the maximal diameter and its occurring time is noted on the top for each of the three LSVs. In the WT, untreated SG, the pore expands (up to 1.8 μm at 00:48) and constricts quickly, as the LSV undergoes membrane crumpling until it cannot be resolved (03:28). In the stalled LSV from Zipper[KD] SG, the pore expands slower (reaching a maximal diameter of 1.2 μm at 01:15) then it constricts partly and slowly, still visible at the end of the video almost 5 min after fusion (04:45). Lastly, the pore of the stalled LSV from the Y-27632 treated SG, expands with the slowest rate out of the three LSVs presented (reaching a maximal pore diameter of 1.5 μm at 03:12) and then arrests without constricting, still seen expanded almost 6 min after fusion (05:52). Time mm:ss; relative to fusion, seen in the second frames. Video frames were taken every 16 (for WT and Y-27632) or 15 s (for Zipper[KD]). LifeAct-Ruby (magenta; left column; expression driven by c135-GAL4) and Glue-GFP (green; middle column; endogenous promoter). The merged image of both imaging channels is presented in the right column. Scale bars = 1 μm.

**Video 6.** **Myosin II localization in modes of exocytosis.** Time-lapse videos of representative LSVs in WT SG expressing Sqh-mCherry (myosin II light chain; green; endogenous promotor) undergoing membrane crumpling (MC, right column), full collapse (FC, second column), kiss-and-run (KAR w or w/o actomyosin, third and fourth columns; corresponds to Fig. 3 D). LSVs are visible in the background of the cytoplasmic Sqh-mCherry signal and vesicle swelling severs as a proxy for fusion as in Glue-GFP. In MC, similarly to actin, myosin is recruited to the LSV after fusion (01:00), followed with content release and membrane crumpling (01:00–03:30). In FC, Sqh-mCherry is recruited after vesicle integration (01:45). In KAR w actomyosin, vesicles undergo deformation and displacement from the apical membrane upon Sqh-mCherry recruitment (01:45–04:15). In KAR w/o actomyosin, vesicles fuse multiple times before recruiting Sqh-mCherry (00:00 and 02:30) and proceeding to membrane crumpling. Time mm:ss; relative to fusion, seen in the second frames. Video frames were taken every 15 s. Scale bars = 1 μm.

Video 7. **Compound exocytosis is apparent in BAR domain protein KDs.** Time-lapse video of representative compound exocytosis events (SRRF intensity projection) from SNX1[KD] (left) and SNX6[KD] (right; from Fig. S3 B bottom) SGs. Compound exocytosis events are characterized by LSVs that fuse consecutively to an LSV fused to the cell membrane, forming a large membrane structure at the cell surface (in SNX6[KD]) or if FC is also prevalent (as in SNX1[KD]), the LSV membranes are integrating to the cell membrane one after the other. LifeAct-Ruby (magenta; expression driven by c135-GAL4) and Glue-GFP (green; endogenous promoter). RNAi expressed under UAS control. UAS expression driven by *fkh*-Gal4. Time mm:ss; from the beginning of the video. Video frames were taken every 13 s. Scale bars = 10 μm.

Video 8. **MIM-Emerald localizes to dynamic cytoplasmic clusters.** Time-lapse video of representative MIM-Emerald (magenta; a functional MIM allele Fig. S5 B; expression driven by c135-GAL4) cluster (confocal Z-projection) in the cytoplasm surrounded by LSVs (marked by Glue-DsRed [Costantino et al., 2008]; green; endogenous promoter). The cluster merges with another cluster. Time mm:ss; relative to the start of the video. Video frames were taken every 16 s. Scale bar 1 μm.

Video 9. **MIM-Emerald localizes to the site of fusion and pore formation.** Time-lapse of representative LSV (SRRF Z projection; from Fig. 5 A) showing MIM localization to the sites of fusion and pore formation, remaining throughout secretion. Glue-DsRed (Costantino et al., 2008; green; endogenous promoter), MIM-Emerald (magenta; a functional MIM allele Fig. S5 B; expression driven by c135-GAL4). Time mm:ss; relative to fusion, seen in the second frame. Video frames were taken every 16 s. Scale bar = 1 μm.

Video 10. **Pore stabilization by MIM is not essential for myosin II recruitment and contractility.** Time-lapse video of representative LSV in MIM[ΔIBAR/+] SGs undergoing full collapse (FC; confocal intensity-projection; corresponds to Fig. S5 C bottom). Sqh-mCherry (myosin II light chain; green; endogenous promotor). The pore quickly expands, and content is released before Sqh localization to the vesicle, while the vesicle appears to integrate into the apical membrane (up to 01:20). Following apparent integration and complete content release, Sqh is localized to the "flat" vesicle membrane and contractile activity can be observed (01:36–03:36). Time mm:ss; relative to fusion; seen in the second frames. Video frames were taken every 12 s. Scale bar = 1 μm.

**Provided online is Table S1. Table S1 lists all the fly lines used in the study including accession numbers or relevant references.**

