## [Peer Review File · The Journal of Cell Biology]

Fusion pore dynamics of large secretory vesicles define a distinct mechanism of exocytosis

Tom Biton, Nadav Scher, Shari Carmon, Yael Elbaz-Alon, Eyal Schejter, Ben-Zion Shilo, and Ori Avinoam

Corresponding Author(s): Ori Avinoam, Weizmann Institute of Science and Ben-Zion Shilo, Weizmann Institute of Science

Review Timeline:

Submission Date:	2023-03-02
Editorial Decision:	2023-05-02
Revision Received:	2023-07-06
Editorial Decision:	2023-08-18
Revision Received:	2023-08-23

Monitoring Editor: Satyajit Mayor

Scientific Editor: Dan Simon

Transaction Report:

DOI: <https://doi.org/10.1083/jcb.202302112>

May 2, 2023

Re: JCB manuscript #202302112

Dr. Ori Avinoam
Weizmann Institute of Science
Department of Biomolecular Sciences
Hertzel 234
Rehovot 7610001
Israel

Dear Dr. Avinoam,

Thank you for submitting your manuscript entitled "Fusion pore dynamics of large secretory vesicles defines a distinct mechanism of exocytosis." The manuscript was assessed by expert reviewers, whose comments are appended to this letter. You will see that the reviewers are enthusiastic about your study and feel that it provides an important advance in our understanding of exocytosis of large secretory vesicles. We invite you to submit a revision having thoroughly addressed the reviewers' key concerns, as outlined here.

Reviewer #3 asks for a few additional experiments to confirm the role of myosin II by depletion in addition to studies with ROCK inhibitor Y-27632 and to show that endogenous MIM localizes to the site of the fusion pores. Characterization of the other BAR domain proteins that were identified in the screen would certainly be interesting, as noted by Reviewer #1, especially since some of the proteins show similar phenotypes as the one you have chosen to explore (e.g. CIP4 and SNX1; see Fig 4). If these studies are beyond the scope of this manuscript, at least an explanation why these proteins that do not localize to the pore affect the exocytic process in a similar way as MIM, should be provided. Since *Drosophila* myosin is insensitive to blebbistatin (Rev2) this is also not required for the revision. The rest of the reviewers' detailed comments ask for improvements in data presentation and text clarity. All of these comments should be thoroughly addressed.

GENERAL GUIDELINES:

Text limits: Character count for an Article is < 40,000, not including spaces. Count includes title page, abstract, introduction, results, discussion, and acknowledgments. Count does not include materials and methods, figure legends, references, tables, or supplemental legends.

Figures: Articles may have up to 10 main text figures. Figures must be prepared according to the policies outlined in our Instructions to Authors, under Data Presentation, <https://jcb.rupress.org/site/misc/ifora.xhtml>. All figures in accepted manuscripts will be screened prior to publication.

*****IMPORTANT:** It is JCB policy that if requested, original data images must be made available. Failure to provide original images upon request will result in unavoidable delays in publication. Please ensure that you have access to all original microscopy and blot data images before submitting your revision. ***

Supplemental information: There are strict limits on the allowable amount of supplemental data. Articles may have up to 5 supplemental figures. Up to 10 supplemental videos or flash animations are allowed. A summary of all supplemental material should appear at the end of the Materials and methods section.

Please note that JCB now requires authors to submit Source Data used to generate figures containing gels and Western blots with all revised manuscripts. This Source Data consists of fully uncropped and unprocessed images for each gel/blot displayed in the main and supplemental figures. Since your paper includes cropped gel and/or blot images, please be sure to provide one Source Data file for each figure that contains gels and/or blots along with your revised manuscript files. File names for Source Data figures should be alphanumeric without any spaces or special characters (i.e., SourceDataF#, where F# refers to the associated main figure number or SourceDataFS# for those associated with Supplementary figures). The lanes of the gels/blots should be labeled as they are in the associated figure, the place where cropping was applied should be marked (with a box), and molecular weight/size standards should be labeled wherever possible. Source Data files will be made available to reviewers during evaluation of revised manuscripts and, if your paper is eventually published in JCB, the files will be directly linked to specific figures in the published article.

Source Data Figures should be provided as individual PDF files (one file per figure). Authors should endeavor to retain a

minimum resolution of 300 dpi or pixels per inch. Please review our instructions for export from Photoshop, Illustrator, and PowerPoint here: <https://rupress.org/jcb/pages/submission-guidelines#revised>

The typical timeframe for revisions is three to four months. While most universities and institutes have reopened labs and allowed researchers to begin working at nearly pre-pandemic levels, we at JCB realize that the lingering effects of the COVID-19 pandemic may still be impacting some aspects of your work, including the acquisition of equipment and reagents. Therefore, if you anticipate any difficulties in meeting this aforementioned revision time limit, please contact us and we can work with you to find an appropriate time frame for resubmission. Please note that papers are generally considered through only one revision cycle, so any revised manuscript will likely be either accepted or rejected.

Thank you for this interesting contribution to Journal of Cell Biology. You can contact us at the journal office with any questions, cellbio@rockefeller.edu or call (212) 327-8588.

Sincerely,

Satyajit Mayor, PhD
Monitoring Editor
Journal of Cell Biology

Dan Simon, PhD
Scientific Editor
Journal of Cell Biology

Reviewer #1 (Comments to the Authors (Required)):

Biton et al JCB review

This manuscript focuses on the pore dynamics that regulate exocrine secretion in the *Drosophila* salivary gland. It follows a series of beautiful contributions on other aspects of the secretion process, one of which showed how aspects of secretion of large vesicles is conserved in mouse exocrine pancreas. It is of general interest of the readership of the Journal of Cell Biology and fits nicely in the tradition started by George Palade and his co-workers and extended to the molecular by Blobel, Rothman, Schekman and Sudof, among others.

Here, the authors show that the pore formed between the fused vesicle and the lumen of the salivary gland is dynamic. Upon fusion (marked by swelling and rounding of the secretory vesicle) the pore expands for ~ 1 min to reach a diameter of ~1.6 μm . The vesicle begins to recruit actomyosin, then contracts over the course of ~ 2 min to below the diffraction limit of the SRRF imaging system they use. They augment analysis of pore diameter with Focused ion beam/scanning EM to evaluate pore diameters of vesicles. They rightly conclude that regulation of pore diameter is an active process. The authors go on to show that expansion and stabilization requires Arp2/3 function while contraction requires nonmuscle myosin 2. They also evaluate the fate of fused vesicles: a majority of them undergo membrane crumpling (see below for my comment regarding the fraction that ultimately undergoes membrane crumpling is a bit ambiguous), but even in wild type salivary glands a fraction "kiss and run" or undergo full collapse. Drug treat and RNAi knockdowns designed to compromise Arp2/3 or myosin indicate that the function of both proteins is required for the membrane crumpling mode of secretion to predominate. Moreover, activities are required for normal pore dynamics. The authors describe treatments that cause vesicles to "jump back" (but as described below, it was not clear to me what exactly that means). The authors next go on to generate a list of candidate Bar proteins that are expressed in salivary glands and test by RNAi knock which might contribute to exocrine vesicle secretion. They show that knock down of 5 of 11 candidate bar domain proteins via RNAi elicits a pore phenotype (though another 2 elicit compound fusion events). They focus on three of the five (or seven) for further characterization showing that CIPP4 increases KAR events, therefore contributes to pore expansion, while SNX1 an MIM increase FC events and are important for pore stabilization. They show that the MIM FC phenotype is rescued by preventing pore expansion (by inhibiting Arp2/3 - but they apparently didn't test SNX1). They also show that FC doesn't require actomyosin if pore stability is compromised. They go on to show that MIM protein localizes to the fusion site and the pore throughout the secretion event and pore expansion and stabilization depends on MIM function in a dose dependent fashion. They next dissect MIM function by showing that MIM's I-BAR domain is necessary but not sufficient for MIM function. They show that a MIM construct with the I-BAR domain deleted functions as a dominant negative, as does a MIM - I-BAR protein. Nevertheless, myosin is recruited to the secretory vesicles independent of MIM function.

Overall, the manuscript is well organized and written. I do have specific comments/questions that should be addressed before it can be published. In all cases I think the authors can provide a fix through more detailed description or corrected writing, though some further characterization of some of the "other BAR" proteins (that the authors may already have in hand would strengthen the manuscript. In final form it should make a nice contribution to the Journal of Cell Biology.

General comments:

Whether or not this is a choice of the authors or a JCB convention the moving images shown are not videos - that technology was surpassed years if not decades ago. From the materials and methods, individual images were obtained via digital (not video) imaging, so that the "moving images" are better called "movies".

Figure and Video Legends. My understanding is that many if not all of these images are generated by taking a stack of confocal or SRRF planes, then generating a XZ view. The images are impressive. The authors should explicitly state whether or not the image shown is XY or XZ (or YZ) so that the reader has an accurate understanding of what they are looking at (the strategy they used in a previous publication, Kamalesh et al 2021, Fig 1B is an ideal way of making it clear to the reader what view is being used and is less ambiguous than the method used in this manuscript, i.e., "Z-projection").

Do the authors really need to use so many acronyms? Seems that the reader would have an easier time of it if only a few were used.

Specific Comments:

Introduction:

Line 46-49: The authors assert:

"We have also shown that diffusion between the vesicle and apical membranes becomes restricted after fusion, thereby maintaining apical membrane composition. These observations suggested that membrane homeostasis in exocrine cells is maintained by mechanochemical sequestration of the LSV membrane (Kamalesh et al., 2021) (Fig. 1 A)."

I am confused: In Rousso et al 2016 Nat Cell Bio 18:181 the authors provide evidence that mixing of lipids from the plasma membrane effect the assembly of actin on the surface of the vesicle. As written, the two observations seem to me in conflict. My guess is that this is more of a writing issue rather than a conceptual one - regardless it should be adjusted to accurately reflect the authors' current view of membrane homeostasis and diffusion between the plasma membrane and the vesicle.

Line 64-66: The authors state:

"Under conditions where the fusion pore was destabilized, vesicle content release took place even in the absence of actomyosin, underscoring the presence of the future pore as the cause for the employment of actomyosin contractility during content release."

I don't understand the logic here:

1. When the fusion pore is destabilized,
2. Vesicle content release took place in the absence of actomyosin

So this

3. Underscores the presence of the future pore as the cause for employment of actomyosin contractility for content release.

I simply don't get how 3 follows from 1 and 2. The authors presumably understand their logic but the sentence as written simply does not convey that logic, at least to this reader/reviewer.

The authors should try again, perhaps expanding to another clause or sentence to make sure it makes sense to all readers.

Line 89: States that actomyosin assembly begins when the pores reach their maximal diameter. I would change the annotation for the red dot in Figure 1D from "Actomyosin Assembly" to "Onset of Actomyosin Assembly" OR change the red line to show where temporally actomyosin assembly and contraction are occurring.

Figure 1. Although mean values for various measured parameters are available in the text, the authors would help the reader by explicitly labeling the calculated mean values in Panel C (diameter and roundness) and Panel E (pore diameter for pre-crumpled and crumpled). An alternative may be to put the numbers in the legend, but I think right on the Panel is preferable. Of course there is redundancy with the text, but it is nevertheless helpful.

Lines 134 to 136. I am trying to unambiguously understand the percentages shown in the text with respect to the bar chart in Figure 2C. Do all KAR events end up with MC? The writing is ambiguous.

Figure 2. I don't fully understand what is happening in this figure.

Panel A, Upper. This is the most straight forward. Actomyosin is recruited after a second fusion event and it appears as if the vesicle begins to crumple. Does it ultimately finish crumpling? It is not important for the authors to show the end point, but the information would be useful.

Panel A, Middle. The authors state: "KAR w. actomyosin, appearing on LSVs that 'jump back' away from the apical membrane

(Arrowhead and Dashed line). The same LSV fused again at 11:28." Do the arrowhead and dashed line indicate the apical membrane? If so, I am not understanding how it could. Do they indicate an LSV that jumps back? If so I don't see that either. It seems these are two vesicles that are interacting. One with actomyosin seems to crumple, the other without actomyosin continues to grow. The figure may indeed show what the authors assert it does, but their legend and annotations on the micrographs need to be expanded so as to better guide the reader through it. Perhaps a diagram of what "jump back" is would be helpful (i.e., something like what is shown in Figure 2B)?

Panel A, Bottom. The authors state: "(Bottom) FC appearing as content release that precedes actomyosin assembly. The fusion pore expands, and the vesicular 41 membrane is integrated into the apical surface. Glue-GFP (Green) and LifeAct-Ruby (Magenta).

I see that the vesicle extrudes its Glue-GFP contents, but the pore delineated by the arrowheads doesn't seem to expand by much (or not at all)? If it is expanding, putting a measurement of the pore in the 3 marked frames would convince the reader. I also don't see how the authors know the fate of the membrane as they are not marking membrane in this experiment.

Panel B. I don't see how the middle panel is KAR with actomyosin. The vesicle with an actin coat appears to contract to expel its contents (either into the lumen or into the other vesicle to which it is attached) and the other vesicle, without actomyosin seems to grow. Is the compound configuration of vesicles shown really the best to schematize KAR with actomyosin?

The legend to Video 3 should be expanded highlight the time in minutes and seconds when key events occur to better inform the reader about how the authors interpret the video. Alternatively the legend could better refer the reader back to the legend for the appropriate panel in Figure 2 to find the times of key events in the video.

Line 182. It is not clear why the authors chose to focus on just 3 of the candidate genes that perturbed exocytosis (indeed, more than 7 of their candidates affected secretion if they were to count instances of compound exocytosis. They should better justify why they chose the three they did.

Figure Legend 5A. Line 969 should read "... the magnified areas. It's not clear why the authors would render dsRed in green and Emerald in magenta.

Video 8. The compound fusion events are easy to spot in the two left hand panels. I find it difficult to understand what I am supposed to see in the right hand panels. Asterisks or arrowheads may help for the upper right, a higher magnification view of a smaller region of the lower right may be more convincing (I think the sequence shown is consistent with compound exocytosis, but the

Figure S1. Aligned and Alignment are both spelled wrong in the figure (they are spelled correctly in the legend).

The legends to Fig. S1 and S4 are properly noted, the Legend to 2S and 3S have the letter S out of order.

Methods: I did not read these carefully. Because of the mistake I found below, I recommend that the authors carefully review them and verify that what they have written is correct.

Lines 710 to 711.

The authors state "The following fluorescence excitation and emission filter sets were used: 525/50 nm for GFP, EGFP and Emerald and 609/54 nm for Ruby, mCherry, DsRed and mScarlet." My guess is that the numbers stated are the emission filter central wavelength and 1/2 height width, as all of the numbers are close to the emission wavelengths. This needs to be corrected. Also, the authors should try to be a bit more specific about the fluorescent proteins. In several instances there are multiple versions of the proteins and the description is not sufficient unless a reference is provided (ok for them to use Bloomington stock numbers if available).

Reviewer #2 (Comments to the Authors (Required)):

This paper examines the mechanisms regulating pore dynamics of large secretory vesicles (LSV) from *Drosophila* larval salivary granules using SRRF, spinning disk imaging, FIB-SEM and CLEM. The conclusions drawn are:

- (i) Arp2/3 and MIM I-BAR are required for pore expansion and stabilization.
- (ii) Myosin II is required for pore constriction.
- (iii) MIM acts with actin and myosin II and additional BAR domain proteins to control fusion pore dynamics.

The paper is not that easy to follow for a number of reasons. First the imaging is rather fuzzy and this makes it difficult to see where the fusion pore they are describing in the text is in the figure. Second, I was confused by the use of LifeAct as a marker of actomyosin when they (correctly) state that LifeAct binds F-actin. This is compounded by the figure legends that refer to 'actomyosin' while the figures are labelled 'LifeAct'. Thirdly graphs such as 1D showing pore size as % max would provide more information if they simply showed the pore sizes.

In Figure 3 I wondered why the authors did not use blebbistatin to disrupt myosin II to address the role of myosin II in fusion pore restriction. As a non-expert in fly salivary gland secretion, it would be great if the figures here could be easier to follow. Perhaps they could reference 'sqh' as the fly MyoII in the legend as well as the text. Consistency between legend, text and figures would be enormously helpful. I also continued to find the reference to 'actomyosin' in the figures confusing, particularly in the 3rd line down of 3D that says it's with actomyosin and yet there is no signal for LifeAct that they refer to as the marker for actomyosin.

Figure S3A shows the results of a screen for BAR-domain proteins that affect secretion, but I cannot find any description of how the screen was carried out and what the final read-out was. Apologies if I missed this. I was also puzzled that there did not seem to be any validation of the genetic deletions.

Reviewer #3 (Comments to the Authors (Required)):

The manuscript entitled "Fusion pore dynamics of large secretory vesicles defines a distinct mechanism of exocytosis" by Biton et al. studies the mechanism of fusion pore expansion and compression in a cell model for exocytosis of large secretory vesicles in *Drosophila* salivary glands. Previous work has shown that glue-containing vesicles in *Drosophila* salivary glands acquire an actin coat which together with myosin II contributes to vesicle compression and content release. In this paper the authors investigate the expansion and compression of fusion pores of individual vesicles and identify Arp2/3 and BAR homology proteins to be involved in this process. They also show that MIM, one of the BAR homology proteins, localizes to fusion pores and demonstrate that inhibition of MIM function changes the fusion pore dynamic.

Overall, this is a very interesting and relevant study which can provide better understanding of exocytosis in large secretory vesicles. The conclusions of the study are generally very well supported by evidence. The paper convincingly demonstrates the dynamic of the fusion pore during exocytosis and involvement of Arp2/3 nucleation factor and MIM in this dynamic. There are a few points that should be addressed to substantiate the findings more:

- Fig1 D shows the dynamic change of fusion pore diameter during exocytosis. The assembly of actin coat is only indicated as a point on the graph. The relationship between fusion pore dynamics and actin coat assembly would be more convincing if the trace of actin assembly on the vesicles was also shown on the same graph to demonstrate how they coincide.
- Figure 3 shows only myosin recruitment, however the text and figure labels refer to myosin assembly relative to actin assembly. It seems that actin was not visualised together with myosin for that experiment. For these particular fusion events the conclusions should be stated more carefully, without reference to actin, but only with reference to the fusion phenotype. Also, myosin II involvement in fusion pore constriction is only evidenced by Y-inhibitor, it would be good to demonstrate that the same happens with e.g. myosin II KD
- When demonstrating the role of MIM it would be good to verify the patchy localisation of the protein at the site of the fusion pores for the endogenously expressed protein using immunofluorescence

Minor points:

- for the first chapter of the results (fig 1), the mean size of vesicles that were measured should also be stated, not just the diameter of the pore. Also it would be good to state in the methods how the roundness of the vesicles was measured
- line 386 states that I-BAR localized to basolateral membrane, whereas the results state it is on the apical and lateral membrane (line 258)
- line 700 in methods suggests that 35-mm glass-bottom dishes contained 100 ml of medium, normally they can contain ~2ml
- line 863 states that " $ns \geq 0.05$ and " $* \leq 0.05$ ", it should probably be $ns > 0.05$
- last p-value in line 912 (for Arp3KD) is not defined; it probably refers to FC?
- asterisks on Fig3D don't seem to be defined in the figure legend. Also, it would be good if concentration of CK666 used for measuring pore diameter in Fig 3C was written in figure legend
- Fig 4A suggests that SNX1 KD increased frequency of KAR events, but this does not appear to be the case on the graphic representation for SNX1 KD on fig 4B where only FC seems increased compared to control. Also, the effect of MIM KD / CD666 shown on fig 4C should be quantified (as done for Y27632 on fig 4D)
- on Fig 5B (right) it would be good to indicate the location of the plasma membrane
- Fig 6A figure legend line 984 should read dark grey, not dark ray

We thank the reviewers for their insightful comments and suggestions. In the following, we provide a point-by-point reply detailing the changes we have made according to their suggestions, which we believe improved the manuscript considerably.

Reviewer #1

This manuscript focuses on the pore dynamics that regulate exocrine secretion in the *Drosophila* salivary gland. It follows a series of beautiful contributions on other aspects of the secretion process, one of which showed how aspects of secretion of large vesicles is conserved in mouse exocrine pancreas. It is of general interest of the readership of the *Journal of Cell Biology* and fits nicely in the tradition started by George Palade and his co-workers and extended to the molecular by Blobel, Rothman, Schekman and Sudof, among others.

Here, the authors show that the pore formed between the fused vesicle and the lumen of the salivary gland is dynamic. Upon fusion (marked by swelling and rounding of the secretory vesicle) the pore expands for ~ 1 min to reach a diameter of ~1.6 μm . The vesicle begins to recruit actomyosin, then contracts over the course of ~ 2 min to below the diffraction limit of the SRRF imaging system they use. They augment analysis of pore diameter with Focused ion beam/scanning EM to evaluate pore diameters of vesicles. They rightly conclude that regulation of pore diameter is an active process. The authors go on to show that expansion and stabilization requires Arp2/3 function while contraction requires nonmuscle myosin 2. They also evaluate the fate of fused vesicles: a majority of them undergo membrane crumpling (see below for my comment regarding the fraction that ultimately undergoes membrane crumpling is a bit ambiguous), but even in wild type salivary glands a fraction "kiss and run" or undergo full collapse. Drug treat and RNAi knockdowns designed to compromise Arp2/3 or myosin indicate that the function of both proteins is required for the membrane crumpling mode of secretion to predominate. Moreover, activities are required for normal pore dynamics. The authors describe treatments that cause vesicles to "jump back" (but as described below, it was not clear to me what exactly that means).

The authors next go on to generate a list of candidate Bar proteins that are expressed in salivary glands and test by RNAi knock which might contribute to exocrine vesicle secretion. They show that knock down of 5 of 11 candidate bar domain proteins via RNAi elicits a pore phenotype (though another 2 elicit compound fusion events). They focus on three of the five (or seven) for further characterization showing that CIPP4 increases KAR events, therefore contributes to pore expansion, while SNX1 an MIM increase FC events and are important for pore stabilization. They show that the MIM FC phenotype is rescued by preventing pore expansion (by inhibiting Arp2/3 – but they apparently didn't test SNX1). They also show that FC doesn't require actomyosin if pore stability is compromised. They go on to show that MIM protein localizes to the fusion site and the pore throughout the secretion event and pore expansion and stabilization depends on MIM function in a dose dependent fashion. They next dissect MIM function by showing that MIM's I-BAR domain is necessary but not sufficient for MIM function. They show that a MIM construct with the I-BAR domain deleted functions as a dominant negative, as does a MIM - I-BAR protein. Nevertheless, myosin is recruited to the secretory vesicles independent of MIM function.

Overall, the manuscript is well organized and written. I do have specific comments/questions that should be addressed before it can be published. In all cases I think the authors can provide a fix through more detailed description or corrected writing, though some further

characterization of some of the "other BAR" proteins (that the authors may already have in hand would strengthen the manuscript. In final form it should make a nice contribution to the Journal of Cell Biology.

We thank the reviewer for their insightful comments and positive feedback on the manuscript. It is encouraging to learn that they recognize the significance of the study and its relevance to the Journal of Cell Biology. In response to their comments, we have critically re-examined the manuscript and made several additions and amendments, as detailed below, to clarify the research more precisely. We believe that these revisions and clarifications have strengthened our manuscript and made our study more solid and precise.

General comments:

Whether or not this is a choice of the authors or a JCB convention the moving images shown are not videos - that technology was surpassed years if not decades ago. From the materials and methods, individual images were obtained via digital (not video) imaging, so that the "moving images" are better called "movies".

Indeed "Video" is a JCB convention.

Figure and Video Legends. My understanding is that many if not all of these images are generated by taking a stack of confocal or SRRF planes, then generating a XZ view. The images are impressive. The authors should explicitly state whether or not the image shown is XY or XZ (or YZ) so that the reader has an accurate understanding of what they are looking at (the strategy they used in a previous publication, Kamalesh et al 2021, Fig 1B is an ideal way of making it clear to the reader what view is being used and is less ambiguous than the method used in this manuscript, i.e., "Z-projection").

Indeed, all images were acquired as a stack of XY planes and projected to generate the XZ view, which is shown in the figures and movies. Following the reviewer's advice, we have included a schematic in Fig 1 A, similar to the approach we used in our previous publication (Kamalesh et al, 2021), to better illustrate our methodology.

Do the authors really need to use so many acronyms? Seems that the reader would have an easier time of it if only a few were used.

We have revised the manuscript to minimize the use of acronyms. Specifically, we have removed the acronyms MC, FC, and KAR from the main text. They are now only used in the figures and figure legends for clarity. We believe this revision has significantly improved readability and ease of understanding for the reader.

Specific Comments:

Introduction:

Line 46-49: The authors assert:

"We have also shown that diffusion between the vesicle and apical membranes becomes restricted after fusion, thereby maintaining apical membrane composition. These observations

suggested that membrane homeostasis in exocrine cells is maintained by mechanochemical sequestration of the LSV membrane (Kamalesh et al., 2021) (Fig. 1 A)." I am confused: In Rousso et al 2016 Nat Cell Bio 18:181 the authors provide evidence that mixing of lipids from the plasma membrane effect the assembly of actin on the surface of the vesicle. As written, the two observations seem to me in conflict. My guess is that this is more of writing issue rather than a conceptual one - regardless it should be adjusted to accurately reflect the authors' current view of membrane homeostasis and diffusion between the plasma membrane and the vesicle.

We have revised the text to reflect our current understanding of membrane homeostasis and diffusion dynamics more accurately. Lines 46-50 now read: "We have also shown that following fusion, the diffusion between the vesicle and apical membranes becomes quickly restricted, permitting only limited initial membrane mixing, while predominantly maintaining the composition of the apical membrane. These observations suggested that membrane homeostasis in exocrine cells is maintained by mechanochemical sequestration of the LSV membrane (Kamalesh et al., 2021; Rousso et al., 2016) (Fig. 1 A)."

Line 64-66: The authors state:

"Under conditions where the fusion pore was destabilized, vesicle content release took place even in the absence of actomyosin, underscoring the presence of the future pore as the cause for the employment of actomyosin contractility during content release."

I don't understand the logic here:

1. When the fusion pore is destabilized,
2. Vesicle content release took place in the absence of actomyosin

So this

3. Underscores the presence of the future pore as the cause for employment of actomyosin contractility for content release.

I simply don't get how 3 follows from 1 and 2. The authors presumably understand their logic but the sentence as written simply does not convey that logic, at least to this reader/reviewer. The authors should try again, perhaps expanding to another clause or sentence to make sure it makes sense to all readers.

We have revised the sentence to provide a clearer explanation of the relationship between fusion pore stabilization, actomyosin contractility, and the mechanism of exocytosis. Lines 65-68 now read: "When the fusion pore fails to stabilize, vesicle content is released by full collapse in the absence of actomyosin. This highlights the importance of stabilizing the fusion pore to prevent full collapse, which then allows for the recruitment and contraction of an actomyosin network, in order to mediate exocytosis by membrane crumpling." We believe that this revision better conveys the logical sequence and clarifies the role of fusion pore stabilization in determining the need for actomyosin contractility during exocytosis.

Line 89: States that actomyosin assembly begins when the pores reach their maximal diameter. I would change the annotation for the red dot in Figure 1D from "Actomyosin Assembly" to "Onset of Actomyosin Assembly" OR change the red line to show where temporally actomyosin assembly and contraction are occurring.

We accepted the suggestion and changed the text and label in Figure 1D to "Onset of Actomyosin Assembly"

Figure 1. Although mean values for various measured parameters are available in the text, the authors would help the reader by explicitly labeling the calculated mean values in Panel C (Δ diameter and Δ roundness) and Panel E (pore diameter for pre-crumpled and crumpled). An alternative may be to put the numbers in the legend, but I think right on the Panel is preferable. Of course there is redundancy with the text, but it is nevertheless helpful.

We accepted the suggestion and added the mean and SEM values to relevant panels. Additionally in other figure panels throughout the manuscript we added relevant mean and SEM values to the legends that were not mentioned.

Lines 134 to 136. I am trying to unambiguously understand the percentages shown in the text with respect to the bar chart in Figure 2C. Do all KAR events end up with MC? The writing is ambiguous.

We thank the reviewer for pointing out the ambiguity in our previous explanation. To clarify, all kiss-and-run (KAR) events result in an apparent detachment of the vesicle from the cell membrane without content release, allowing the vesicle to undergo subsequent fusion. Ultimately most kiss-and-run events in WT cells progress to membrane crumpling (MC), with a smaller proportion stalling or transitioning to full collapse (FC). Hence, the percentages shown in the bar chart in Figure 2C represent fusion events rather than individual vesicles. If we only quantified the terminal fate of each vesicle, the percentage of vesicles undergoing membrane crumpling would be closer to 90%, but this would not provide an accurate description of the distribution, because all modes except for kiss-and-run are terminal events. We clarify this in the text. Lines 128-134 now read: "Interestingly, detailed examination of multiple fusion events in WT glands revealed that, while membrane crumpling is the prevalent mode of exocytosis, a significant fraction of fusion events exhibits a kiss-and-run-like behavior, and a smaller cohort leads to full collapse or stalling of the LSV ($67 \pm 3\%$, $23 \pm 4\%$, $6 \pm 3\%$ and $4 \pm 1\%$ respectively; Fig. 2 C and Fig. S2 A). Importantly, unlike full collapse and stalling, kiss-and-run events are not terminal, allowing LSVs to undergo subsequent fusion and exocytosis. As a result, the majority of Kiss-and-run events in WT SGs eventually proceed to membrane crumpling exocytosis."

Figure 2. I don't fully understand what is happening in this figure.

Panel A, Upper. This is the most straight forward. Actomyosin is recruited after a second fusion event and it appears as if the vesicle begins to crumple. Does it ultimately finish crumpling? It is not important for the authors to show the end point, but the information would be useful. Panel A, Middle. The authors state: "KAR w. actomyosin, appearing on LSVs that 'jump back' away from the apical membrane (Arrowhead and Dashed line). The same LSV fused again at 11:28." Do the arrowhead and dashed line indicate the apical membrane? If so, I am not understanding how it could. Do they indicate an LSV that jumps back? If so I don't see that either. It seems these are two vesicles that are interacting. One with actomyosin seems to crumple, the other without actomyosin continues to grow. The figure may indeed show what the authors assert it does, but their legend and annotations on the micrographs need to be expanded so as to better guide the reader through it. Perhaps a diagram of what "jump back" is would be helpful (i.e., something like what is shown in Figure 2B? Panel A, Bottom. The authors state: "(Bottom) FC appearing as content release that precedes actomyosin assembly. The fusion pore expands, and the vesicular membrane is integrated into the apical surface. Glue-GFP (Green) and LifeAct-Ruby (Magenta).

I see that the vesicle extrudes its Glue-GFP contents, but the pore delineated by the arrowheads

doesn't seem to expand by much (or not at all)? If it is expanding, putting a measurement of the pore in the 3 marked frames would convince the reader. I also don't see how the authors know the fate of the membrane as they are not marking membrane in this experiment.

Panel B. I don't see how the middle panel is KAR with actomyosin. The vesicle with an actin coat appears to contract to expel its contents (either into the lumen or into the other vesicle to which it is attached) and the other vesicle, without actomyosin seems to grow. Is the compound configuration of vesicles shown really the best to schematize KAR with actomyosin?

We have made significant revisions to Figure 2 and the accompanying text to improve clarity. In Figure 2, the upper panel A now shows an example of an LSV in which the pore expands irreversibly, effectively flattening the vesicular membrane. We have adjusted our conclusions to avoid overstatement, due to the fact that we cannot directly visualize membrane integration. The revised text now acknowledges that full collapse events involve rapid and complete opening of the pore, leading to an apparent integration of the LSV into the apical membrane, resulting in content release before actomyosin recruitment. Lines 119-122 now read: "Full collapse events are characterized by pores that expand irreversibly without stabilizing, leading to rapid and complete opening of the pore, apparent integration of the LSV into the apical membrane, and content release before actomyosin recruitment (Fig. 2, A and B, Fig. S2, A and B and Videos 2 and 3 top panels)."

In the middle panel A of Figure 2, we have added dashed lines around the vesicle to indicate kiss-and-run with actomyosin recruitment. Additionally, the schematic in Figure 2B has been improved to better illustrate our interpretation. The accompanying text now describes kiss-and-run events as LSVs that fuse with the apical membrane but subsequently detach, often undergoing multiple cycles of fusion and detachment with or without actomyosin recruitment and contraction. We refer the reader to the relevant supplementary movies for further clarification. Lines 122-127 now read: "Kiss-and-run events are characterized by LSVs that fuse with the apical membrane, but subsequently detach. During such events, LSVs will often undergo multiple cycles of fusion and detachment (detected by vesicle swelling), with or without actomyosin recruitment and contraction (Fig. 2, A and B, Fig S2, A and B and Videos 2 and 3 middle and bottom panels). Conversely, actomyosin recruitment to vesicles with a narrow or sealed pore can deform and displace the vesicle from the apical membrane, with no apparent content release (Fig. 2, A and B, Fig. S2, A and B and Videos 2 and 3)."

For additional clarity, we also refer the reader to the relevant supplementary movie in the figure legend. Lastly, the fate of the vesicle shown in the original upper panel A (now lower panel) is explicitly stated in the revised figure legend.

The legend to Video 3 should be expanded highlight the time in minutes and seconds when key events occur to better inform the reader about how the authors interpret the video.

Alternatively the legend could better refer the reader back to the legend for the appropriate panel in Figure 2 to find the times of key events in the video.

The legend for Video 3 (now Videos 2 and 3) has been expanded to provide more detailed information about the timing of key events. Additionally, the legend now clearly refers the reader back to the corresponding panel in Figure 2, for further reference on the timing of these events. Additionally, throughout the figures, figure legends now refer to the relevant Videos. This ensures that the movie complements the snapshots shown in the figures, providing a

comprehensive understanding of the observed dynamics. We have also adjusted the speed of the Videos to ensure clarity and ease of observation.

Line 182. It is not clear why the authors chose to focus on just 3 of the candidate genes that perturbed exocytosis (indeed, more than 7 of their candidates affected secretion if they were to count instances of compound exocytosis. They should better justify why they chose the three they did.

We are following up on all the candidate genes, but have decided to start with these three because they represent different, major classes of BAR domain proteins. We clarified this in the text. Lines 206-208 now read: "To further analyze the contribution of different BAR domains to the regulation of fusion pore dynamics, we focused on CIP4, SNX1, and MIM, which contain F-BAR, PX-BAR and I-BAR domains, respectively (Fricke et al., 2009; Zhang et al., 2011; Quinones et al., 2010)."

Figure Legend 5A. Line 969 should read "... the magnified areas.

Corrected.

It's not clear why the authors would render dsRed in green and Emerald in magenta.

The decision to render dsRed in green and Emerald in magenta was made to maintain consistency throughout the figures, by consistently representing the cargo (Glue) in green. This choice also takes into consideration color blindness, as using magenta for Emerald allows for better differentiation for color-blind readers. The identity of the fluorescent protein used is stated in the figure legend and in the materials and methods.

Video 8. The compound fusion events are easy to spot in the two left hand panels. I find it difficult to understand what I am supposed to see in the right hand panels. Asterisks or arrowheads may help for the upper right, a higher magnification view of a smaller region of the lower right may be more convincing (I think the sequence shown is consistent with compound exocytosis, but the

To provide a clearer and more understandable representation of the phenomenon we observed, we have decided to focus solely on the left-hand panels, which clearly depict the compound fusion events and remove the right-hand panels from the Video. We acknowledge that these panels were challenging to interpret. The compound Video is now Video 7.

Figure S1. Aligned and Alignment are both spelled wrong in the figure (they are spelled correctly in the legend.

Corrected.

The legends to Fig. S1 and S4 are properly noted, the Legend to 2S and 3S have the letter S out of order.

Corrected.

Methods: I did not read these carefully. Because of the mistake I found below, I recommend that the authors carefully review them and verify that what they have written is correct.

Lines 710 to 711.

The authors state "The following fluorescence excitation and emission filter sets were used: 525/50 nm for GFP, EGFP and Emerald and 609/54 nm for Ruby, mCherry, DsRed and mScarlet." My guess is that the numbers stated are the emission filter central wavelength and 1/2 height width, as all of the numbers are close to the emission wavelengths. This needs to be corrected.

We have revised the methods as suggested by the reviewers and made several improvements and clarifications. The numbers provided indeed represent the emission filter central wavelength and 1/2 height width. We have rectified the description in the revised manuscript.

Also, the authors should try to be a bit more specific about the fluorescent proteins. In several instances there are multiple versions of the proteins and the description is not sufficient unless a reference is provided (ok for them to use Bloomington stock numbers if available).

For all fly lines Bloomington stock numbers, VDRC stock numbers or the specific reference to the original publication are now provided in table S1. We have also added references for the sequences used to generate UAS-MIM-Emerald and mScarlet, in the Methods section under "generation of transgenic flies".

Reviewer #2 (Comments to the Authors (Required)):

This paper examines the mechanisms regulating pore dynamics of large secretory vesicles (LSV) from *Drosophila* larval salivary granules using SRRF, spinning disk imaging, FIB-SEM and CLEM. The conclusions drawn are:

- (i) Arp2/3 and MIM I-BAR are required for pore expansion and stabilization.
- (ii) Myosin II is required for pore constriction.
- (iii) MIM acts with actin and myosin II and additional BAR domain proteins to control fusion pore dynamics.

The paper is not that easy to follow for a number of reasons. First the imaging is rather fuzzy and this makes it difficult to see where the fusion pore they are describing in the text is in the figure.

We appreciate the reviewer's feedback and understanding of the key findings of our study. In response to the reviewer's comment, we have made significant efforts to improve the quality of the figures and clarify the localization of the fusion pore. This includes enhancing the imaging techniques, adjusting the figure annotations, and providing additional explanations in the figure legends. We believe these revisions will greatly improve the clarity and visibility of the fusion pore, allowing for a better understanding of our results. We have revised the manuscript to minimize the use of acronyms. Specifically, we have removed the acronyms MC, FC, and KAR from the main text. They are now only used in the figures and figure legends for clarity. We believe this revision has significantly improved readability and ease of understanding for the reader.

The following revisions were made to improve the visualization of the fusion pore in the figures:

- 1. Figure 1: We have included a schematic in Figure 1A to clearly illustrate the methodology used to generate the XZ view of the fusion pore. This schematic provides a visual guide for the reader to understand the orientation of the imaging planes.*
- 2. Figure 2 A : In the upper panel, the image has been adjusted to better show an LSV undergoing full collapse (arrowheads), providing a clearer representation of the fusion pore dynamics. In the middle panel we outlined the vesicle edges, to provide a clearer representation of LSVs undergoing kiss and run. The text and annotations in Figure 2A have been expanded to provide a more detailed explanation of the observed events.*
- 3. Figure 2 B: The schematic in Figure 2B has been improved to better illustrate the process of kiss-and-run with actomyosin recruitment. This revision aims to enhance the understanding of the interactions between the vesicles, and the role of actomyosin in the contraction process.*
- 4. Figure Legends: The figure legends have been revised to include specific details about the imaging techniques used, the orientation of the images (XY or XZ), and the measured parameters, such as pore diameter and roundness.*

By incorporating these revisions, we believe that the clarity and visibility of the fusion pore in the figures have been significantly improved, facilitating a better understanding of the results.

Second, I was confused by the use of LifeAct as a marker of actomyosin when they (correctly) state that LifeAct binds F-actin. This is compounded by the figure legends that refer to 'actomyosin' while the figures are labelled 'LifeAct'.

We have made revisions to clarify the use of the F-actin probe as a proxy for actomyosin assembly. We have included a clarification in the Results section and in the figure legend of figure 1, explicitly stating that we used the F-actin marker, LifeAct, as a proxy for actomyosin. We emphasize that F-actin polymerization, detected by LifeAct, precedes and is essential for the recruitment of myosin II to the large secretory vesicles, as previously shown in Rouso et al., 2016. Lines 84-86 now read: “We used the F-actin probe as an indicator for actomyosin assembly, because F-actin polymerization precedes-and is essential for myosin II localization to the LSV (Rouso et al., 2016) (Fig. 1, A and B and Video 1)”.

Our assessment of actomyosin recruitment is not solely based on the visualization of the F-actin marker. We also consider other indicators, such as vesicle contraction and content release. In instances where we do not observe vesicle contraction, such as in Figure 2D where vesicles are stalled by Arp2/3 inhibition using CK666 treatment, we state that the vesicles are Actin-coated, rather than stating that they recruited actomyosin. Further support for our conclusions regarding actomyosin recruitment and its role in pore dynamics is given by the localization of the light chain of myosin II using the Sqh-mCherry marker, which complements the observations made with LifeAct (Fig. 3 D and Video 6).

We believe that these clarifications and revisions address the concerns raised by the reviewer and provide a more accurate and comprehensive explanation of the use of the F-actin probe as a proxy for actomyosin and the assessment of actomyosin recruitment in our study.

In Figure 3 I wondered why the authors did not use blebbistatin to disrupt myosin II to address the role of myosin II in fusion pore restriction.

We thank the reviewer for bringing up the question regarding the use of blebbistatin to disrupt myosin II in our study. We acknowledge that blebbistatin is a widely used inhibitor for myosin II in various experimental systems. However, the Drosophila Myosin-II is insensitive to blebbistatin treatment (Straight et al. 2003; Science, DOI: 10.1126/science.1081412).

To address the reviewer's concern and provide further insights into the role of myosin II in fusion pore regulation, we opted for a genetic approach specifically targeting Drosophila myosin-II. We used RNAi to downregulate the Drosophila myosin-II heavy chain (called Zipper). This genetic approach allowed us to evaluate the impact of reduced myosin-II expression on pore dynamics. Our results in Figure 3 A and B show that myosin-II heavy chain knockdown led to LSV stalling and alterations in pore dynamics, consistent with the findings from our ROCK inhibitor experiments. Lines 150-167 now read: "We have previously shown that perturbation of myosin II, either through RNAi mediated KD of the Drosophila myosin II heavy chain homolog Zipper ($Zipper^{KD}$), or treatment with the Rho-associated protein kinase (ROCK) inhibitor Y-27632 ($100\mu M$), which blocks myosin II recruitment, results in fused but stalled vesicles (Kamalesh et al., 2021; Segal et al., 2018). To investigate the role of myosin II in regulating the fusion pore, we analyzed fusion pore dynamics under these conditions. Confirming our previous findings, both $Zipper^{KD}$ and ROCK inhibition induced significant LSV stalling ($36\% \pm 4\%$ and $70\% \pm 9\%$ of fusion events, respectively; Fig. 3, A and B), consistent with the role of myosin II in actomyosin contractility and membrane crumpling.

We further examined fusion pore behavior in these stalled LSVs and observed slower expansion and constriction kinetics compared to WT, untreated glands (Fig. 3, B and C and Video 5). The typical fusion pores in WT glands expand for 68 ± 6 seconds and rapidly constrict ($n=12$). In contrast, in glands treated with the ROCK inhibitor ($Y-27632^{100\mu M}$), the pores expand significantly slower for 266 ± 38 seconds and remain expanded without constricting ($n=10$). Similarly, the fusion pores of stalled LSVs in $Zipper^{KD}$ glands expand apparently slower for 111 ± 32 seconds and do not fully constrict ($n=6$) (Fig. 3 B and Video 5). Notably, the maximal pore diameter in untreated SGs was comparable to the mean maximal pore diameter of LSVs in $Zipper^{KD}$ glands, and to the arrested diameter of the expanded pores in $Y-27632^{100\mu M}$ treated glands ($1.6 \pm 0.1 \mu m$, $1.4 \pm 0.2 \mu m$ and $1.4 \pm 0.1 \mu m$, respectively; Fig. 3 B). Taken together, these results support the involvement of myosin II in fusion pore expansion and constriction.

As a non-expert in fly salivary gland secretion, it would be great if the figures here could be easier to follow. Perhaps they could reference 'sqh' as the fly MyoII in the legend as well as the text

In order to enhance clarity and facilitate understanding for non-experts in fly genetics, we have made revisions to the figure legends and throughout the text, to explicitly reference sqh as the myosin II light chain and Zipper as the myosin II heavy chain.

Consistency between legend, text and figures would be enormously helpful.

We have carefully reviewed and revised the figures, legends, and text to ensure consistency and improve clarity. We have clarified the labeling and annotations in the figures, to provide a clear

visual representation of the key findings. Additionally, we have made adjustments to the legends, to accurately describe the content of the figures and provide a better understanding of the observed phenomena.

I also continued to find the reference to 'actomyosin' in the figures confusing, particularly in the 3rd line down of 3D that says it's with actomyosin and yet there is no signal for LifeAct that they refer to as the marker for actomyosin.

We used LifeAct as a marker for F-actin, which serves as a proxy for actomyosin assembly. As correctly mentioned by the reviewer, the third line in Figure 3D refers to the phenotype "KAR w actomyosin," which was defined in Figure 2. This phenotype indicates the presence of actomyosin, and is based on the observation of vesicular contraction and displacement from the apical membrane. To further support our findings, we also utilized sqh (myosin II regulatory light chain) as a proxy for actomyosin. Sqh is a known component of actomyosin complexes, and its recruitment to LSVs is dependent on F-actin polymerization mediated by the formin Diaphanous (Rousso et al., 2016; NCB). By examining the behavior of sqh in relation to the observed fusion pore dynamics, we gained additional insights into the involvement of actomyosin in the process. We have revised the figure legends and text to clarify that LifeAct and sqh are used as proxies for actomyosin.

Figure S3A shows the results of a screen for BAR-domain proteins that affect secretion, but I cannot find any description of how the screen was carried out and what the final read-out was. Apologies if I missed this.

We appreciate the reviewer's attention to the missing details and acknowledge the incomplete description of the screen in Figure S3A. We have revised the manuscript to provide additional details on how the screen was conducted in the "Statistics and Reproducibility" section of the Methods. Lines 939-943 now read: "Qualitative phenotypic characterization of knock-down experiments (summarized in Fig. 4 A) was based on at least 4 independent data sets using at least 2 different RNAi constructs for each gene (complete list of RNAi lines used in supplementary Table S1), and was based on visual inspection and detection of full collapse, kiss-and-run or compound exocytosis events at apparent higher frequencies than WT SGs".

To complement the screen and account for potential biases, we quantified the frequencies for three specific BAR-domain proteins: CIP4, SNX1, and MIM. This quantification was performed in Figure 4B, and provided a more quantitative assessment of the effects of these proteins on secretion. Interestingly, our results revealed distinct outcomes for each protein. Knockdown (KD) of MIM caused full collapse, KD of CIP4 causing kiss-and-run, and KD of SNX1 resulting in both phenotypes and in compound exocytosis. Albeit kiss-and-run in SNX1^{KD} was only strongly apparent in some RNAi lines and was significantly elevated when quantified (Fig. 4 A-B).

I was also puzzled that there did not seem to be any validation of the genetic deletions.

We conducted several experiments to validate the specific deletions and perturbations related to MIM. Here are the key findings that support the confidence in our results:

Phenotype consistency: We observed that both MIM knockdown (MIM^{KD}) using validated commercial RNAi lines and MIM^{Δ-IBAR/+} (specific insertion into the I-BAR sequence using the mimic

system) displayed the same phenotype. This observation, as shown in Figure 7B, indicates that the perturbations of MIM lead to consistent and comparable effects on fusion pore dynamics. *Phenotype similarity:* We further investigated the phenotype of $MIM^{null/def}$ (using a previously made and validated deletion) and $MIM^{\Delta-IBAR/-}$, as well as the overexpression of I-BAR. As depicted in Figure 7B, these different genetic manipulations yielded similar phenotypes, supporting the notion that MIM plays a crucial role in regulating fusion pore dynamics.

Rescue experiments: To strengthen our findings, we conducted rescue experiments. We observed that the loss-of-function (LOF) phenotype of $MIM^{null/def}$ and $MIM^{\Delta-IBAR/-}$ could be rescued by overexpression (OE) of MIM-Emerald, as demonstrated in Figure S5B. This rescue experiment further supports the specific contribution of MIM to the observed phenotype.

Taken together, these findings provide compelling evidence for the functional relevance of the genetic deletions and perturbations related to MIM in our study. The consistency and similarity of the phenotypes, as well as the rescue experiments, enhance our confidence in the role of MIM in regulating fusion pore dynamics.

Reviewer #3 (Comments to the Authors (Required)):

The manuscript entitled "Fusion pore dynamics of large secretory vesicles defines a distinct mechanism of exocytosis" by Biton et al. studies the mechanism of fusion pore expansion and compression in a cell model for exocytosis of large secretory vesicles in *Drosophila* salivary glands. Previous work has shown that glue-containing vesicles in *Drosophila* salivary glands acquire an actin coat which together with myosin II contributes to vesicle compression and content release. In this paper the authors investigate the expansion and compression of fusion pores of individual vesicles and identify Arp2/3 and BAR homology proteins to be involved in this process. They also show that MIM, one of the BAR homology proteins, localizes to fusion pores and demonstrate that inhibition of MIM function changes the fusion pore dynamic.

Overall, this is a very interesting and relevant study which can provide better understanding of exocytosis in large secretory vesicles. The conclusions of the study are generally very well supported by evidence. The paper convincingly demonstrates the dynamic of the fusion pore during exocytosis and involvement of Arp2/3 nucleation factor and MIM in this dynamic. There are a few points that should be addressed to substantiate the findings more:

We thank the reviewer for the positive and constructive feedback on our manuscript. In light of their feedback, we have incorporated numerous additions and adjustments to elucidate our research with greater precision. We thoroughly addressed each comment and provide detailed replies outlining the modifications made below. We believe that these revisions and clarifications have strengthened the manuscript considerably.

- Fig1 D shows the dynamic change of fusion pore diameter during exocytosis. The assembly of actin coat is only indicated as a point on the graph. The relationship between fusion pore dynamics and actin coat assembly would be more convincing if the trace of actin assembly on the vesicles was also shown on the same graph to demonstrate how they coincide.

To provide a clearer representation of this relationship, we have revised the annotation on the plot to "onset of actomyosin assembly", instead of simply indicating the assembly as a point on

the graph. This change aims to better highlight the temporal correlation between actomyosin recruitment and fusion pore dynamics. The dynamics of the actin signal on the vesicle during exocytosis have been previously demonstrated in our works (Kamalesh et al., 2021, and Rousso et al., 2016). These previous studies provide detailed information and visualizations of the actin assembly dynamics on the vesicles.

- Figure 3 shows only myosin recruitment, however the text and figure labels refer to myosin assembly relative to actin assembly. It seems that actin was not visualised together with myosin for that experiment. For these particular fusion events the conclusions should be stated more carefully, without reference to actin, but only with reference to the fusion phenotype.

We acknowledge that in the specific experiments presented in Figure 3, we did not visualize actin with myosin. To avoid any confusion, we have carefully revised the conclusions in the text and figure legends to focus solely on myosin and the observed modes of exocytosis, without reference to actin. It is important to note that myosin recruitment to the LSVs is dependent on F-actin polymerization on the vesicle, as demonstrated in our previous work (Rousso et al., 2016). Therefore, the presence of the sqh signal on the LSVs indicates the presence of F-actin as well, and we have also clarified this in the text. Lines 84-86 now read: “We used the F-actin probe as an indicator for actomyosin assembly, because F-actin polymerization precedes-and is essential for myosin II localization to the LSV (Rousso et al., 2016)“. The labels in the figure (e.g., KAR w/w/o actomyosin) are intended to indicate the different modes of exocytosis as defined in Figure 2A and 2B, rather than the direct presence of actomyosin.

Also, myosin II involvement in fusion pore constriction is only evidenced by Y-inhibitor, it would be good to demonstrate that the same happens with e.g. myosin II KD

We appreciate the reviewer's insightful suggestion to include data from myosin II knockdown to fortify our findings. As we have previously demonstrated, the RNAi-mediated knockdown of the Drosophila myosin II heavy chain homolog, Zipper (Zipper^{KD}), leads to vesicle fusion followed by stalling—a phenotype akin to Y-27632 treatment (Kamalesh et al., 2021; Segal et al., 2018). To delve deeper into the role of myosin II in modulating fusion pore dynamics, we analyzed the behavior of fusion pores in these stalled LSVs under Zipper^{KD} conditions.

Consistently, in both Zipper^{KD} and ROCK inhibitor-treated glands, we noted a pattern of decelerated pore expansion kinetics compared to wild-type (WT) untreated glands. This observation illuminated that myosin II perturbations invariably induce a substantial delay in pore expansion—a finding that we initially overlooked, which is more consistent with the role of myosin II in the literature.

Moreover, our further analysis revealed that stalled LSVs in both Zipper^{KD} glands and Y-27632-treated glands possess expanded pores with similar mean maximal pore diameters. However, constriction was only fully inhibited in the Y-27632-treated glands.

We have incorporated the data from Zipper^{KD} into the quantifications presented in Fig 3 A and B. The text from lines 158-167 has been revised to read: “We further examined fusion pore behavior in these stalled LSVs and observed slower expansion and constriction kinetics compared to WT, untreated glands (Fig. 3, B and C and Video 5). The typical fusion pores in WT glands expand for 68 ± 6 seconds and rapidly constrict ($n=12$). In contrast, in glands treated with the ROCK inhibitor

(Y-27632^{100μM}), the pores expand significantly slower for 266 ± 38 seconds and remain expanded without constricting (n=10). Similarly, the fusion pores of stalled LSVs in Zipper^{KD} glands expand apparently slower for 111 ± 32 seconds and do not fully constrict (n=6) (Fig. 3 B and Video 5). Notably, the maximal pore diameter in untreated SGs was comparable to the mean maximal pore diameter of LSVs in Zipper^{KD} glands, and to the arrested diameter of the expanded pores in Y-27632^{100μM} treated glands (1.6 ± 0.1 μm, 1.4 ± 0.2 μm and 1.4 ± 0.1 μm, respectively; Fig. 3 B). Taken together, these results support the involvement of myosin II in fusion pore expansion and constriction.”

We have also revised the discussion accordingly. Lines 377-384 now read: “Myosin II has also been implicated in pore expansion in pancreatic acinar cells and chromaffin cells utilizing large and medium sized vesicles respectively, which is consistent with the slower expansion kinetics we observed in Zipper^{KD} and Y-27632^{100μM} treated glands (Doreian et al., 2008; Ñeco et al., 2008; Bhat and Thorn, 2009). Additionally, we find that in Drosophila SGs, myosin II is essential for pore constriction in a dose dependent manner (Fig. 3). Collectively, our results support the emerging view that a combination of actin polymerization and myosin motor activity supply, at least in part, the forces needed to expand, stabilize, and constrict the LSV fusion pore, and independently to fold and retain the vesicular membrane while extruding its content.”

- When demonstrating the role of MIM it would be good to verify the patchy localisation of the protein at the site of the fusion pores for the endogenously expressed protein using immunofluorescence

Regrettably, we do not have access to an antibody for endogenous Drosophila MIM at present. However, we have provided corroborative evidence for MIM localization through alternative methodologies. As demonstrated in our work, the overexpression of MIM-Emerald can compensate for the loss-of-function phenotype, which suggests that the tagged protein maintains its original functionality (Fig. S5), and localizes correctly at the relevant sites. Furthermore, we have conducted localization studies with two distinct fluorescent-protein tags attached to MIM. Both versions of the tagged MIM consistently display localization at the fusion sites, adding further validation to our hypothesis that MIM localizes specifically to the fusion pores.

Nevertheless, we clarify this limitation of our study in the discussion section. Lines 392-395 now read: “One limitation of our study is that we cannot directly verify the localization of endogenous MIM, due to the lack of suitable reagents. Nevertheless, the ability of MIM-Emerald to compensate for the loss-of-function phenotype, suggests that the tagged MIM protein is localized correctly.”

Minor points:

-for the first chapter of the results (fig 1), the mean size of vesicles that were measured should also be stated, not just the diameter of the pore.

We state the mean size of vesicles that were measured. Lines 91-93 now read: “After fusion, LSVs with a mean diameter of 5.3 ± 0.2 μm post-swelling (n=12), displayed pore expansion for 68 ± 6 seconds, reaching a maximum mean diameter of 1.6 ± 0.2 μm. This was followed by constriction to diameters below the detection limit, typically under 100 nm.”

Also it would be good to state in the methods how the roundness of the vesicles was measured

*We have added the missing method description to the methods section of the manuscript. Lines 787-798 now read: "**Measurement of vesicle swelling – diameter and roundness**
To measure changes in vesicle diameter and roundness upon fusion as shown in Fig. 1 C, we used the polygon tool of Fiji to outline the vesicle before and after swelling is completed. In most LSVs the swelling occurs in one frame (15-18 seconds), but in some cases swelling is slower and can take up to 3 frames (45-56 seconds). We used the roundness measurement of Fiji, calculated using the formula - $4 \cdot (\text{area} / \pi \cdot \text{major-axis}^2)$, which gives a value between 0 and 1, where 1 is a perfect round object and 0 is not round. The ratio represents the relationship between an object's area and the length of the major axis of the shape. Thus, a perfect circle, a square or a polygon with equal faces is considered "round", but an ellipse or non-equal faced polygon or rectangle would be less "round". A line would have 0 roundness since all its area is spread across its major axis. To measure diameter, we used the Fit circle function in Fiji, which creates a circle with the same area and centroid of the marked polygon (the outlines of the vesicle), and we extracted the diameter of that circle."*

-line 386 states that I-BAR localized to basolateral membrane, whereas the results state it is on the apical and lateral membrane (line 258)

Thank you for bringing this error to our attention. We have revised the discussion section to accurately reflect the localization of I-BAR. It is indeed localized only to the apical and lateral membranes, as stated in line 258 of the results section.

-line 700 in methods suggests that 35-mm glass-bottom dishes contained 100 ml of medium, normally they can contain ~2ml

Corrected. The correct volume is 100 μ l

-line 863 states that "ns{greater than or equal to}0.05 and *{less than or equal to}0.05", it should probably be ns>0.05

Corrected.

-last p-value in line 912 (for Arp3KD) is not defined; it probably refers to FC?

Corrected.

-asterisks on Fig3D don't seem to be defined in the figure legend.

Corrected.

Also, it would be good if concentration of CK666 used for measuring pore diameter in Fig 3C was written in figure legend

Corrected.

-Fig 4A suggests that SNX1 KD increased frequency of KAR events, but this does not appear to be

the case on the graphic representation for SNX1 KD on fig 4B where only FC seems increased compared to control

We appreciate the reviewer's attention to the apparent discrepancy in Figure 4A and 4B. The observed increase in kiss-and-run (KAR) events in SNX^{KD} samples in Figure 4A was not consistently represented across all RNAi constructs and replicates. Therefore, while there appeared to be a trend towards an increased frequency of KAR events in the qualitative screen, this did not reach statistical significance when quantified for Figure 4B. Conversely, the increased frequency of full collapse events was consistent and statistically significant, pointing to a decrease in fusion pore stabilization in SNX^{KD}. The less penetrant kiss-and-run phenotype and the observed compound exocytosis under SNX^{KD} conditions suggest that SNX1 may have broader functions in LSV membrane dynamics, which merits further investigation. We have now clarified this in the discussion section. Lines 356-361 now read: "SNX-1^{KD} resulted in a notable decrease in fusion pore stabilization efficiency, as evidenced by an increase in the frequency of full collapse events. Although we noted an apparent increase in kiss-and-run events in some of the RNAi lines, this was not fully penetrant and consistently observed, and thus was not statistically significant in the quantification. These observations, coupled with the noted compound exocytosis in SNX-1^{KD}, may hint at a more global role of SNX-1 in LSV membrane remodeling (Fig. 4)."

. Also, the effect of MIM KD / CD666 shown on fig 4C should be quantified (as done for Y27632 on fig 4D)

The response to CK666^{500μM} treatment in our model presents a distinct phenotype characterized by an "all-or-none" effect. Unlike the distribution of fusion events seen in ROCK inhibitor treatment, where LSVs stall but may eventually manage to release their content after several rounds of actin assembly and disassembly, CK666^{500μM} treated LSVs consistently stall without progressing to secretion. Actin remains assembled on the vesicles, leading to an accumulation of stalled vesicles at the apical membrane and a rapid cessation of new fusion events. This stark and consistent response to CK666 does not lend itself to the same type of quantification applied to the Y27632 data. We are confident that the presented images convincingly illustrate the stalling effect of CK666, which was also reported previously. Our innovation lies in the measurement of pore dynamics under these stalled conditions (Fig. 2F and Fig. 3C), thereby extending the current understanding of exocytosis that has been previously reported.

-on Fig 5B (right) it would be good to indicate the location of the plasma membrane

We have added arrowheads and the label PM to the figure.

-Fig 6A figure legend line 984 should read dark grey, not dark ray

Corrected.

August 18, 2023

RE: JCB Manuscript #202302112R

Dr. Ori Avinoam
Weizmann Institute of Science
Department of Biomolecular Sciences
Hertzel 234
Rehovot 7610001
Israel

Dear Dr. Avinoam,

Thank you for submitting your revised manuscript entitled "Fusion pore dynamics of large secretory vesicles define a distinct mechanism of exocytosis." We would be happy to publish your paper in JCB pending final revisions necessary to meet our formatting guidelines (see details below). Please also consider Rev#1's remaining comments as you prepare your final files.

A. MANUSCRIPT ORGANIZATION AND FORMATTING:

1) Text limits: Character count for Articles is < 40,000, not including spaces. Count includes title page, abstract, introduction, results, discussion, and acknowledgments. Count does not include materials and methods, figure legends, references, tables, or supplemental legends.

2) Figure formatting: Articles may have up to 10 main text figures. Scale bars must be present on all microscopy images, including inset magnifications. Also, please avoid pairing red and green for images and graphs to ensure legibility for color-blind readers. If red and green are paired for images, please ensure that the particular red and green hues used in micrographs are distinctive with any of the colorblind types. If not, please modify colors accordingly or provide separate images of the individual channels.

3) Statistical analysis: Error bars on graphic representations of numerical data must be clearly described in the figure legend. The number of independent data points (n) represented in a graph must be indicated in the legend. Please, indicate whether 'n' refers to technical or biological replicates (i.e. number of analyzed cells, samples or animals, number of independent experiments). If independent experiments with multiple biological replicates have been performed, we recommend using distribution-reproducibility SuperPlots (please see Lord et al., JCB 2020) to better display the distribution of the entire dataset, and report statistics (such as means, error bars, and P values) that address the reproducibility of the findings.

Statistical methods should be explained in full in the materials and methods. For figures presenting pooled data the statistical measure should be defined in the figure legends. Please also be sure to indicate the statistical tests used in each of your experiments (both in the figure legend itself and in a separate methods section) as well as the parameters of the test (for example, if you ran a t-test, please indicate if it was one- or two-sided, etc.). Also, if you used parametric tests, please indicate if the data distribution was tested for normality (and if so, how). If not, you must state something to the effect that "Data distribution was assumed to be normal but this was not formally tested."

4) Materials and methods: Should be comprehensive and not simply reference a previous publication for details on how an experiment was performed. Please provide full descriptions (at least in brief) in the text for readers who may not have access to referenced manuscripts. The text should not refer to methods "...as previously described."

5) For all cell lines, vectors, constructs/cDNAs, etc. - all genetic material: please include database / vendor ID (e.g., Addgene, ATCC, etc.) or if unavailable, please briefly describe their basic genetic features, even if described in other published work or gifted to you by other investigators (and provide references where appropriate). Please be sure to provide the sequences for all of your oligos: primers, si/shRNA, RNAi, gRNAs, etc. in the materials and methods. You must also indicate in the methods the source, species, and catalog numbers/vendor identifiers (where appropriate) for all of your antibodies, including secondary. If antibodies are not commercial, please add a reference citation if possible.

6) Microscope image acquisition: The following information must be provided about the acquisition and processing of images:
a. Make and model of microscope

- b. Type, magnification, and numerical aperture of the objective lenses
- c. Temperature
- d. Imaging medium
- e. Fluorochromes
- f. Camera make and model
- g. Acquisition software
- h. Any software used for image processing subsequent to data acquisition. Please include details and types of operations involved (e.g., type of deconvolution, 3D reconstitutions, surface or volume rendering, gamma adjustments, etc.).

7) References: There is no limit to the number of references cited in a manuscript. References should be cited parenthetically in the text by author and year of publication. Abbreviate the names of journals according to PubMed.

8) Supplemental materials: There are strict limits on the allowable amount of supplemental data. Articles may have up to 5 supplemental figures and 10 videos. Please also note that tables, like figures, should be provided as individual, editable files. A summary of all supplemental material should appear at the end of the Materials and methods section. Please include one brief sentence per item.

9) Video legends: Should describe what is being shown, the cell type or tissue being viewed (including relevant cell treatments, concentration and duration, or transfection), the imaging method (e.g., time-lapse epifluorescence microscopy), what each color represents, how often frames were collected, the frames/second display rate, and the number of any figure that has related video stills or images.

10) eTOC summary: A ~40-50 word summary that describes the context and significance of the findings for a general readership should be included on the title page. The statement should be written in the present tense and refer to the work in the third person. It should begin with "First author name(s) et al..." to match our preferred style.

11) Conflict of interest statement: JCB requires inclusion of a statement in the acknowledgements regarding competing financial interests. If no competing financial interests exist, please include the following statement: "The authors declare no competing financial interests." If competing interests are declared, please follow your statement of these competing interests with the following statement: "The authors declare no further competing financial interests."

12) A separate author contribution section is required following the Acknowledgments in all research manuscripts. All authors should be mentioned and designated by their first and middle initials and full surnames. We encourage use of the CRediT nomenclature (<https://casrai.org/credit/>).

13) ORCID IDs: ORCID IDs are unique identifiers allowing researchers to create a record of their various scholarly contributions in a single place. Please note that ORCID IDs are required for all authors. At resubmission of your final files, please be sure to provide your ORCID ID and those of all co-authors.

14) Journal of Cell Biology now requires a data availability statement for all research article submissions. These statements will be published in the article directly above the Acknowledgments. The statement should address all data underlying the research presented in the manuscript. Please visit the JCB instructions for authors for guidelines and examples of statements at (<https://rupress.org/jcb/pages/editorial-policies#data-availability-statement>).

B. FINAL FILES:

**The license to publish form must be signed before your manuscript can be sent to production. A link to the electronic license to publish form will be sent to the corresponding author only. Please take a moment to check your funder requirements before

choosing the appropriate license.**

Thank you for this interesting contribution, we look forward to publishing your paper in Journal of Cell Biology.

Sincerely,

Satyajit Mayor, PhD
Monitoring Editor
Journal of Cell Biology

Dan Simon, PhD
Scientific Editor
Journal of Cell Biology

Reviewer #1 (Comments to the Authors (Required)):

I believe my previous review covered well the positive aspects of the work presented.

The revised version of Biton et al has now addressed my concerns regarding the initial submission. I believe that it is now acceptable for publication with two minor revisions.

There are a number of typos that the authors should fix - if they don't presumably the copy editors will. For example, on the first page of the Introduction I found two: line 27 is missing a period after the closing parenthesis and line 38 is missing a space after "kiss-and-run". I ran across more such typos.

My one request is that the authors address, at least in the discussion a proposed mechanism by which nonmuscle myosin 2 might contribute to pore expansion. They probably should refer to that discussion in the results where they show that myosin is required for pore expansion. This may simply require that the authors state that they have no simple model for the mechanism. This reviewer and most readers will be curious as to how myosin 2 might contribute to pore expansion.

August 23rd, 2023

Dr. Dan Simon
Scientific Editor, Journal of Cell Biology
950 Third Ave. New York
NY 10022

Dear Dan,

We are delighted to resubmit our manuscript entitled “**The mechanism controlling fusion pore dynamics in large secretory vesicles define a distinct mode of exocytosis**” (manuscript ID: 202302112).

Below please find a point-by-point reply to the specific points and formatting guidelines you've provided.

Regarding the remaining comment from Reviewer 1, we have added a paragraph in the discussion section as suggested by the reviewer. Lines 387-394 now read: “*We propose that myosin II regulates the fusion pore at two distinct steps by distinct mechanisms: Immediately after fusion, myosin II in concert with cortical actin reorganization favors conditions on the apical membrane that drive pore expansion while preventing the pore from resealing, which is in good agreement with previous studies. At later stages of LSV exocytosis, when the actomyosin contractile machinery compresses the vesicle to crumple the membrane and release the content, myosin II is essential for pore constriction, which might be attributed either to its contractile activity within the actomyosin meshwork coating the LSV, or to a dedicated machinery localized specifically around the pore*”. We also refer to this discussion from the Results section as the reviewer suggested.

We believe that these revisions have made the manuscript suitable for publication in The Journal of Cell Biology and look forward to your response.

Sincerely,

Ori Avinoam

Point-by-point reply

1) Text limits: Character count for Articles is < 40,000, not including spaces. Count includes title page, abstract, introduction, results, discussion, and acknowledgments. Count does not include materials and methods, figure legends, references, tables, or supplemental legends.

The text is within these limits.

2) Figure formatting: Articles may have up to 10 main text figures.

The manuscript has 8 main figures.

Scale bars must be present on all microscopy images, including inset magnifications.

Scale bars are present in all images.

Also, please avoid pairing red and green for images and graphs to ensure legibility for color-blind readers. If red and green are paired for images, please ensure that the particular red and green hues used in micrographs are distinctive with any of the colorblind types. If not, please modify colors accordingly or provide separate images of the individual channels.

The color schemes used (Green-Magenta/Green-Cyan/Magenta-Cyan) are legible for color-blind readers. Where needed we presented separate images for each channel.

3) Statistical analysis: Error bars on graphic representations of numerical data must be clearly described in the figure legend. The number of independent data points (n) represented in a graph must be indicated in the legend. Please, indicate whether 'n' refers to technical or biological replicates (i.e. number of analyzed cells, samples or animals, number of independent experiments).

The required information is present in all figure legends.

If independent experiments with multiple biological replicates have been performed, we recommend using distribution-reproducibility SuperPlots (please see Lord et al., JCB 2020) to better display the distribution of the entire dataset, and report statistics (such as means, error bars, and P values) that address the reproducibility of the findings.

We used GraphPad Prism version 8.4.2 (GraphPad software) for statistical analysis and for generating all plots.

Statistical methods should be explained in full in the materials and methods. For figures presenting pooled data the statistical measure should be defined in the figure legends. Please also be sure to indicate the statistical tests used in each of your experiments (both in the figure legend itself and in a separate methods section) as well as the parameters of the test (for example, if you ran a t-test, please indicate if it was one- or two-sided, etc.).

The required information is available in the figure legends and in a separate section in the materials and methods section.

Also, if you used parametric tests, please indicate if the data distribution was tested for normality (and if so, how). If not, you must state something to the effect that "Data distribution was assumed to be normal but this was not formally tested."

We used non-parametric tests as indicated in the figure legends and methods section.

4) Materials and methods: Should be comprehensive and not simply reference a previous publication for details on how an experiment was performed. Please provide full descriptions (at least in brief) in the text for readers who may not have access to referenced manuscripts. The text should not refer to

methods "...as previously described."

The methods are comprehensive and provide a full description.

5) For all cell lines, vectors, constructs/cDNAs, etc. - all genetic material: please include database / vendor ID (e.g., Addgene, ATCC, etc.) or if unavailable, please briefly describe their basic genetic features, even if described in other published work or gifted to you by other investigators (and provide references where appropriate).

Please be sure to provide the sequences for all of your oligos: primers, si/shRNA, RNAi, gRNAs, etc. in the materials and methods.

All oligo information and the appropriate references have been included.

You must also indicate in the methods the source, species, and catalog numbers/vendor identifiers (where appropriate) for all of your antibodies, including secondary. If antibodies are not commercial, please add a reference citation if possible.

No antibodies were used.

6) Microscope image acquisition: The following information must be provided about the acquisition and processing of images:

a. Make and model of microscope

b. Type, magnification, and numerical aperture of the objective lenses

c. Temperature

d. Imaging medium

e. Fluorochromes

f. Camera make and model

g. Acquisition software

h. Any software used for image processing subsequent to data acquisition. Please include details and types of operations involved (e.g., type of deconvolution, 3D reconstitutions, surface or volume rendering, gamma adjustments, etc.).

All the required information is available in the methods section under "Confocal image acquisition"

7) References: There is no limit to the number of references cited in a manuscript. References should be cited parenthetically in the text by author and year of publication. Abbreviate the names of journals according to PubMed.

The references are written according to the JCB format and are cited parenthetically in the text as requested.

8) Supplemental materials: There are strict limits on the allowable amount of supplemental data. Articles may have up to 5 supplemental figures and 10 videos.

We have 5 supplemental figures and 10 videos.

Please also note that tables, like figures, should be provided as individual, editable files.

Table S1 is provided as an editable .xlsx file.

A summary of all supplemental material should appear at the end of the Materials and methods section. Please include one brief sentence per item.

We have included such a summary describing each of the supplemental figures, videos and table.

9) Video legends: Should describe what is being shown, the cell type or tissue being viewed (including relevant cell treatments, concentration and duration, or transfection), the imaging method (e.g., time-lapse epifluorescence microscopy), what each color represents, how often frames were

collected, the frames/second display rate, and the number of any figure that has related video stills or images.

The video legends include all the required information.

10) eTOC summary: A ~40-50 word summary that describes the context and significance of the findings for a general readership should be included on the title page. The statement should be written in the present tense and refer to the work in the third person. It should begin with "First author name(s) et al..." to match our preferred style.

An eTOC summary was added to the title page.

11) Conflict of interest statement: JCB requires inclusion of a statement in the acknowledgements regarding competing financial interests. If no competing financial interests exist, please include the following statement: "The authors declare no competing financial interests." If competing interests are declared, please follow your statement of these competing interests with the following statement: "The authors declare no further competing financial interests."

A conflict of interest statement has been included.

12) A separate author contribution section is required following the Acknowledgments in all research manuscripts. All authors should be mentioned and designated by their first and middle initials and full surnames. We encourage use of the CRediT nomenclature (<https://casrai.org/credit/>).

Author contribution section using the CRediT nomenclature is included.

13) ORCID IDs: ORCID IDs are unique identifiers allowing researchers to create a record of their various scholarly contributions in a single place. Please note that ORCID IDs are required for all authors. At resubmission of your final files, please be sure to provide your ORCID ID and those of all co-authors.

ORCID IDs of all authors have been provided during re-submission.

14) Journal of Cell Biology now requires a data availability statement for all research article submissions. These statements will be published in the article directly above the Acknowledgments. The statement should address all data underlying the research presented in the manuscript. Please visit the JCB instructions for authors for guidelines and examples of statements at (<https://rupress.org/jcb/pages/editorial-policies#data-availability-statement>).

We have included a data availability statement.